# ARTICLES

# Lysosome lipid signalling from the periphery to neurons regulates longevity

Marzia Savini[1,2], Andrew Folick[1,3,12], Yi-Tang Lee[2,4,5], Feng Jin[2,6], André Cuevas[7], Matthew C. Tillman[7], Jonathon D. Duffy[1,2], Qian Zhao[2,4], Isaiah A. Neve[2,4], Pei-Wen Hu[2,4], Yong Yu[2,4,13], Qinghao Zhang[2,4], Youqiong Ye[8], William B. Mair[9], Jin Wang[6], Leng Han[8,10], Eric A. Ortlund[7] and Meng C. Wang[2,4,11] ✉

Lysosomes are key cellular organelles that metabolize extra- and intracellular substrates. Alterations in lysosomal metabolism are implicated in ageing-associated metabolic and neurodegenerative diseases. However, how lysosomal metabolism actively coordinates the metabolic and nervous systems to regulate ageing remains unclear. Here we report a fat-to-neuron lipid signalling pathway induced by lysosomal metabolism and its longevity-promoting role in *Caenorhabditis elegans*. We discovered that induced lysosomal lipolysis in peripheral fat storage tissue upregulates the neuropeptide signalling pathway in the nervous system to promote longevity. This cell-non-autonomous regulation is mediated by a specific polyunsaturated fatty acid, dihomo-γ-linolenic acid, and LBP-3 lipid chaperone protein transported from the fat storage tissue to neurons. LBP-3 binds to dihomo-γ-linolenic acid, and acts through NHR-49 nuclear receptor and NLP-11 neuropeptide in neurons to extend lifespan. These results reveal lysosomes as a signalling hub to coordinate metabolism and ageing, and lysosomal signalling mediated inter-tissue communication in promoting longevity.

Ageing is a process of progressive decline occurring at all levels. Mechanisms that govern the crosstalk across different organelles and among different tissues contribute to longevity regulation[1-3]. In particular, lipids are crucial signals in mediating organelle crosstalk and tissue interactions[1,4,5], and dietary supplementation of specific lipids influences lifespan[6,7]. Lysosomes actively participate in lipid metabolism, and lipid breakdown by lysosomal acid lipases releases free fatty acids (FFA) from triacylglycerols (TAGs) and cholesteryl esters (CEs)[8]. Lysosomes also serve as a signalling hub inside the cell. In *Caenorhabditis elegans*, LIPL-4 is a lysosomal acid lipase specifically expressed in the intestine, the peripheral fat storage tissue. It is upregulated upon fasting[9,10] and in the long-lived mutant that reduces insulin/IGF-1 signalling (IIS) or lacks germline[1,10,11]. In the intestine, the induction of *lipl-4* activates a lysosome-to-nucleus retrograde lipid signalling pathway to regulate transcription and mitochondria, leading to increased lipolysis and lifespan[1,12]. So far, the signalling role of lysosomes in inter-tissue communication remains unknown. In this Article, we reveal that LIPL-4-induced lysosomal lipolysis in the peripheral fat storage tissue upregulates dihomo-γ-linolenic acid (DGLA), which binds to a secreted lipid chaperone protein, LBP-3, and cell-non-autonomously induces the neuropeptide signalling pathway to promote longevity. Downstream of LIPL-4–LBP-3 lipid signalling, the nuclear receptor NHR-49, a *C. elegans* homologue of PPARα, specifically acts in neurons and mediates the transcriptional induction of neuropeptide genes and pro-longevity effect. These studies reveal that lysosome-derived signals are crucial not only for organellar crosstalk in the cell[12], but also for tissue coordination in the organism, making them exciting targets for optimal pro-longevity intervention at the systemic level.

## Results

**Peripheral lysosomal lipolysis turns on neuronal signalling.** We first systemically profiled transcriptional changes associated with LIPL-4-induced lysosomal lipolysis using RNA sequencing (RNA-seq) analysis of *lipl-4* transgenic worms (*lipl-4 Tg*), which constitutively express this lipase in the intestine and have extended lifespan (Fig. 1a and Supplementary Table 4). A series of genes are differentially expressed in *lipl-4 Tg* compared with wild-type (WT) worms (fold change >1.5, *P* < 0.05; Supplementary Table 1, and PCA analysis in Extended Data Fig. 1a). DAVID functional annotation of the genes upregulated in *lipl-4 Tg* revealed the enrichment of distinct biological processes (Fig. 1b). Besides 'immune response' and 'defence response' Gene Ontology categories that are commonly associated with longevity, we discovered the enrichment of 'neuropeptide signalling pathway', which consists of genes encoding neuropeptides and their processing enzymes. We used quantitative PCR with reverse transcription (qRT–PCR) to confirm the induction of neuropeptide-processing genes *egl-3*, *egl-21*, *pgal-1/pghm-1* and *sbt-1* (Fig. 1c), which encode neuroendocrine convertase,

[1]Graduate Program in Developmental Biology, Baylor College of Medicine, Houston, TX, USA. [2]Huffington Center on Aging, Baylor College of Medicine, Houston, TX, USA. [3]Medical Scientist Training Program, Baylor College of Medicine, Houston, TX, USA. [4]Department of Molecular and Human Genetics, Baylor College of Medicine, Houston, TX, USA. [5]Integrative Program of Molecular and Biochemical Sciences, Baylor College of Medicine, Houston, TX, USA. [6]Department of Pharmacology and Chemical Biology, Baylor College of Medicine, Houston, TX, USA. [7]Department of Biochemistry, Emory University School of Medicine, Atlanta, GA, USA. [8]Department of Biochemistry and Molecular Biology, University of Texas Health Science Center at Houston, Houston, TX, USA. [9]Department of Molecular Metabolism, Harvard T.H. Chan School of Public Health, Boston, MA, USA. [10]Center of Epigenetics and Disease Prevention, Institute of Bioscience and Technology, Texas A&M University, Houston, TX, USA. [11]Howard Hughes Medical Institute, Baylor College of Medicine, Houston, TX, USA. [12]Present address: Department of Medicine, University of California San Francisco, San Francisco, CA, USA. [13]Present address: State Key Laboratory of Cellular Stress Biology, Innovation Center for Cell Signaling Network, School of Life Sciences, Xiamen University, Xiamen, China. ✉e-mail: wmeng@bcm.edu

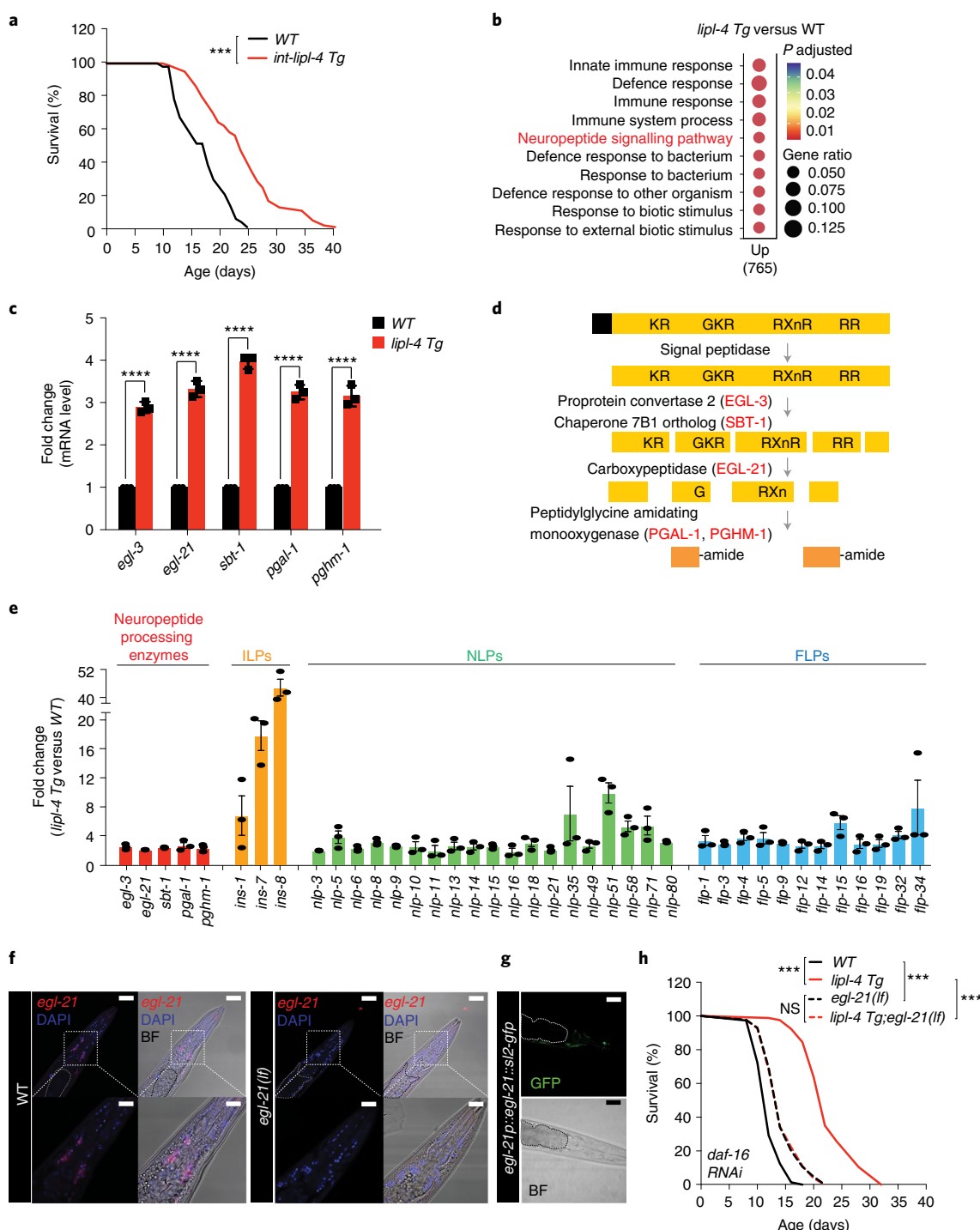

**Fig. 1 | Peripheral lysosomal lipolysis upregulates neuropeptide signalling. a**, Transgenic strains constitutively expressing *lipl-4* in the intestine (*lipl-4 Tg*) show lifespan extension. **b**, Gene Ontology of upregulated genes in *lipl-4 Tg* compared with WT worms. **c**, Genes encoding neuropeptide-processing enzymes *egl-3*, *egl-21*, *sbt-1*, *pgal-1* and *pghm-1* are transcriptionally upregulated by *lipl-4 Tg*. **d**, Schematic diagram of the neuropeptide processing and maturation pathway. **e**, List of significantly upregulated neuropeptide genes (*P* < 0.05) by *lipl-4 Tg* in RNA-seq transcriptome profiling. **f**, *egl-21* mRNA is detected by smFISH (Red Quasar 670), while nuclei are stained by DAPI. *egl-21* is detected in neurons (boxed region) of WT but not *egl-21(lf)* worms. The intestine region marked by dashed lines shows no *egl-21*. Scale bar, 30 μm and 10 μm in the inset. **g**, GFP driven by the *egl-21* endogenous promoter is expressed in neurons but not the intestine (marked by dashed lines). Scale bar, 10 μm. **h**, The loss-of-function mutation of *egl-21(lf)* suppresses *lipl-4 Tg* longevity. *daf-16* RNAi knockdown is used to eliminate the influence from ILP reduction in *egl-21(lf)*. In **a** and **h**, *n* = 3 biologically independent samples; NS, not significant (*P* > 0.05), ***\*P* < 0.001 by log-rank test, 60–120 worms per replicate. See Supplementary Tables 4 and 5 for full lifespan data. In **c**, error bars represent mean ± standard error of the mean (s.e.m.), *n* = 3 biologically independent samples, \*\*\*\**P* < 0.0001 by two-tailed Student's *t*-test, ~2,000 worms per replicate. In **e**, error bars represent mean ± s.e.m., *n* = 3 biologically independent samples, DESeq2 |fold change| ≥1.5; *P* < 0.05 by two-sided Wald test (*lipl-4 Tg* versus WT), ~3,000 worms per replicate. Source numerical data are available in source data.

carboxypeptidase E, peptidylglycine α-amidating monooxygenase and neuroendocrine chaperone 7B2, respectively (Fig. 1d). Among them, *egl-21*, *pgal-1* and *pghm-1* are specifically expressed in neurons[13,14]. Their inductions suggest a cell-non-autonomous regulation of neuronal genes by peripheral lysosomal lipolysis. There are also neuropeptides transcriptionally upregulated in *lipl-4 Tg*, including 3 insulin-like peptides (ILPs), 12 FMRFamide-related peptides (FLPs) and 19 neuropeptide-like proteins (NLPs) (Fig. 1e and Supplementary Table 1).

Next, we examined the role of the neuropeptide signalling pathway in longevity regulation using the loss-of-function mutant of *egl-21*, *egl-21(lf)*. The EGL-21 enzyme is required for neuropeptide processing by removing basic residues from the C-terminus of cleaved peptides[15], and the *egl-21* gene is exclusively expressed in neurons as visualized by both single-molecule fluorescence in situ hybridization (smFISH) and its green fluorescent protein (GFP) transgenic reporter (Fig. 1f,g). We found that *lipl-4 Tg* cannot prolong the lifespan of *egl-21(lf)* (Extended Data Fig. 1b and Supplementary Table 4), suggesting that the induction of neuropeptide signalling contributes to *lipl-4*-induced longevity. Previous genomic RNA interference (RNAi) screens found that inactivation of *egl-3*, encoding the convertase upstream of EGL-21, extends lifespan, which is suppressed by inactivation of the *daf-16*/FOXO transcription factor[16]. Similarly, we found that *egl-21(lf)* has extended lifespan and this lifespan extension requires *daf-16* (Extended Data Fig. 1c and Supplementary Table 5). Given that ILPs regulate lifespan[17,18] and DAF-16/FOXO is the key mediator of longevity caused by IIS reduction[19], the requirement of DAF-16 suggests that the longevity effect conferred by *egl-21(lf)* is possibly due to reduced agonist ILP maturation and IIS. In contrast, the lifespan extension in *lipl-4 Tg* is not suppressed by *daf-16* RNAi (Extended Data Fig. 1d and Supplementary Table 5), indicating a negligible role of ILPs for *lipl-4*-induced longevity. These results suggest that, in order to test whether the reduction of NLPs or FLPs in *egl-21(lf)* affects *lipl-4*-induced longevity, we should inactivate *daf-16* to eliminate the contribution from ILP reduction. We found that, with *daf-16* RNAi, *egl-21(lf)* fully abrogates the lifespan extension conferred by *lipl-4 Tg* (Fig. 1h and Supplementary Table 5). Together, these results suggest that neuropeptide processing is required for intestinal lysosomal lipolysis to promote longevity, which is probably associated with NLPs and/or FLPs but not ILPs.

## Neuronal NLP-11 neuropeptide promotes longevity

To identify specifically involved neuropeptides, we performed an RNAi-based screen to search for neuropeptide-encoding genes whose inactivation suppresses the lifespan extension in *lipl-4 Tg* (Supplementary Table 3). We discovered that RNAi inactivation of *nlp-11* in a neuronal RNAi-sensitive background suppresses *lipl-4 Tg* longevity without affecting WT lifespan (Fig. 2a and Supplementary Tables 3 and 5). We further generated a clustered regularly interspaced short palindromic repeats (CRISPR) deletion mutant for *nlp-11*, *nlp-11(lf)* (Extended Data Fig. 2a) and crossed it with *lipl-4 Tg*. We found that *nlp-11(lf)* reduces the lifespan extension in *lipl-4 Tg* but not WT lifespan (Fig. 2b and Supplementary Table 4). *nlp-11* is transcriptionally upregulated in *lipl-4 Tg* (Extended Data Fig. 2b), and overexpression of *nlp-11* driven by its endogenous promoter sufficiently prolongs lifespan (Fig. 2c, Extended Data Fig. 2c and Supplementary Table 4). *nlp-11* expresses in both neurons and the intestine (Extended Data Fig. 2d). To examine where *nlp-11* functions to regulate longevity, we knocked down *nlp-11* selectively in the intestine and found that this intestine-only inactivation does not affect the lifespan extension in *lipl-4 Tg* (Fig. 2d and Supplementary Table 5). We also overexpressed *nlp-11* in either neurons or the intestine using tissue-specific promoters and found that only neuron-specific overexpression of *nlp-11* is sufficient to prolong lifespan (Fig. 2e,f, Extended Data Fig. 2e,f and Supplementary Table 4).

Together, these results demonstrate that neuronal *nlp-11* is specifically responsible for the longevity effect conferred by intestinal lysosomal lipolysis.

## Lysosome-derived PUFAs regulate neuropeptide and longevity

Lysosomal acid lipase catalyses FFA release from TAGs and/or CEs[8]. Through lipidomic profiling of FFAs, we found that the levels of polyunsaturated fatty acids (PUFAs) are increased in *lipl-4 Tg* (Fig. 3a). To test whether these PUFAs are derived from lysosomal lipolysis, we purified lysosomes and profiled different classes of lipids. We found that, compared with WT, the level of TAGs is reduced by approximately threefold in lysosomes purified from *lipl-4 Tg* (Extended Data Fig. 3a). Moreover, 186 out of 305 detected TAG species (61%) are decreased in *lipl-4 Tg* lysosomes, with 63% of them containing PUFAs (Fig. 3b). These results suggest that the induction of PUFAs is probably due to increased lysosomal lipolysis of TAGs.

To test the hypothesis that these PUFAs serve as cell-non-autonomous signals to regulate neuropeptides, we utilized loss-of-function mutants of *fat-1* and *fat-3* that encode ω-3 fatty acid desaturases and Δ6-desaturase, respectively required for PUFA biosynthesis[20] (Fig. 3c). With these desaturase mutants, the upregulation of neuropeptide genes (Fig. 3d) and the lifespan extension are suppressed in *lipl-4 Tg* (Extended Data Fig. 3b,c and Supplementary Table 4). FAT-1 and FAT-3 function in the intestine and neurons to catalyse PUFA biosynthesis locally[21,22]. We selectively reduced intestinal PUFAs by knocking down *fat-1* and *fat-3* only in the intestine and found that intestine-only inactivation of either *fat-1* or *fat-3* fully abrogates the lifespan extension in *lipl-4 Tg* (Fig. 3e,f and Supplementary Table 5). Together, these results suggest that PUFAs derived from intestinal lysosomal lipolysis mediate both neuropeptide induction and longevity.

**Peripheral lipid chaperone LBP-3 promotes longevity.** FFAs have low aqueous solubility and must be bound to proteins in order to diffuse through the lipophobic environment. A family of proteins termed fatty acid binding proteins (FABPs) function as lipid chaperones, which reversibly bind FFAs and their derivatives to mediate their trafficking and signalling effects[23,24]. We tested whether specific FABPs facilitate the action of intestinal PUFAs on neurons, and focused on three FABPs, LBP-1, LBP-2 and LBP-3, that carry putative secretory signals. We found that RNAi inactivation of *lbp-2* or *lbp-3* but not *lbp-1* specifically suppresses the induction of neuropeptide genes caused by *lipl-4 Tg* (Extended Data Fig. 4a,b). To confirm the RNAi knockdown results, we generated CRISPR deletion mutants of *lbp-2* and *lbp-3* (Extended Data Fig. 4c) and crossed them with *lipl-4 Tg*. We found that only *lbp-3* but not *lbp-2* deletion suppresses the induction of neuropeptide genes (Fig. 4a). Deletion of *lbp-3* also suppresses *lipl-4 Tg* longevity without affecting WT lifespan (Fig. 4b and Supplementary Table 4).

We also found that transgenic strains that constitutively express *lbp-3* (*lbp-3 Tg*) live longer than WT worms (Fig. 4c, Extended Data Fig. 4d and Supplementary Table 4) and show upregulation of *egl-3*, *egl-21* and *nlp-11* (Fig. 4d). We further profiled transcriptome changes in *lbp-3 Tg* using RNA-seq (Supplementary Table 2, and PCA analysis in Extended Data Fig. 4e). Among 39 neuropeptide genes upregulated in *lipl-4 Tg*, all 5 neuropeptide-processing genes and 16 neuropeptide genes were upregulated in *lbp-3 Tg* (Fig. 4e and Supplementary Table 2). Similar to *lipl-4 Tg*, the lifespan extension conferred by *lbp-3 Tg* is not suppressed by *daf-16* RNAi (Extended Data Fig. 4f and Supplementary Table 5), but it is suppressed by *egl-21(lf)* in the *daf-16* RNAi background (Fig. 4f and Supplementary Table 5). Moreover, *nlp-11* inactivation partially suppresses the lifespan extension in *lbp-3 Tg* (Fig. 4g and Supplementary Table 4). These results support that specific neuropeptides act downstream of

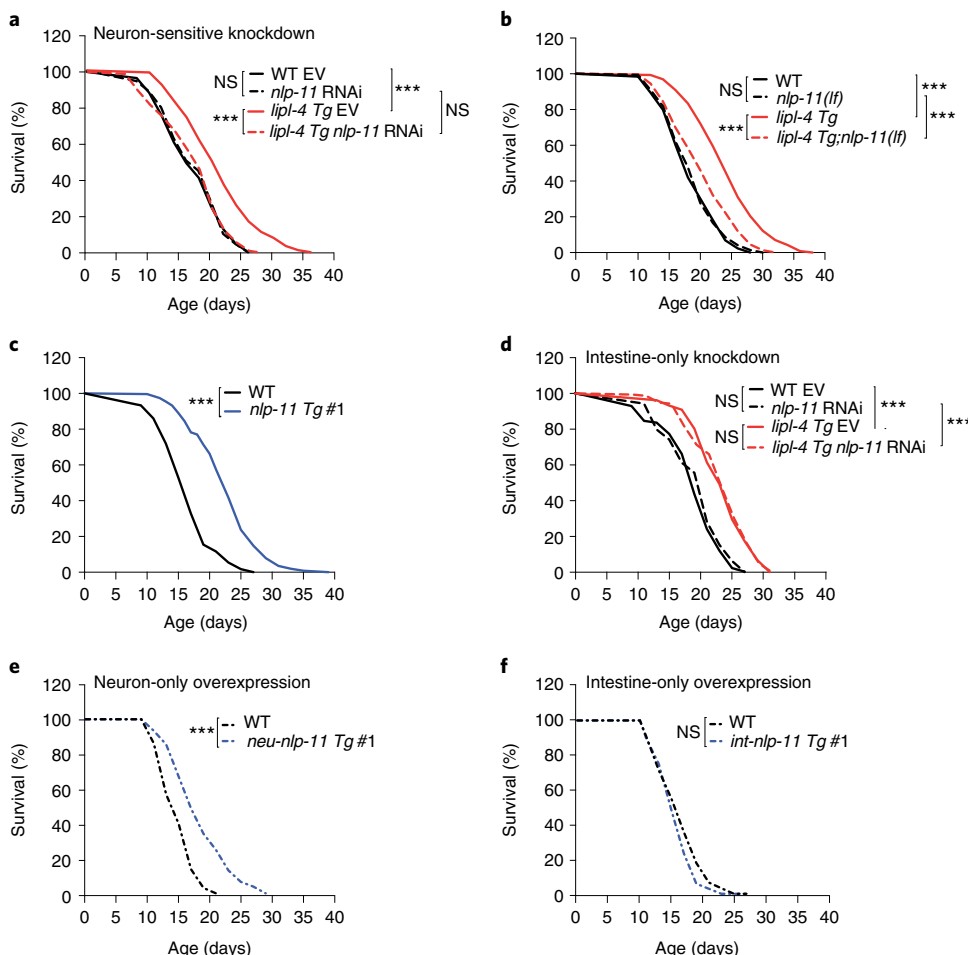

**Fig. 2 | NLP-11 neuropeptide acts in neurons to promote longevity. a**, Knockdown of *nlp-11* in a neuronal RNAi-sensitive background suppresses *lipl-4 Tg* longevity. **b**, The loss-of-function mutation of *nlp-11(lf)* suppresses *lipl-4 Tg* longevity. **c**, Constitutive expression of *nlp-11* driven by its endogenous promoter extends lifespan. **d**, RNAi knockdown of *nlp-11* selectively in the intestine shows no suppression of *lipl-4 Tg* longevity. **e,f**, Neuron-specific overexpression of *nlp-11* prolongs lifespan (**e**), but intestine-specific overexpression has no such effect (**f**). In **a–f**, *n* = 3 biologically independent samples, \*\*\*P < 0.001 by log-rank test, 60–120 worms per replicate, EV, empty vector control for RNAi. Lifespan of one transgenic strain (#1) is shown, and the others are in Extended Data Fig. 2. See Supplementary Tables 4 and 5 for full lifespan data.

LIPL-4-LBP-3 signalling to regulate longevity. Next, we found that transgenic strains that selectively overexpress *lbp-3* in the intestine exhibit the upregulation of neuropeptide genes (Fig. 4h) and lifespan extension (Fig. 4i, Extended Data Fig. 4g and Supplementary Table 4). Together, these results support that the specific lipid chaperone LBP-3 mediates fat-to-neuron communication to regulate neuropeptides and longevity.

**LBP-3 relies on secretion for its regulation.** To further understand the function of LBP-3 in this endocrine regulation, we examined whether LBP-3 can be secreted from the intestine. In *C. elegans*, coelomocytes are scavenger cells that take up secreted materials from the body cavity and serve as a monitor of secreted proteins[25]. We generated a transgenic strain expressing an intestine-specific polycistronic transcript encoding both LBP-3–red fluorescent protein (RFP) fusion and GFP, such that GFP indicates cells expressing *lbp-3* and RFP directly labels LBP-3 protein. Without tagging with any proteins, GFP was detected ubiquitously within intestinal cells (Fig. 5a). LBP-3–RFP fusion, on the other hand, was detected within intestinal cells and also in coelomocytes (Fig. 5a), which shows LBP-3 secretion from the intestine into the body cavity. We also discovered that this secretion of LBP-3–RFP is elevated in *lipl-4 Tg* (Fig. 5b,c). Within intestinal cells, LBP-3 protein is detected in the

cytosol and also at lysosomes that are marked by LMP-1 and stained with LysoTracker (Extended Data Fig. 5a). Moreover, we generated transgenic strains that overexpress LBP-3 without its secretory signal only in the intestine and found no neuropeptide gene induction (Fig. 5d) or lifespan extension (Fig. 5e, Extended Data Fig. 5b and Supplementary Table 4) in these strains. RFP fusion of this non-secretable LBP-3 was not detected in coelomocytes (Fig. 5f). Thus, LBP-3 protein requires secretion from the intestine to systemically regulate neuropeptides and longevity, which is triggered by LIPL-4-induced lysosomal lipolysis.

**DGLA regulates LBP-3 secretion, neuropeptides and longevity.** To examine how specific PUFAs and LBP-3 coordinate with each other, we first examined the effect of PUFAs on LBP-3 secretion. We used *fat-3* RNAi to reduce PUFA biosynthesis in peripheral tissues and found reduction of LBP-3–RFP secretion in *lipl-4 Tg* (Fig. 6a,b). Thus, PUFA induction by *lipl-4 Tg* promotes LBP-3 secretion from the intestine. It is known that the *fat-3(lf)* mutant lacks 20-carbon PUFAs, including ω-6 DGLA and arachidonic acid (AA) that are induced by *lipl-4 Tg* and ω-3 eicosapentaenoic acid (EPA) and eicosatetraenoic acid (ETA) with no induction (Fig. 3a). We then tested whether LBP-3 binds to DGLA, AA, EPA and ETA using a competitive fluorescence-based binding assay. In this assay, when bound to

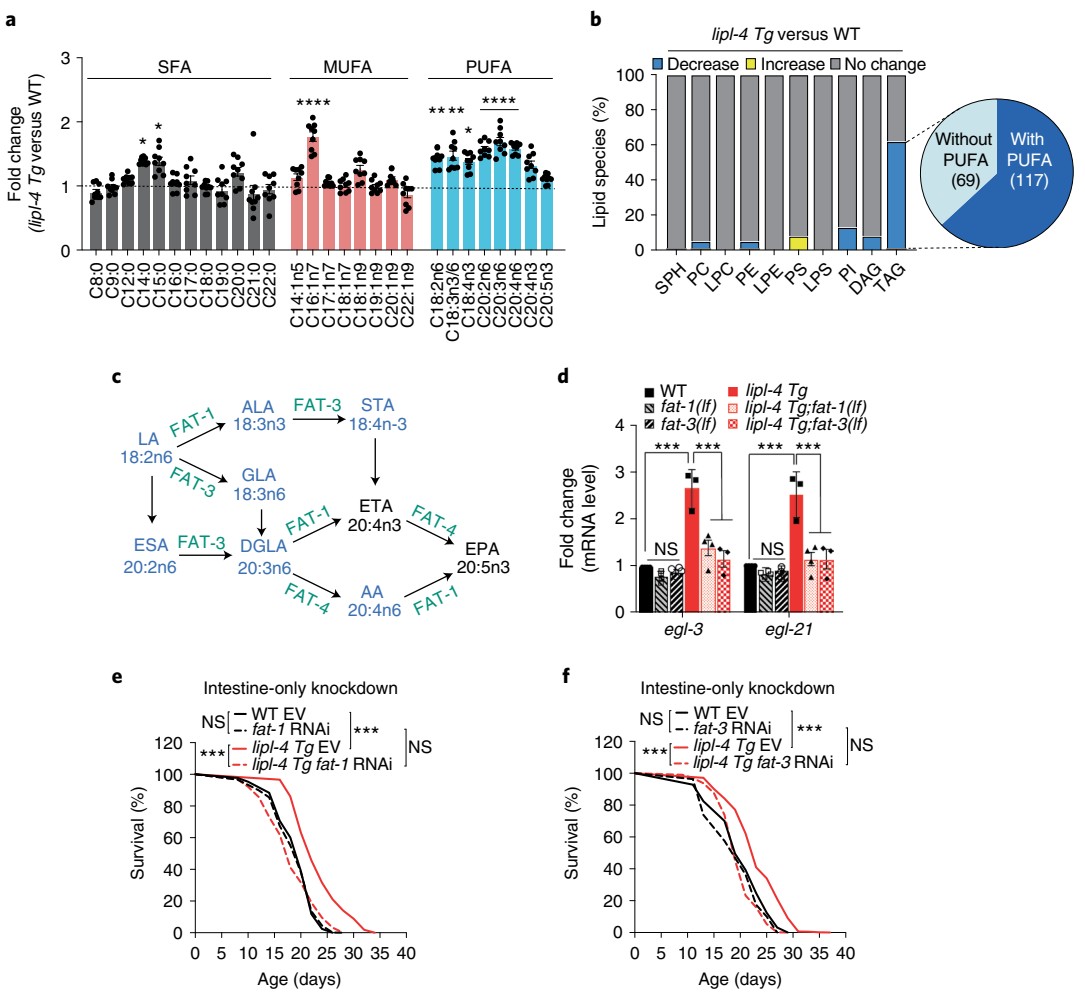

**Fig. 3 | Lysosome-derived PUFAs in the periphery regulate neuropeptide and longevity. a**, Relative levels of FFAs, saturated fatty acids (SFAs), MUFAs and PUFAs are quantified by liquid chromatography–mass spectrometry in *lipl-4 Tg* versus WT. **b**, Percentage of lipid species showing significant changes ($P < 0.05$) in the lysosome purified from *lipl-4 Tg* versus WT worms. Out of 186 TAG species decreased in *lipl-4 Tg*, 117 contain PUFAs. Sphingophospholipids (SPH), phosphatidylcholines (PC), lysophosphatidylcholines (LPC), phosphatidylethanolamines (PE), lysophosphatidylethanolamines (LPE), phosphatidylserines (PS), lysophosphatidylserines (LPS), phosphatidylinositols (PI), diacylglycerols (DAG) and TAG. **c**, Schematic diagram of PUFA biosynthesis in *C. elegans*. PUFAs enriched in *lipl-4 Tg* are highlighted in blue, while desaturase enzymes are marked in green. **d**, The loss-of-function mutation of the fatty acid desaturase *fat-1(lf)* or *fat-3(lf)* suppresses the transcriptional upregulation of *egl-3* and *egl-21* neuropeptide genes by *lipl-4 Tg*. **e,f**, RNAi knockdown of either *fat-1* (**e**) or *fat-3* (**f**) selectively in the intestine suppresses *lipl-4 Tg* longevity. In **a**, error bars represent mean ± s.e.m., $n = 6$ (WT) and $n = 9$ (*lipl-4 Tg*) biologically independent samples, *$P = 0.028$ for C18:4n3, $P = 0.023$ for C14:0 and $P = 0.026$ for C15:0, **$P = 0.007$ for C18:2n6 and $P = 0.002$ for C18:3n3/6, ****$P < 0.0001$ for C16:1n7, C20:2n6, C20:3n6 and C20:4n6 by two-way ANOVA with Holm–Sidak correction (*lipl-4 Tg* versus WT), ~40,000 worms per replicate. In **d**, error bars represent mean ± s.e.m., $n = 4$ (WT, *fat-1(lf)*, *fat-3(lf)* and *lipl-4 Tg;fat-1(lf)*) and $n = 3$ (*lipl-4 Tg* and *lipl-4 Tg;fat-3(lf)*) biologically independent samples, ***$P < 0.001$ by two-way ANOVA with Holm–Sidak correction, ~3,000 worms per replicate. In **e** and **f**, $n = 3$ biologically independent samples, ***$P < 0.001$ by log-rank test, 60–120 worms per replicate. See Supplementary Table 5 for full lifespan data. Source numerical data are available in source data.

LBP-3, amphipathic 1-anilinonaphthalene-8-sulfonic acid (1,8-ANS) shows enhanced fluorescence that is quenched once outcompeted by FFAs. We found that DGLA ($K_d = 10.96\ \mu M$), AA ($K_d = 2.9\ \mu M$) and EPA ($K_d = 4.76\ \mu M$) but not ETA bind to LBP-3 (Fig. 6c).

Next, we supplemented DGLA, AA or EPA to worms and measured neuropeptide gene expression. We found that, in *fat-3(lf)*, DGLA supplementation is able to restore the upregulation of neuropeptide genes caused by *lipl-4 Tg* (Fig. 6d). Neither AA nor EPA supplementation shows such an ability (Fig. 6d). To examine tissue specificity of this restoration, we conducted qRT–PCR analysis using dissected intestine and found that DGLA supplementation causes no intestinal induction of *egl-3* or *egl-21* (Extended Data Fig. 6a). We also imaged *egl-21* messenger RNA transcripts using smFISH and found an increase in neurons by DGLA supplementation

(Extended Data Fig. 6b). Moreover, DGLA supplementation sufficiently restores the increased LBP-3–RFP secretion (Fig. 6e,f) and lifespan extension in *lipl-4 Tg* with *fat-3* inactivation (Fig. 6g and Supplementary Table 5). Together, these results suggest that the induction of DGLA by lysosomal lipolysis promotes secretion of the LBP-3 lipid chaperone from the intestine, and LBP-3–DGLA signals to neurons to regulate neuropeptides and longevity.

**DGLA binding specificity of LBP-3 mediates its effects.** To further confirm that the effect of DGLA on neurons is dependent on LBP-3, we supplemented DGLA to *lipl-4 Tg* with both *fat-3(lf)* and *lbp-3(lf)* mutants. We found that, in the absence of LBP-3, DGLA supplementation fails to restore the neuropeptide gene induction (Fig. 7a). Thus, DGLA requires LBP-3 to regulate neuropeptides. Next, we

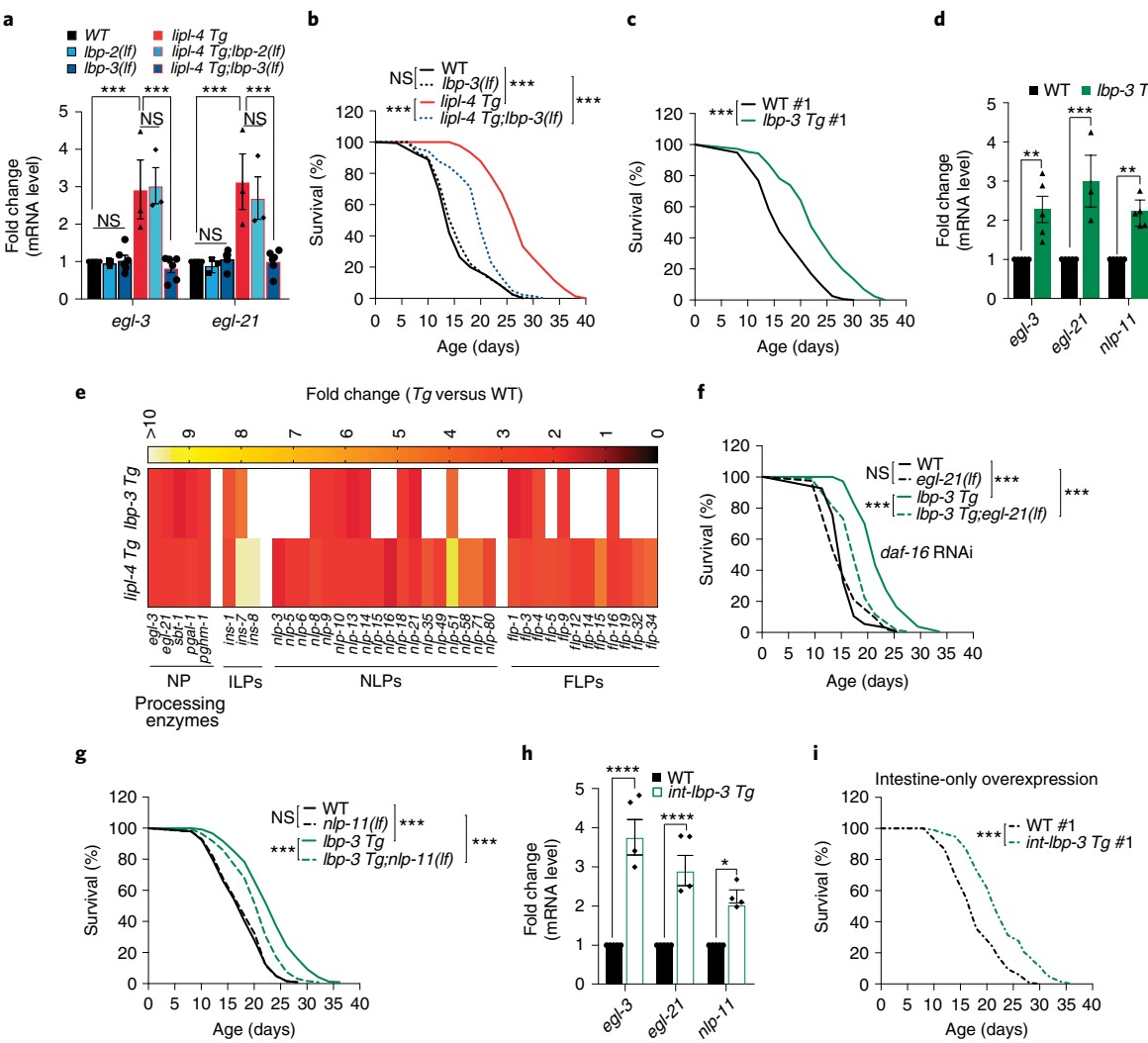

**Fig. 4 | Peripheral lipid chaperone LBP-3 regulates neuropeptide and longevity. a**, The loss-of-function mutation of the lipid chaperone *lbp-3(lf)* but not *lbp-2 (lf)* suppresses the transcriptional up-regulation of *egl-3* and *egl-21* by *lipl-4 Tg*. **b**, *lbp-3(lf)* suppresses *lipl-4 Tg* longevity. **c**, Constitutive expression of *lbp-3* driven by its own endogenous promoter (*lbp-3 Tg*) prolongs lifespan. **d**, The transcriptional levels of *egl-3*, *egl-21* and *nlp-11* are induced in the *lbp-3 Tg* worms. **e**, Out of 39 upregulated neuropeptide genes by *lipl-4 Tg*, 22 are also significantly induced by *lbp-3 Tg* (*P* < 0.05). **f**, The loss-of-function mutation of *egl-21(lf)* suppresses *lbp-3 Tg* longevity. *daf-16* RNAi knockdown is used to eliminate the influence from ILP reduction in *egl-21(lf)*. **g**, The loss-of-function mutation of *nlp-11(lf)* suppresses *lbp-3 Tg* longevity. **h**, Intestine-specific *lbp-3* overexpression upregulates the transcriptional levels of *egl-3*, *egl-21* and *nlp-11*. **i**, Overexpression of *lbp-3* selectively in the intestine extends lifespan. In **a**, error bars represent mean ± s.e.m., *n* = 3 (*lbp-2(lf)*, *lipl-4 Tg* and *lipl-4 Tg;lbp-2(lf)*) and *n* = 6 (WT, *lbp-3(lf)* and *lipl-4 Tg;lbp-3(lf)*) biologically independent samples, ***P* = 0.0001 by two-way ANOVA with Holm–Sidak correction, ~2,000 worms per replicate. In **b**, **c**, **f**, **g** and **i**, *n* = 3 biologically independent samples, ***P* < 0.001 by log-rank test, 60–120 worms per replicate. Lifespan of one transgenic strain (#1) is shown and the others are in Extended Data Fig. 4. See Supplementary Tables 4 and 5 for full lifespan data. In **d**, error bars represent mean ± s.e.m., *n* = 5 (WT and *lbp-3 Tg* for *egl-3* and *nlp-11*) and *n* = 3 (WT and *lbp-3 Tg* for *egl-21*) biologically independent samples, ***P* = 0.007 for *egl-3*, ***P* = 0.007 for *nlp-11* and ****P* = 0.0002 for *egl-21* by two-way ANOVA with Holm–Sidak correction, ~2,000 worms per replicate. In **h**, error bars represent mean ± s.e.m., *n* = 4 biologically independent samples, **P* = 0.0497, *****P* < 0.0001 by two-way ANOVA with Holm–Sidak correction, ~2,000 worms per replicate. Source numerical data are available in source data.

tested whether the lipid binding specificity of LBP-3 is responsible for its regulatory effects. Despite the close homology between LBP-2 and LBP-3, LBP-2 is not required for the upregulation of neuropeptide genes in *lipl-4 Tg* (Fig. 4a), and its overexpression does not induce neuropeptide genes (Fig. 7b). We then analysed their lipid binding preferences through profiling *C. elegans* liposome that binds to either LBP-2 or LBP-3. We found that LBP-2 and LBP-3 exhibit distinct lipid binding preferences (Fig. 7c and Extended Data Fig. 7a). In particular, DGLA shows 25% occupancy among LBP-3-bound PUFAs, but only 1% occupancy among LBP-2-bound PUFAs (Fig. 7c). Interestingly, for EPA, the highest-abundant

PUFAs in *C. elegans* liposome (Fig. 7d), both LBP-2 and LBP-3 show low percentage of occupancy (6% and 3%, respectively) (Fig. 7c). These results suggest that LBP-2 and LBP-3 have different binding specificity towards PUFAs.

Next, we compared the predicted structures of LBP-2 and LBP-3 using AlphaFold2 and found changes in two cap-like α-helixes that are responsible for lipid binding (Fig. 7e). A sequence alignment between LBP-2 and LBP-3 reveals that ten amino acids are different in these regions (Fig. 7f). We thus designed a chimeric protein by replacing the two cap-like LBP-3 α-helixes from N38 to K60 with those present in LBP-2 (Fig. 7f and Extended Data Fig. 7b). We generated

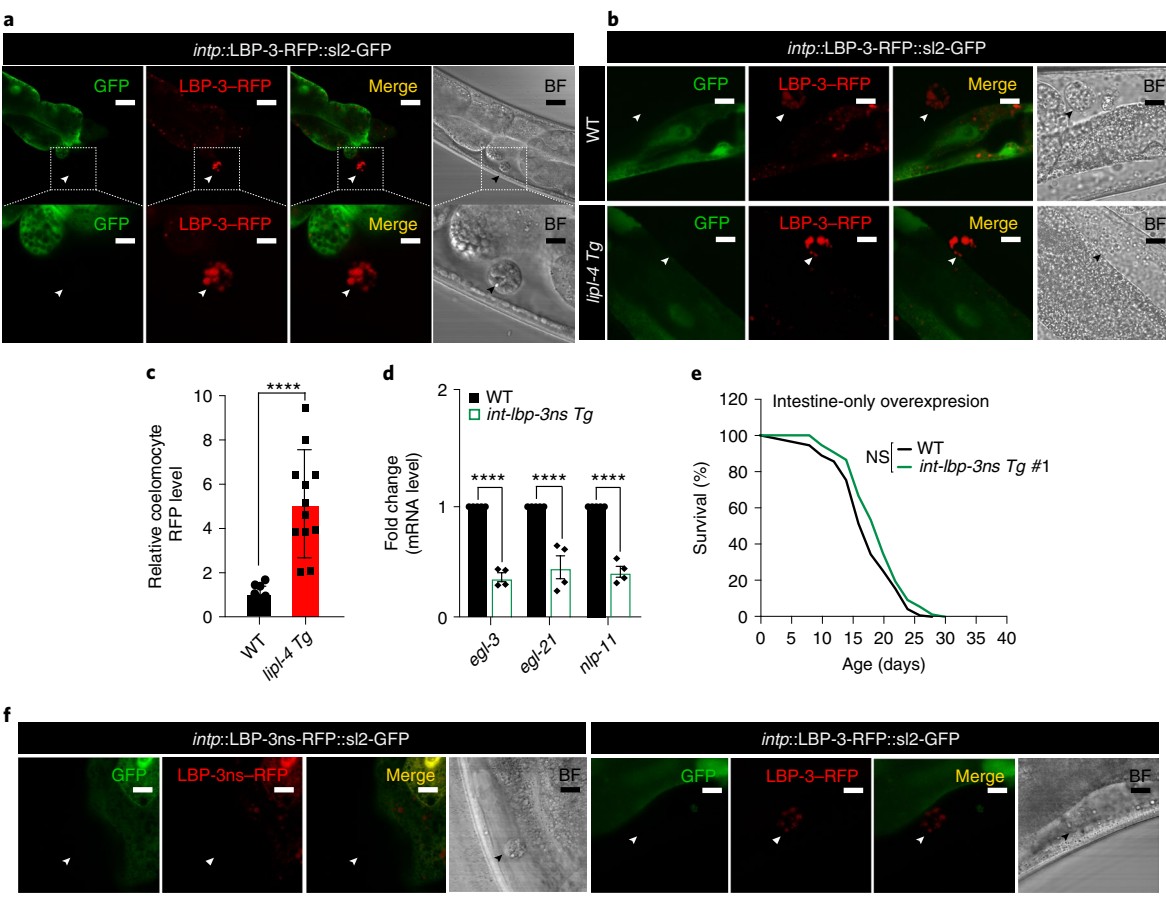

**Fig. 5 | LBP-3 secreted from the periphery regulates neuropeptide and longevity. a**, The gene expression of *lbp-3* is indicated by polycistronic GFP, while the LBP-3 protein is visualized by its RFP fusion. Secreted LBP-3–RFP fusion is detected in coelomocytes marked with arrowheads. Scale bars, 30 μm and 10 μm in the inset. **b,c**, The level of secreted LBP-3–RFP fusion is increased by *lipl-4 Tg*. Secreted LBP-3–RFP fusion proteins in coelomocytes are marked by arrowheads (**b**), and their levels are quantified in *lipl-4 Tg* versus WT (**c**). Scale bar, 10 μm. Representative images from three biological repeats. **d**, Intestine-specific overexpression of *lbp-3ns*, which does not carry the secretory signal sequence, decreases the expression of the neuropeptide genes. **e**, Intestine-specific overexpression of *lbp-3ns* fails to extend lifespan. **f**, RFP fusion of the LBP-3ns protein is not detected in coelomocytes, marked by arrowheads. Scale bar, 10 μm. In **c**, error bars represent mean ± s.e.m., $n = 11$ (WT) and $n = 15$ (*lipl-4 Tg*) biologically independent samples, ****$P < 0.0001$ by two-tailed Student's *t*-test. In **d**, error bars represent mean ± s.e.m., $n = 4$ biologically independent samples, ****$P < 0.0001$ by two-way ANOVA with Holm–Sidak correction ~2,000 worms per replicate. In **e**, $n = 3$ biologically independent samples; NS, $P > 0.05$ by log-rank test, 75–100 worms per replicate. Lifespan of one transgenic strain in shown, and the others are in Extended Data Fig. 5. See Supplementary Table 4 for full lifespan data. Source numerical data are available in source data.

transgenic lines expressing this chimeric protein selectively in the intestine and confirmed that the chimeric protein expresses normally (Extended Data Fig. 7c). We found that overexpression of the chimeric protein, like LBP-2, does not induce neuropeptide gene expression (Fig. 7g and Extended Data Fig. 7d). No lifespan extension was detected in these transgenic strains either (Fig. 7h, Extended Data Fig. 7e and Supplementary Table 4). These results suggest that the lipid binding specificity of LBP-3 towards DGLA is necessary for its regulation of neuropeptides and longevity.

**Neuronal transduction of peripheral lipid signals.** To examine whether secreted LBP-3 is taken up by neurons, we generated a transgenic strain that specifically expresses GFP nanobody (GBP) in neurons together with polycistronic mKate, and then crossed it with a transgenic strain expressing GFP-fused LBP-3 only in the intestine. In this line, if secreted LBP-3–GFP proteins from the intestine are taken up by neurons, neuronal GBP will capture them, making GFP visible in neurons (Fig. 8a). In supporting LBP-3–GFP uptake by neurons, we detected GFP signals in mKate-positive neurons (Fig. 8b). As controls, we did not detect neuronal GFP

signals in either the GBP or the LBP-3–GFP transgenic strain alone (Extended Data Fig. 8a,b). Moreover, we generated a transgenic strain that expresses GBP fused with the extracellular domain of SAX-7 (GBP-SAX-7)[26] in neurons, and then crossed it with the intestine-specific LBP-3–GFP transgenic strain. In this line, LBP-3–GFP signals were also detected in neurons (Extended Data Fig. 8c), supporting a close proximity between secreted LBP-3 and the neuronal surface. Together, these results reveal LBP-3 as an endocrine lipid chaperone, which is transported from the intestine to neurons to mediate *lipl-4*-induced longevity.

Previously, we discovered LBP-8 as a cell-autonomous mediator of *lipl-4*-induced longevity. To investigate the interaction between LBP-3 and LBP-8, we examined the transcriptional levels of the neuropeptide genes in the long-lived *lbp-8* transgenic strain (*lbp-8 Tg*) and found only negligible changes (<35%) (Fig. 8c). Both *lbp-3* and *lbp-8* are partially required for the lifespan extension caused by *lipl-4 Tg* (Fig. 4b)[1], and *lbp-3 Tg* and *lbp-8 Tg* have an additive effect in prolonging lifespan (Fig. 8d). In the intestine, LBP-8 facilitates the lysosome-to-nucleus retrograde transport of lipid signals that activate nuclear receptors NHR-49 and NHR-80 to promote longevity[12].

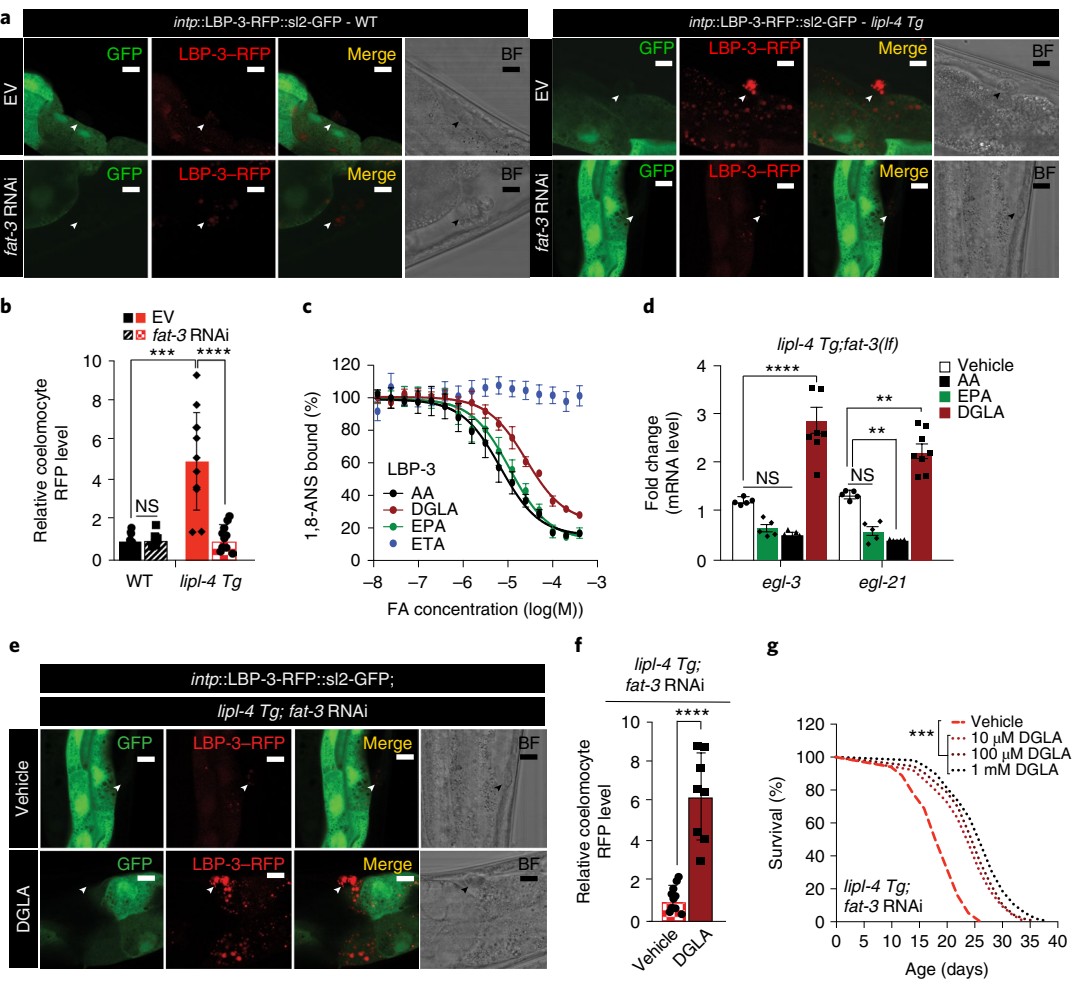

**Fig. 6 | DGLA regulates LBP-3 secretion, neuropeptide and longevity. a,b,** In WT conditions, LBP-3–RFP secretion is not affected by *fat-3* RNAi knockdown. However, the increased secretion of LBP-3–RFP fusion by *lipl-4 Tg* is suppressed by *fat-3* RNAi knockdown. Secreted LBP-3–RFP fusion proteins in coelomocytes are marked by arrowheads (**a**), and their levels are quantified (**b**). Scale bar, 10 μm. Representative images from three biological repeats (**a**). **c**, Fluorescence signals derived from 1,8-ANS bound to LBP-3 are decreased, when 1,8-ANS is outcompeted by the increasing amount (M, molar) of AA, DGLA and EPA, but not ETA. **d**, The supplementation of DGLA but not AA or EPA restores the transcriptional induction of *egl-3* and *egl-21* by *lipl-4 Tg* in the *fat-3(lf)* mutant. DMSO serves as the vehicle control. **e,f**, DGLA supplementation restores LBP-3–RFP secretion in *lipl-4 Tg* with *fat-3* RNAi knockdown. Secreted LBP-3–RFP proteins in coelomocytes are marked by arrowheads (**e**), and their levels are quantified (**f**). Scale bar, 10 μm. Representative images from three biological repeats (**e**). **g**, DGLA supplementation restores the lifespan extension in *lipl-4 Tg* with *fat-3* RNAi knockdown. In **b**, error bars represent mean ± s.e.m., n = 10 biologically independent samples, \*\*\*P < 0.001 and \*\*\*\*P < 0.0001 by one-way ANOVA with Holm–Sidak correction. In **c**, n = 16 biological replicates, one-site Fit Ki, $R^2 = 0.9945$ for AA, $R^2 = 0.9954$ for DGLA, $R^2 = 0.9903$ for EPA and $R^2 = 0.04758$ for ETA. In **d**, error bars represent mean ± s.e.m., n = 5 (*lipl-4 Tg;fat-3(lf)* on EV, AA and EPA) and n = 8 (*lipl-4 Tg;fat-3(lf)* on DGLA) biologically independent samples, \*\*P = 0.009 for AA and P = 0.008 for DGLA, \*\*\*\*P < 0.0001 by two-way ANOVA with Holm–Sidak correction, ~2,000 worms per replicate. In **f**, error bars represent mean ± s.e.m., n = 12 biologically independent samples, \*\*\*\*P < 0.0001 by one-way ANOVA with Holm–Sidak correction. In **g**, n = 3 biologically independent samples, P < 0.001 by log-rank test, 80–100 worms per replicate. See Supplementary Table 5 for full lifespan data. Source numerical data are available in source data.

When examining the involvement of NHR-49 and/or NHR-80, we found that the loss-of-function mutation of *nhr-49* fully suppresses the upregulation of neuropeptide genes in *lipl-4 Tg* and *lbp-3 Tg* (Fig. 8e,f), but the *nhr-80* mutation has a negligible effect (<18% reduction; Fig. 8e). Importantly, in the *nhr-49* mutant background, neuron-specific restoration of *nhr-49* fully rescues the upregulation of neuropeptide genes (Fig. 8f) and the lifespan extension (Fig. 8g and Supplementary Table 4) conferred by *lbp-3 Tg*. Furthermore, we confirmed the neuronal expression of *nhr-49* using a transgenic strain expressing NHR-49-mKate2 fusion driven by its endogenous promoter (Fig. 8h). After crossing this line with the GFP reporter line of *nlp-11*, we found many overlaps between neurons with *nlp-11* expression and NHR-49 localization (Fig. 8h). On the basis of

CenGenApp analysis[27], there are 57 overlapping neurons (Extended Data Fig. 8d and Supplementary Table 6). Thus, lysosomal lipid signals from the periphery act through neuronal NHR-49 to regulate neuropeptides and longevity (Fig. 8i).

## Discussion

This study supports an emerging paradigm that lysosomes are the critical signalling hub for longevity regulation. We have identified two lipid chaperones mediating the signalling role of lysosomes, LBP-8 for lysosome-to-nucleus retrograde signalling and LBP-3 for fat-to-neuron endocrine signalling, which act in parallel to regulate longevity. Overexpression of non-secretable LBP-3 causes decreased transcription of neuropeptide genes, which is probably

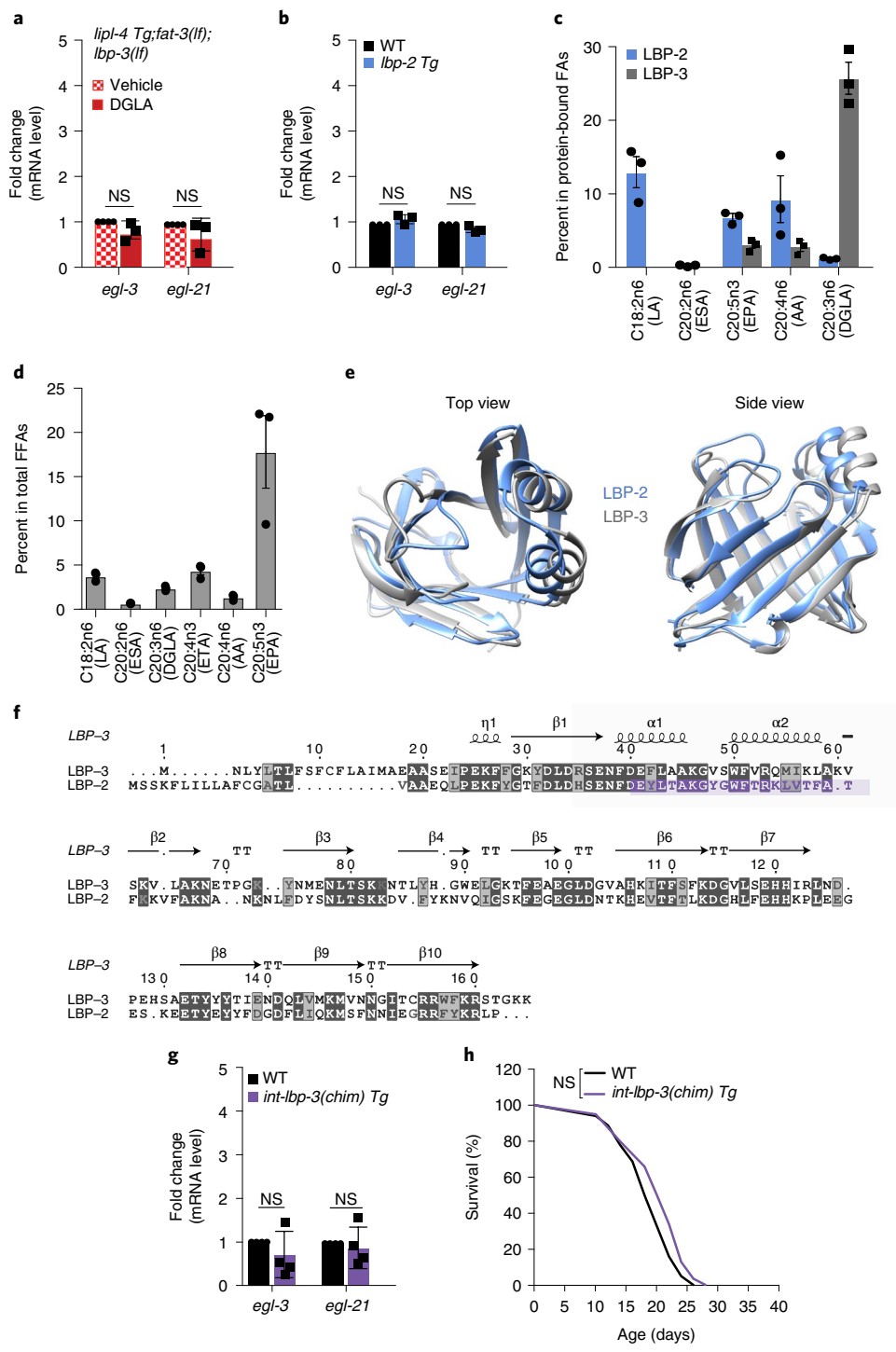

**Fig. 7 | DGLA binding specificity of LBP-3 mediates its effects. a**, DGLA supplementation fails to restore the induction of *egl-3* and *egl-21* by *lipl-4 Tg* in the *fat-3(lf)* and *lbp-3(lf)* double-mutant background. DMSO serves as the vehicle control. **b**, Constitutive expression of *lbp-2* (*lbp-2 Tg*) does not affect the transcription of *egl-3* or *egl-21*. **c**, After incubation with *C. elegans* liposome, fatty acids bound to LBP-2 or LBP-3 proteins were analysed with mass spectrometry. DGLA shows a high percentage of occupancy in LBP-3 but not LBP-2. **d**, In the *C. elegans* liposome, the percentage of EPA is more than ten times higher than that of DGLA or AA. **e**, LBP-2 (blue) and LBP-3 (grey) superposition structures predicted using AlphaFold2. **f**, An LBP-2 and LBP-3 protein alignment generated using t-coffee. Secondary structures are displayed above the alignment. The LBP-2 sequence utilized for the replacement in the LBP-3 chimeric protein is highlighted in purple. **g,h**, Intestine-specific overexpression of chimeric *lbp-3(chim)* does not affect the transcription of *egl-3* and *egl-21* (**g**) nor prolongs lifespan (**h**). In **a** and **b**, error bars represent mean ± s.e.m., *n* = 3 biologically independent samples; NS, *P* > 0.05 by two-way ANOVA with Holm–Sidak correction, ~2,000 worms per replicate. In **c** and **d**, *n* = 3 biologically independent samples. In **g**, error bars represent mean ± s.e.m., *n* = 4 biologically independent samples; NS, *P* > 0.05 by two-way ANOVA with Holm–Sidak correction, ~2,000 worms per replicate. In **h**, *n* = 3 biologically independent samples; NS, *P* > 0.05 by log-rank test, 98–120 worms per replicate. See Supplementary Table 4 for full lifespan data. Source numerical data are available in source data.

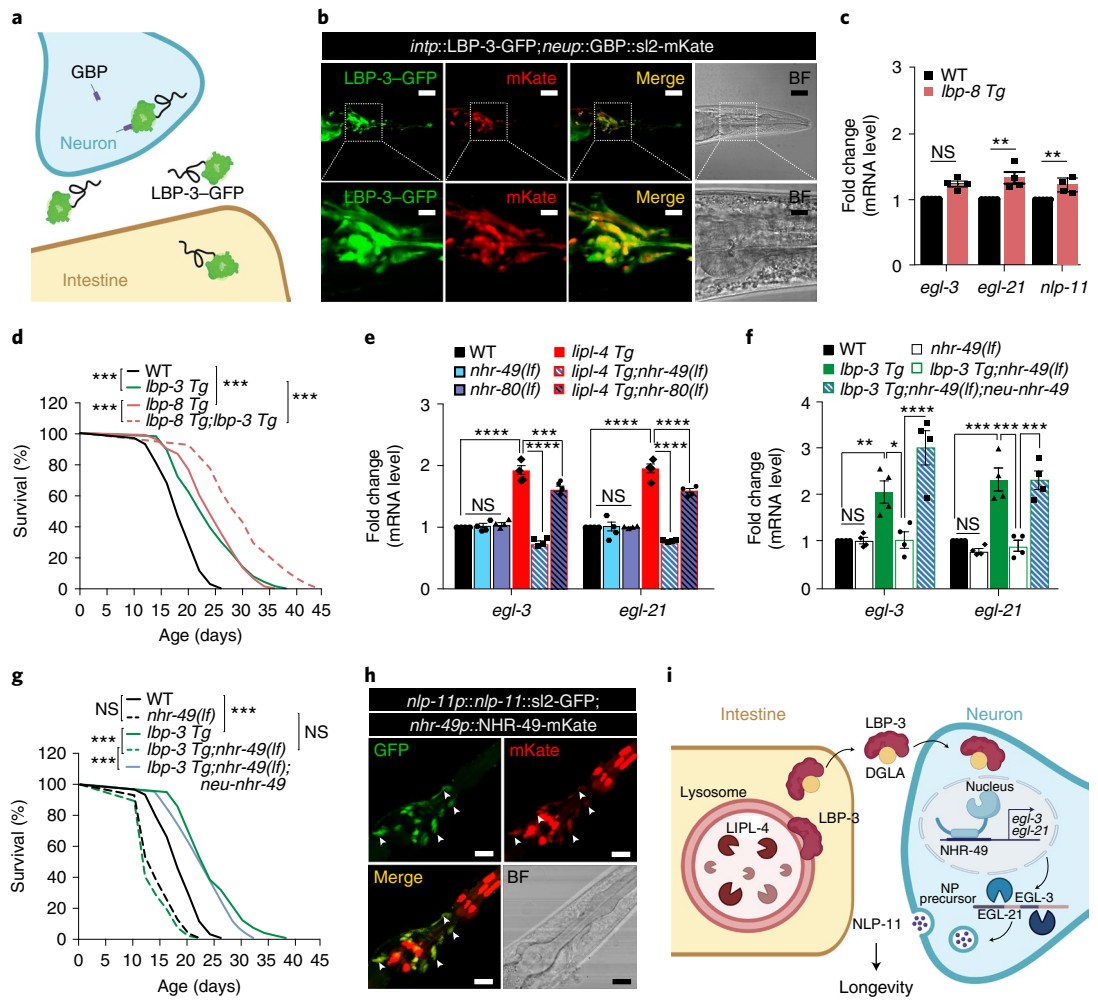

**Fig. 8 | Neuronal transduction of peripheral lipid signals to regulate longevity. a**, Schematic design using neuron-expressing GBP to detect the uptake of intestine-secreted LBP-3 into neurons. **b**, In the transgenic line expressing LBP-3–GFP in the intestine and GBP polycistronic mKate in neurons, GFP signals from secreted LBP-3 are detected in neurons (mKate positive). Scale bars, 30 μm and 10 μm in the inset. **c**, Constitutive expression of *lbp-8* (*lbp-8 Tg*) does not affect the transcription of *egl-3*, while the induction of either *egl-21* or *nlp-11* is negligible (<35%). **d**, *lbp-3 Tg* and *lbp-8 Tg* have an additive effect on lifespan extension. **e**, The loss-of-function mutation of the nuclear receptor *nhr-49(lf)* fully suppresses the induction of the neuropeptide genes by *lipl-4 Tg*, while the loss-of-function of *nhr-80(lf)* only decreases the induction by less than 17%. **f**, *nhr-49(lf)* fully suppresses the induction of the neuropeptide genes by *lbp-3 Tg*, which is rescued by neuronal restoration of *nhr-49*. **g**, *nhr-49(lf)* fully suppresses the lifespan extension in the *lbp-3 Tg* worms, which is rescued by neuronal restoration of *nhr-49*. **h**, In the transgenic line expressing *nlp-11* polycistronic GFP and mKate-fused NHR-49 driven by their endogenous promoters, GFP and mKate signals overlap in many neurons. Scale bar, 10 μm. **i,** Cartoon illustrating the overall model. In **c**, error bars represent mean ± s.e.m., *n* = 4 biologically independent samples, *\*P* = 0.034, *\*\*P* = 0.002 by two-way ANOVA with Holm–Sidak correction, ~2,000 worms per replicate. In **d** and **g**, *n* = 3 biologically independent samples, *\*\*\*P* < 0.001 by log-rank test, 80–101 worms per replicate. See Supplementary Table 4 for full lifespan data. In **e**, error bars represent mean ± s.e.m., *n* = 4 biologically independent samples, *\*\*\*P* = 0.0004 and *\*\*\*\*P* < 0.0001 by two-way ANOVA with Holm–Sidak correction, ~2,000 worms per replicate. In **f**, error bars represent mean ± s.e.m., *n* = 4 biologically independent samples, *\*P* = 0.019, *\*\*P* = 0.009 and *\*\*\*P* = 0.0006 for WT versus *lbp-3 Tg*, *P* = 0.0003 for *lbp-3 Tg* versus *lbp-3Tg;nhr-49(lf)* and *P* = 0.0003 for *lbp-3Tg;nhr-49(lf)* versus *lbp-3Tg;nhr-49(lf);neu-nhr-49*, *\*\*\*\*P* < 0.0001 by two-way ANOVA with Holm–Sidak correction, ~2,000 worms per replicate. Source numerical data are available in source data.

due to reduced DGLA secretion. Thus, secreted LBP-3 may regulate the basal expression of neuropeptides even without the induction of lysosomal lipolysis. Mammalian FABP4 secreted from adipocytes has been implicated in the hormonal control of metabolism[28], and FABP5 at the blood–brain barrier contributes to the brain uptake of docosahexaenoic acid, a PUFA essential for cognitive function[29]. Therefore, FABP secretion may function as an evolutionarily conserved mechanism to facilitate lipid transportation from peripheral metabolic tissues to the central nervous system.

Nuclear receptors are the best-known mediators of lipid signals in transcriptional responses, and several *C. elegans* nuclear receptors have been implicated in regulating longevity, including NHR-49, NHR-80, NHR-62 and DAF-12 (refs. [1,30–34]). In particular, PPARα has high binding affinity for FAs and plays a crucial role in metabolic tissues to regulate lipid catabolism[35]. PPARα also expresses at a high level in the nervous system; however, its neuronal function and regulation remain poorly understood. Our studies reveal that, in the nervous system, NHR-49 regulates neuroendocrine gene expression in response to peripheral lipid signals. Previous studies have also shown that neuronal NHR-49 mediates the longevity effect conferred by neuronal AMPK activation[36]. Thus, lipids may be crucial endocrine signals that couple peripheral metabolic status

with neuronal transcription via PPARα. We observed the internalization of secreted LBP-3 into neurons, and FABPs are known to cooperate with PPARs in regulating transcriptional responses[37]. We thus hypothesize that LBP-3–DGLA could directly act in concert with neuronal NHR-49 to regulate neuropeptide genes. However, we could not rule out the possibility that, within neurons, secondary lipid signals are derived from internalized DGLA and activate NHR-49 in the nucleus.

Our work highlights the crucial role of PUFAs in regulating *lipl-4*-induced longevity. The induction of PUFAs has also been linked with dietary-restriction-mediated longevity[38]. Previous studies have reported that monounsaturated fatty acid (MUFA) supplementation is sufficient to extend *C. elegans* lifespan[6]. One MUFA species, palmitoleic acid, is increased in *lipl-4 Tg*. However, through profiling the liposome bound to LBP-3, we did not detect an enrichment of palmitoleic acid. Although we could not rule out the possibility that the induction of palmitoleic acid contributes to *lipl-4*-induced longevity, it might not be involved in the LBP-3-mediated endocrine signalling mechanism.

## Online content

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

## Methods

**C. elegans strain maintenance.** *C. elegans* strains obtained from Caenorhabditis Genome Center or generated in this study are listed in Supplementary Table 7. Strains were maintained on standard nematode growth medium (NGM) agar plates seeded with corresponding bacteria at 20 °C. The full list of *C. elegans* strains is provided in Supplementary Table 7.

**Molecular cloning and generating transgenics.** Tissue-specific *lbp-3* or *nlp-11* expression vectors were generated using Multisite Gateway System (Invitrogen) as previously described[39].

For generating *nlp-11* and *lbp-3* transgenic lines driven by their endogenous promoters, the whole genomic region including the promoter, 5′ untranslated region (UTR), coding sequence and 3′ UTR were first PCR-amplified and then fused together with *sl2-GFP::unc-54 3′UTR* via fusion PCR. For *nlp-11* and *lbp-3*, 2.2 kbp and 1.1 kbp of the upstream promoter region was used, respectively.

The Gibson Assembly Method (NEB) was used to generate the following vectors. To amplify *egl-21* and *nhr-49*, 612 bp and 2.4 kbp of the upstream promoter region was used, respectively. *lbp-3ns* fused to both RFP and sl2::gfp, *lbp-3* fused to GFP, SAX-7 fused to GBP and GBP alone were amplified using tissue-specific promoter vectors. The chimeric *lbp-3* sequence was ordered using IDT, while the 3xHA sequence was PCR-amplified and ligated into the tissue-specific promoter vectors.

Transgenic strains were generated and integrated as previously described[1] and backcrossed to N2 at least five times.

**Generating deletion mutants using CRISPR.** All gene-specific mutations were generated using saturated single guide RNA (sgRNA) targeting throughout the *nlp-11* locus. sgRNAs were identified using the http://crispr.mit.edu/ website. Possible sgRNAs were then screened for predicted efficacy using http://crispr.wustl.edu/. For *nlp-11* deletion, we followed the protocol suggested by Dickinson et al.[40] and Ward et al.[41], while for *lbp-3* deletion, we identified candidates using the protocol suggested by Paix et al.[42] and Arribere et al.[43]. Genotyping PCR was performed using *nlp-11* and *lbp-3* spanning primers listed in Supplementary Table 8. Candidate worms with notable band shifting were saved and back-crossed at least four times with N2.

**Lifespan assays.** Lifespan assays were performed as previously described[1]. For integrated transgenic strains and newly isolated CRISPR deletion mutants, the strains were backcrossed at least five times before lifespan analysis (Supplementary Tables 4 and 5).

For lifespan assays involving strains containing mutation of *egl-21* or *fat-3*, 5′-fluorodeoxyuridine (FUDR) at a final concentration of 100 μM was added at L4 stage to prevent ageing-irrelevant lethality due to internal eggs' hatching. All the other lifespan assays did not use FUDR.

**qRT–PCR.** Total RNA was isolated as previously described[1]. Synthesis of complementary DNA was performed using the amfiRivert Platinum cDNA Synthesis Master Mix (GenDEPOT). Quantitative PCR was performed using Kapa SYBR fast PCR kit (Kapa Biosystems) in a Realplex 4 PCR machine (Eppendorf), and values were normalized to *rpl-32* as an internal control. All data shown represent three to four biologically independent samples. Primers used in this study are listed in Supplementary Table 8.

Intestines were dissected from day 1 adult *lipl-4 Tg* with *fat-3* RNAi supplemented by vehicle or DGLA for 12 h, and spun down at 20,000*g* for 2 min at 4 °C. Supernatant was removed, and 10 μl of worm lysis buffer (containing 1:100 diluted DNase; both from Ambion Power SYBR Green Cells-to-Ct kit) was added. Lysis reaction was incubated at room temperature for 5 min, stopped by adding 2 μl of Stop Solution and incubated at room temperature for 2 min following the manufacturer protocol. The synthesized cDNA was used undiluted for qRT–PCR.

**Fluorescent microscopy.** Day 1 adult worms were mounted on 2% agarose pads containing 0.5% NaN₃ as anaesthetic on glass microscope slides. Fluorescent images were taken using confocal FV3000 (Olympus). Polygon selection tool was used to select coelomocytes' area to be quantified, and average pixel intensity was calculated with the 'analyze-measure' command. A similar strategy was used when performing smFISH quantification using the polygon selection on the head region of the worms. After subtracting the background intensity, all measurements were averaged to obtain mean and standard deviation. In each imaging session, around 10–20 animals were analysed.

**Lysotracker staining.** LysoTracker Red DND-99 (Molecular Probes) was diluted in ddH₂O to 1,000 μM, and 6 μl was added to each 3.5 cm standard NGM plate (containing 3 ml of agar) seeded with OP50. The plates were kept in the dark for 24 h to allow the lysotracker solution to diffuse evenly throughout the plate. Approximately 10–20 worms were added to each plate at the L4 stage and kept in the dark for 1 day at 20 °C before confocal imaging.

**FFA profiling.** For each sample, 40,000 age-synchronized worms were grown on NGM plates seed with OP50 bacteria and collected as young adults. The worms were washed three times in M9 buffer and returned to empty NGM plates for 30 min for gut clearance. Following intestinal clearance, worms were washed twice more in M9 buffer, pelleted in a minimal volume of M9 and flash-frozen in liquid nitrogen. Proprietary recovery standards were added to each sample before extraction, and the samples were extracted using methanol, chloroform and water. The extracted samples in chloroform were dried and resuspended in methanol and isopropanol (50:50, vol/vol). The samples were analysed using a Vanquish UPLC and an LTQ-FT mass spectrometer with a linear ion-trap front end and a Fourier-transform ion cyclotron resonance back end (Thermo Fisher Scientific). The mobile phase A was 5 mM ammonium acetate with pH 5 and mobile phase B was 2-propanol and acetonitrile (20:80, vol/vol). The FFAs were first identified using lipidsearch 4.2.27 software (Thermo Fisher Scientific), and then validated using standards.

**Lysosome-specific lipidomics.** Briefly, we generated a transgenic strain expressing a C-terminal RFP- and HA-tagged lysosomal membrane protein LMP-1 driven by the ubiquitous promoter *sur-5* and crossed with *lipl-4 Tg* worms. For each sample, 200,000 age-synchronized worms were grown on NGM plates seed with OP50 bacteria and collected as young adults. The worms were washed three times in M9 and one time in KPBS followed by Dounce homogenization in ice until half of the worms were broken in half. The lysate was spun at 1,000*g* for 3 min at 4 °C to remove debris. The supernatant was incubated with 160 μl of anti-HA magnetic beads (Thermo Fisher Scientific) for 6 min at 20 °C and transferred to a magnetic stand followed by KPBS cold wash. The lipids were extracted by adding solvent methanol:water:methyl *tert*-butyl ether (3:2.5:10, v/v/v) into the beads. The organic top layer was dried and reconstituted with 100 μl methanol:isopropanol (1:1, v/v). The samples were analysed using a Vanquish UPLC and a Lumos orbitrap mass spectrometer (Thermo Fisher Scientific). The mobile phase A was 5 mM ammonium formate with 0.1% formic acid in water and acetonitrile (50:50, vol/vol) and mobile phase B consisted of 2-propanol, acetonitrile and water (88:10:2, vol/vol). A reverse-phase column Thermo Accucore Vanquish C18+ was used to separate the lipids, which were detected in both positive and negative ionization modes. Mass spectra were acquired in full-scan and data-dependent MS2 mode. For MS2 scanning, 20 dependent scans were acquired in each cycle. The MS2 resolution was 30k; HCD was used to fragment precursor ions with stepped collision energy 25, 30, 35; AGC target was 50,000. High-throughput analysis of lipidomic data was performed using Lipidsearch software (Thermo Fisher Scientific). Lipid quantification use precursor ion area and lipid were identified by matching product ion spectra to lipidsearch library. Both precursor and product ion mass tolerance were set at 5 ppm. M-score threshold was set 2.0. Both positive and negative data were aligned on the basis of retention time tolerance 0.1 min and mean value. The final data were filtered on the basis of preferred ion adduct for each lipid class.

**Lipid feeding.** Age-synchronized worms were grown on NGM plates seeded with OP50 bacteria to day 1 adulthood. AA, DGLA and EPA (Nu-Check Prep) were dissolved in DMSO and diluted into OP50 bacterial food to a final concentration of 1 mM. Then, 300 μL of each mixture was added to standard 6-cm NGM plates that were dried in a laminar flow hood under dark conditions. Worms were collected after 12 h of lipid feeding under dark conditions followed by RNA extraction and qRT–PCR.

**Protein expression and purification.** WT *C. elegans* LBP-2 (residues 19–161) and LBP-3 (residues 16–165) were subcloned into pMCSG7-His vector. LBP-2 and LBP-3 in the pMCSG7 vector was transformed into *Escherichia coli* strain BL21 (DE3) cells. Cultures (1 litre in LB) were grown to an $A_{600}$ of ~0.6 and induced with 0.5 mM isopropyl β-d-1-thiogalactopyranoside at 30 °C for 4 h, then harvested by centrifugation. For affinity purification and mass spectrometry studies with LBP-2, cells were lysed through sonication in a buffer containing 20 mM Tris–HCl pH 7.4, 500 mM NaCl, 25 mM imidazole, 5% glycerol, 5 mM 2-mercaptoethanol and 8 M urea. Unfolded LBP-2 was purified by nickel affinity chromatography in buffers containing 8 M urea. LBP-2 was refolded through stepwise dialysis over multiple days to remove urea. Refolded LBP-2 was further purified through size exclusion chromatography using a HiLoad 16/60 Superdex 75 column into a buffer containing 20 mM HEPES pH 7.5, 150 mM NaCl and 5% glycerol, 0.5 mM tris(2-carboxyethyl)phosphine. For affinity purification, ligand binding assays and mass spectrometry with LBP-3, cells were lysed through sonication in a buffer containing 20 mM Tris–HCl pH 7.4, 500 mM NaCl, 25 mM imidazole, 5% glycerol, 5 mM 2-mercaptoethanol, lysozyme, Dnase A and 100 μM phenylmethylsulfonyl fluoride. LBP-3 was purified by nickel affinity chromatography and followed by size exclusion chromatography using a HiLoad 16/60 Superdex 75 column.

**Competitive fluorescence-based binding assay.** Quantification of ligand binding was conducted via competition of the probe 1,8-ANS as previously described[1].

**LBP-2 and LBP-3 binding with *C. elegans* liposome.** For each sample, 100,000 age-synchronized worms were grown on NGM plates seeded with OP50 bacteria and collected as young adults. The worms were washed three times in M9 buffer, one time in 1× PBS, pelleted in a minimal volume of 1× PBS and flash-frozen in liquid nitrogen. Lipids were extracted from *C. elegans* lysates using the Bligh and Dyer method[44].

Fatty acids were extracted from LBP-2 and LBP-3 after binding with *C. elegans* lipid extracts. Fatty acid derivatives were generated as previously described[45]. Briefly, dried lipid extracts were incubated with 200 µl of oxalyl chloride (2 M in dichloromethane) at 65 °C for 5 min, then dried down with nitrogen gas. Then, 3-picolylamide fatty acid derivatives were formed through incubation with 3-picolylamine (1% in acetonitrile) at room temperature for 5 min and then dried down with nitrogen gas.

Mass spectrometry for LBP-2: 5 µl of the LBP-2-derived fatty acid sample resuspended in methanol was injected onto a ThermoScientific Accucore C18 (4.6 × 100 mm, 2.6 µm) column using the ExionLC AD UPLC system at a 0.8 ml min⁻¹ flow rate, and a gradient solvent system containing 10 mM ammonium acetate, pH 7 in $H_2O$ (solvent A) and 10 mM ammonium acetate, pH 7 in 100% acetonitrile (solvent B). Samples were chromatographically resolved using a stepwise gradient starting at 40% solvent B for 3 min, 100% solvent B for 5 min and then 65% solvent B for 2 min. Derivatized fatty acids were detected using ABSciex QTrap5500 triple quadrupole mass spectrometer in positive ion mode. The following multiple reaction-monitoring transitions were used to detect the most abundant derivatized fatty acids. Derivatized fatty acids were quantified in Multiquant 3.0.2 software (AB Sciex) using a calibration curve with the following fatty acids: myristic acid, palmitic acid, oleic acid, linoleic acid, AA and docosahexaenoic acid.

Mass spectrometry for LBP-3: 20 µl of the LBP-3-derived fatty acid sample resuspended in ethanol was injected onto a Zorbax Eclipse Plus C18 (2.1 × 100 mm, 3.5 µm) at a 0.15 ml min⁻¹ flow rate with column temperature of 45 °C, and a gradient solvent system containing solvent A and solvent B (both as above). Samples were chromatographically resolved using a stepwise gradient starting at 10% solvent B for 4 min, 40% solvent B for 3 min, 100% solvent B for 4 min, 65% solvent B for 1 min, 40% solvent B for 2 min and then 10% solvent B for 6 min. Derivatized fatty acids were detected using an Agilent 6495c triple quadrupole mass spectrometer coupled to an Agilent 1295 II UPLC in positive ion mode. The following multiple reaction-monitoring transitions were used to detect the most abundant derivatized fatty acids. Derivatized fatty acids were quantified in Mass Hunter Quantitative Analysis 10.1 software using a calibration curve with the following fatty acids: decanoic acid, undecanoic acid, dodecanoic acid, myristic acid, palmitic acid, oleic acid, linoleic acid, stearic acid, EPA, AA and DGLA.

**Western blot.** At least 300 worms per genotype were grown on seeded NGM plates before being collected and snap-frozen on dry ice. The samples were lysed in worm lysis buffer (50 mM Tris–HCl pH 7.4, 150 mM NaCl, 1 mM EDTA and 0.1% NP-40) containing a protease and phosphatase inhibitor cocktail (cOmplete Protease Inhibitor Cocktail, cat. no. 11697498001; PhosSTOP, cat. no. 4906845001; both from Sigma) and homogenized with motorized pellet pestle. The lysates were then centrifuged, and the supernatants were used for protein quantification and western blotting analysis. Next, the proteins were separated with the NuPAGE system (Thermo Fisher, 4–12% Bis–Tris protein gel), and transferred to PVDF membrane (Thermo Fisher). The membranes were blocked with 5% BSA in TBST. The primary antibody against HA is anti-HA rabbit mAb (Cell Signaling, #C29F4, 1:1,000), which detects the HA tag and was tested for its specificity using WT and LBP-3::HA lysates. The anti-β-actin antibody is from Santa Cruz (sc-47778, 1:2,000). Protein detection was performed using chemiluminescent substrate (ECL Western Blotting Reagents, Sigma-Aldrich, GERPN2106) and images acquired using a gel imaging system (ImageQuant LAS 500, Thermo Fisher Scientific).

**smFISH.** Custom Stellaris FISH Probe was designed against *egl-21* by utilizing the Stellaris RNA FISH Probe Designer (Biosearch Technologies, www.biosearchtech.com/stellarisdesigner). The samples were hybridized with the *egl-21* Stellaris RNA FISH Probe set labelled with Quasar 670 (Biosearch Technologies), following the manufacturer's instruction. Briefly, around 300 worms were washed from small plates with M9, transferred to 1.5 ml Eppendorf tubes, resuspended in 1 ml fixation buffer (3.7% v/v formaldehyde in 1× PBS) and incubated at room temperature for 45 min with rotation. Fixed worms were washed twice with 1× PBS, resuspended in borate triton ß-mercaptoethanol solution buffer and kept at 4 °C overnight to permeabilize the cuticle. Worms were washed twice with Borate triton solution, twice with PBST and once in 1× PBS. Samples were resuspended in 1 ml wash buffer (10% v/v formamide in 1× Wash Buffer A, Biosearch Technologies, cat. no. SMF-WA1-60) at room temperature for 2–5 min, before being hybridized for 12 h at 37 °C in 100 µl hybridization buffer (10% v/v formamide in Hybridization Buffer, Biosearch Technologies, cat. no. SMF-HB1-10) containing 1 µl of 125 nM final concentration reconstituted probe. Samples were incubated in wash buffer at 37 °C for 30 min, then resuspended in 1 ml of DAPI nuclear staining (5 ng ml⁻¹ DAPI in wash buffer). Worms were incubated in the dark at 37 °C for 30 min and resuspended in 1 ml of wash buffer at room temperature for 2–5 min. Worms were transferred using a Pasteur pipette into a small drop of Vectashield Mounting Medium (Vector Laboratories), before imaging.

**RNA-seq preparation and analysis.** Total RNA from WT and *lipl-4 Tg* was extracted from around 3,000 worms in three different biological replicates using Trizol extraction combined with column purification (Qiagen). Total RNA from WT and *lbp-3 Tg* was extracted using phenol–chloroform method. Sequencing libraries were prepared using the TruSeq Stranded mRNA Sample Preparation kit (Illumina) following the manufacturer's instructions. Libraries were pooled together and sequenced using Illumina NextSeq 500 system. RNA-seq reads were aligned to the *C. elegans* reference genome using hisat2 with the default setting. HTSeq was used to count the read numbers mapped to each gene. DESeq2 was used to normalize the raw counts and identify differentially expressed genes (|fold change| ≥1.5; false discovery rate <0.05).

**Protein sequence alignment.** The alignment was generated by tcoffee and illustrated by ESPript.

**AlphaFold2 structures of the *C. elegans* LBP-2 and LBP-3.** The *C. elegans* LBP-2 (F40F4.2) and LBP-3 protein (F40F4.4) sequences were downloaded from wormbase. The LBP-2 and LBP-3 structures were predicted using AlphaFold2 through the jupyter notebook for ColabFold (https://doi.org/10.1101/2021.08.15.456425 and https://doi.org/10.1038/s41586-021-03819-2). The ColabFold was used with the default parameters, and sequence alignments were generated by MMseqs2 (https://doi.org/10.1101/2021.08.15.456425 and https://doi.org/10.1038/nbt.3988). The structure images were generated by UCSF Chimera.

**Statistics and reproducibility.** For all figure legends, asterisks indicate statistical significance as follows: NS, not significant ($P > 0.05$), *$P < 0.05$, **$P < 0.01$, ***$P < 0.001$ and ****$P < 0.0001$. Data were obtained by performing independently at least three biological replicates, unless specified in the figure legends. No statistical method was used to pre-determine the sample size. No data were excluded from the analyses. Two-tailed Student's *t*-test or one-way or two-way analysis of variance (ANOVA) with Holm–Sidak corrections was used as indicated in the corresponding figure legends. *n* indicates the number of biological replicates. For survival analysis, statistical analyses were performed with SPSS software (IBM) using Kaplan–Meier survival analysis and log-rank test. Details on samples size, number of biological replicates and statistics for each experiment are provided in Supplementary Tables 4 and 5. For FFA profiling, statistical analysis was performed using two-way ANOVA test with Holm–Sidak correction comparing *lipl-4 Tg* ($n = 9$ biological replicates) vs. WT (n=6 biological replicates), while for lipid profiling using lysosomal isolation, t-test was used to compare *lipl-4 Tg* (n=4 biological replicates) versus WT ($n = 3$ biological replicates). For RNA-seq, two-sided Wald test in R package DEseq2 was used. For qRT–PCR, *t*-test or one-way or two-way ANOVA with Holm–Sidak correction was used as indicated in the corresponding figure legends. Figures and graphs were constructed using GraphPad Prism 7 (GraphPad Software) and Illustrator (CC 2019; Adobe). The researchers involved in the study were not blinded during experiments or outcome assessment.

**Reporting summary.** Further information on research design is available in the Nature Research Reporting Summary linked to this article.

## Data availability

Deep-sequencing (RNA-seq) data that support the findings of this study have been deposited into the NCBI Sequence Read Archive, and the accession codes for each biological sample are SAMN25414087, SAMN25414088, SAMN25414089, SAMN25414090, SAMN25414091, SAMN25414002, SAMN25414093, SAMN25414094, SAMN25414095, SAMN25414096, SAMN25414097 and SAMN25414098. Source data are provided with this paper. The lipidomics data are deposited into the metabolights database at the following link: https://www.ebi.ac.uk/metabolights/MTBLS4654. The wormbase https://wormbase.org is used for searches related to *C. elegans*. All other data supporting the findings of this study are available from the corresponding author on reasonable request.

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

## Acknowledgements

We thank A. Dervisefendic and P. Svay for maintenance support; J. Mello (Harvard Medical School, USA) for providing strain JM45; K. Shen for sharing the vector encoding for sax-7::GBP; S. Mutlu, P. Rohs, S. M. Gao, X. Ma, L. Ding, C. Herman, H. Dierick and B. Arenkiel for critical reading of the manuscript. Some strains were obtained from the Caenorhabditis Genetics Center (CGC), which is funded by NIH Office of Research Infrastructure Programs (P40 OD010440). This work was supported by NIH grants R01AG045183 (M.C.W.), R01AT009050 (M.C.W.), R01AG062257 (M.C.W.), DP1DK113644 (M.C.W.), P01AG066606 (M.C.W.), RF1AG074540 (M.C.W.), T32 ES027801 pre-doctoral fellow (M.S.), R03AG070417 (L.H.), R01HG011633 (L.H.), R01CA262623 (L.H.), Welch Foundation Q-1912-20190330 (M.C.W.) and HHMI investigator (M.C.W.). We thank WormBase.

## Author contributions

M.S., A.F. and M.C.W. conceived the project and designed the experiments. M.S., A.F., L.Y., F.J., A.C., M.C.T., I.A.N., P.H. and Y. Yu performed experiments, Q. Zhang conducted *lip-4 Tg* RNA-seq, and J.D.D. conducted *lbp-3 Tg* RNA-seq. Q. Zhao conducted the structural simulation and alignment, and Y. Ye conducted the bioinformatic transcriptome analysis. M.S. and M.C.W. wrote the manuscript. M.S., F.J., Q. Zhao, M.C.T., A.C., W.B.M., E.A.O., L.H., J.W. and M.C.W. edited the manuscript.

## Competing interests

J.W. is a cofounder of Chemical Biology Probes LLC and Coactigon Inc. The focuses of these companies are unrelated to this study. The other authors declare no competing interests.

## Additional information

**Extended data** is available for this paper at https://doi.org/10.1038/s41556-022-00926-8.

**Correspondence and requests for materials** should be addressed to Meng C. Wang.

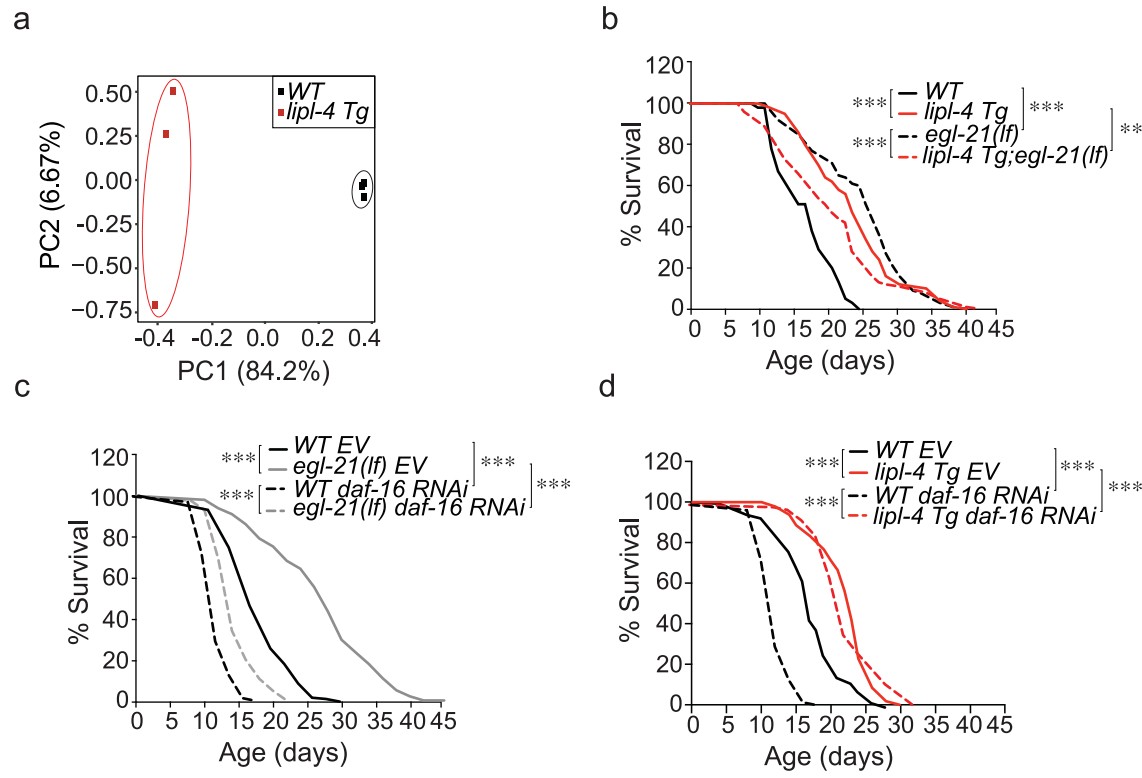

**Extended Data Fig. 1 | Peripheral lysosomal lipolysis up-regulates neuropeptide signaling (related to Fig. 1). a)** Principal Component Analysis (PCA) of *lipl-4 Tg* and WT worms falling in two distinct clusters. **b)** The loss-of-function mutant of *egl-21(lf)* shows lifespan extension, and *lipl-4 Tg* cannot further enhance this lifespan extension as it does in the WT condition. **c)** In the background of *daf-16* RNAi inactivation, the longevity effect caused by *egl-21(lf)* is suppressed. *daf-16* RNAi is thus used to eliminate the contribution from ILP reduction in the *egl-21(lf)* mutant. **d)** Inactivation of *daf-16* does not affect the longevity effect of *lipl-4 Tg*. (a) PCA analysis using two-sided statistical test, (b-d) n=3 biologically independent samples, n.s. *p*>0.05 and *** *p*<0.001 by long-rank test, 60-120 worms per replicate. See Supplementary Tables 4-5 for full lifespan data.

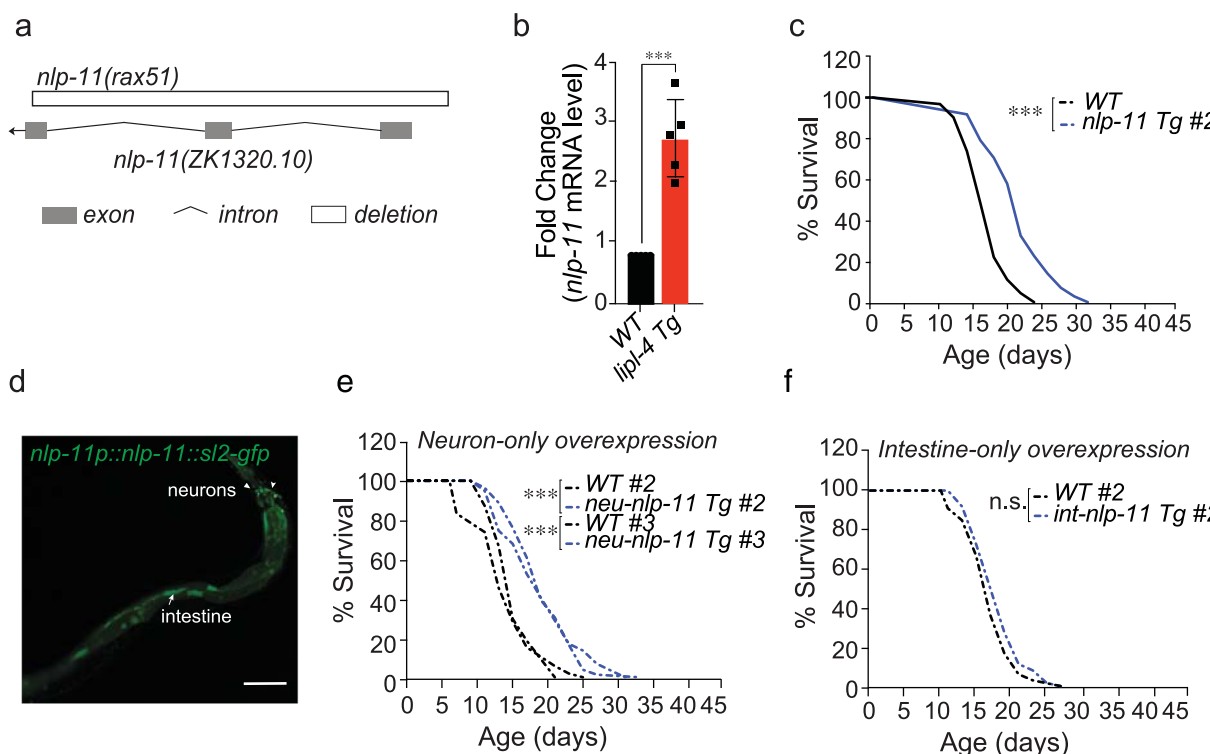

**Extended Data Fig. 2 | NLP-11 neuropeptide acts in neurons to promote longevity (related to Fig. 2). a**) A schematic representation of the *nlp-11(rax51)* loss-of-function mutation. White boxes represent the *nlp-11(rax51)* deletion, black lines represent introns. The mutant lacks the three exons and the transcriptional start site. **b**) *nlp-11* is transcriptionally up-regulated by *lipl-4 Tg*. **c**) Constitutive expression of *nlp-11* driven by its endogenous promoter extends lifespan. **d**) The transgenic strains expressing *nlp-11* under its endogenous promoter *(nlp-11p::nlp-11::sl2-GFP)* reveals the expression of *nlp-11* in intestinal and neuronal cells. Scale bar 100μm. **e, f**) Neuron-specific overexpression of *nlp-11* prolongs lifespan (**e**), but intestine-specific overexpression has no such effect (**f**). (b) Error bars represent mean ± s.e.m., n=3 biologically independent samples, **** *p*<0.0001 by two-tailed Student's t-test, ~2000 worms per replicate. (c, e, f) n=3 biologically independent samples, n.s. *p*>0.05 and *** *p*<0.001 by long-rank test, 60-120 worms per replicate. See Supplementary Table 4 for full lifespan data. Source numerical data are available in source data.

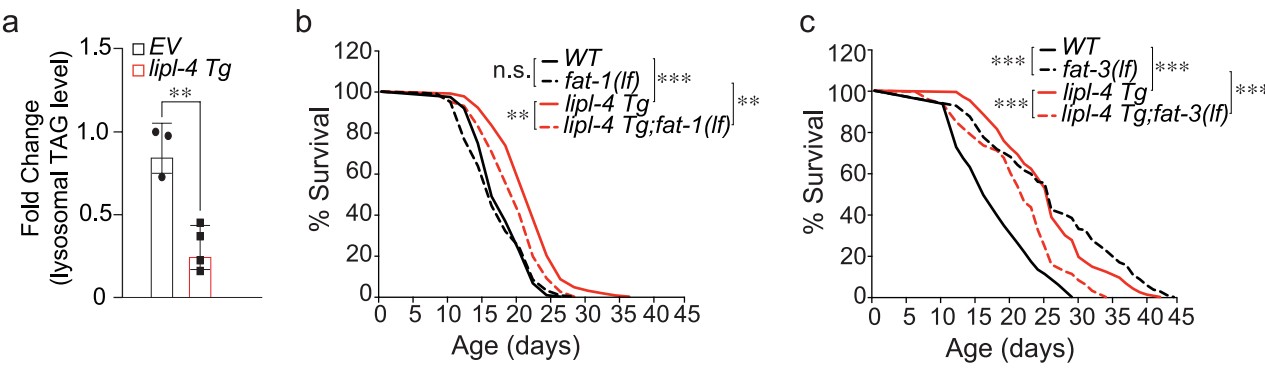

**Extended Data Fig. 3 | Lysosome-derived PUFAs in the periphery regulate neuropeptide and longevity (related to Fig. 3). a)** The level of TAG is decreased in the purified lysosomes from *lipl-4 Tg* compared to those from WT worms. **b)** The loss-of-function mutant *fat-1(lf)* suppresses *lipl-4 Tg* longevity. **c)** In the loss-of-function mutant of *fat-3(lf)*, the lifespan-extending effect of *lipl-4 Tg* is shortened. (a) Error bars represent mean ± s.e.m., n=3 (WT) and n=4 (*lipl-4 Tg*) biologically independent samples, **p=0.0025 by two-tailed Student's t-test, ~ 200,000 worms per replicate. (b, c) n=3 biologically independent samples, n.s. *p*>0.05 and *** *p*<0.001 by long-rank test, 60-120 worms per replicate. See Supplementary Table 4 for full lifespan data. Source numerical data are available in source data.

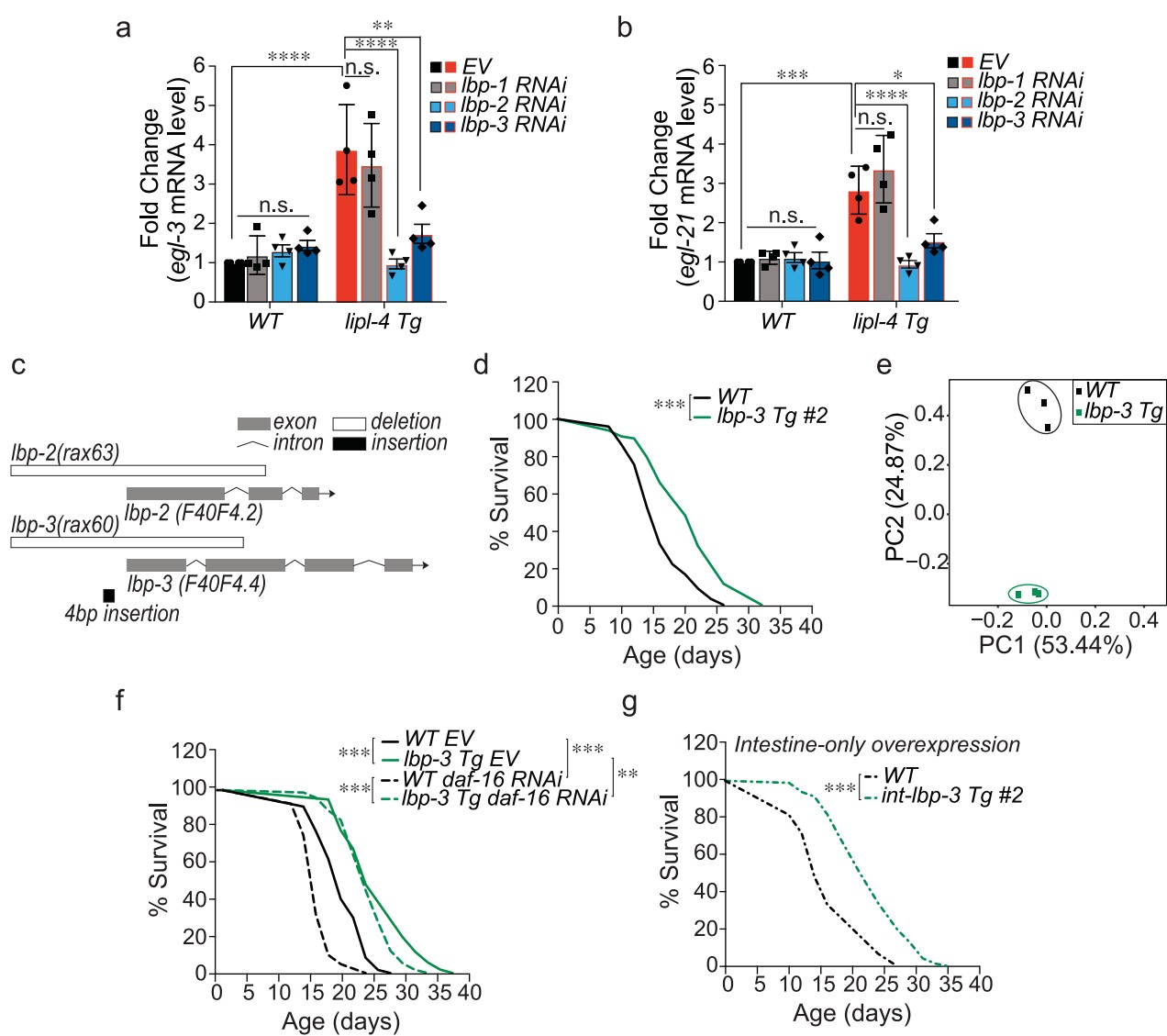

**Extended Data Fig. 4 | Peripheral lipid chaperone LBP-3 regulates neuropeptide and longevity (related to Fig. 4). a, b)** The induction of *egl-3* **(a)** and *egl-21* **(b)** in *lipl-4 Tg* is suppressed by RNAi inactivation of either *lbp-2* or *lbp-3*, but not *lbp-1*. **c)** Schematic representation of *lbp-2* and *lbp-3* loss-of-function mutants. White boxes represent the *lbp-2(rax63)* and *lbp-3(rax60)* deletions, black boxes represent insertions, while black lines represent introns. The mutants lack the entire first exon and transcriptional start site. **d)** The transgenic strain that carries constitutive expression of *lbp-3* (*lbp-3 Tg*) driven by its own endogenous promoter prolongs lifespan. **e)** PCA analysis of *lbp-3 Tg* and WT worms falling in two distinct clusters. **f)** Inactivation of *daf-16* does not affect the longevity effect of *lbp-3 Tg*. *daf-16* RNAi knockdown used to eliminate the influence from ILP reduction in *egl-21(lf)*. **g)** Overexpression of *lbp-3* selectively in the intestine extends lifespan. (a, b) Error bars represent mean ± s.e.m., n=4 biologically independent samples, n.s. *p*>0.05, *\*p=0.0104*, *\*\*p=0.0018*, *\*\*\*p=0.0001* and *\*\*\*\*p<0.0001* by two-way ANOVA with Holm-Sidak correction, ~2000 worms per replicate. (e) PCA analysis using two-sided statistical test. (d, f, g) n=3 biologically independent samples, n.s. *p*>0.05 and *\*\*\* p<0.001* by long-rank test, 60-120 worms per replicate. See Supplementary Tables 4-5 for full lifespan data. Source numerical data are available in source data.

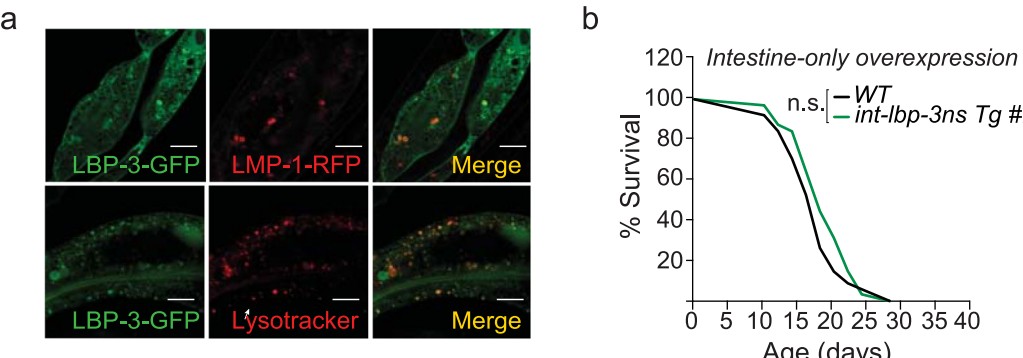

**Extended Data Fig. 5 | LBP-3 secreted from the periphery regulates neuropeptide and longevity (related to Fig. 5). a)** LBP-3 and the lysosomal membrane protein LMP-1 are visualized by their GFP and RFP fusions, respectively. LBP-3::GFP colocalizes with LMP-1::RFP and Lysotracker Red staining at lysosomes, and is also detected in the cytosol. Scale bar 10μm. **b)** Intestine-specific overexpression of *lbp-3* lacking its secretory signal (*lbp-3ns*) fails to extend lifespan. n=3 biologically independent samples, n.s. *p*>0.05 by long -rank test, 72-100 worms per replicate. See Supplementary Table 4 for full lifespan data.

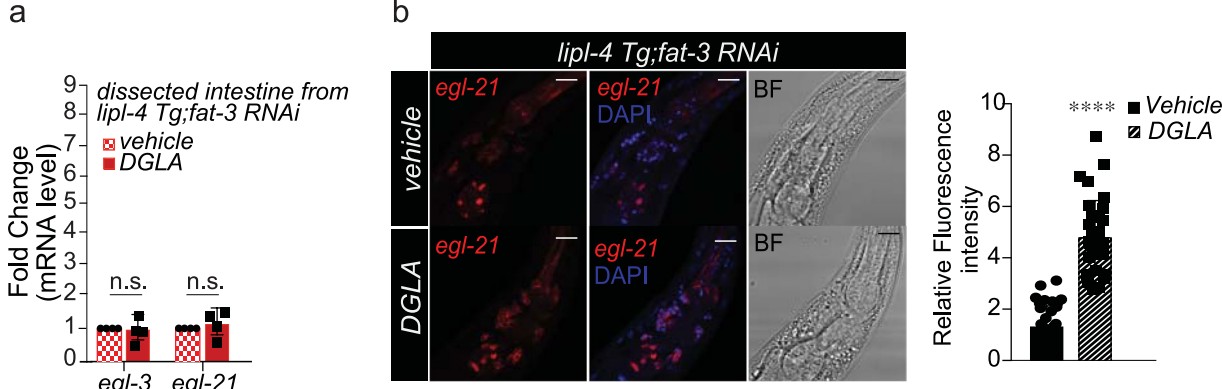

**Extended Data Fig. 6 | DGLA regulates LBP-3 secretion, neuropeptides and longevity (related to Fig. 6). a)** Intestine was dissected from *lipl-4 Tg* with *fat-3* RNAi knockdown and used for qPCR analysis. The expression of *egl-21* is not detectable in the dissected intestine, while the expression of *egl-3* is weakly detected. The transcriptional level of either *egl-3* or *egl-21* is not affected by DGLA supplementation. DMSO serves as the vehicle control. **b)** *egl-21* transcripts are measured in *lipl-4 Tg* with *fat-3* RNAi using smFISH (red Quasar 670). Upon DGLA supplementation, the *egl-21* transcript level is increased compared to the vehicle control. Scale bar 10µm. Representative images from three biological repeats. (a) Error bars represent mean ± s.e.m., n=3 biologically independent samples, n.s. *p*>0.05 by two-way ANOVA with Holm-Sidak correction, ~20 dissected intestines per replicate. (b) n=3 biologically independent samples, ****p<0.0001 by two-tailed Student's t-test, 12 worms per replicate. Source numerical data are available in source data.

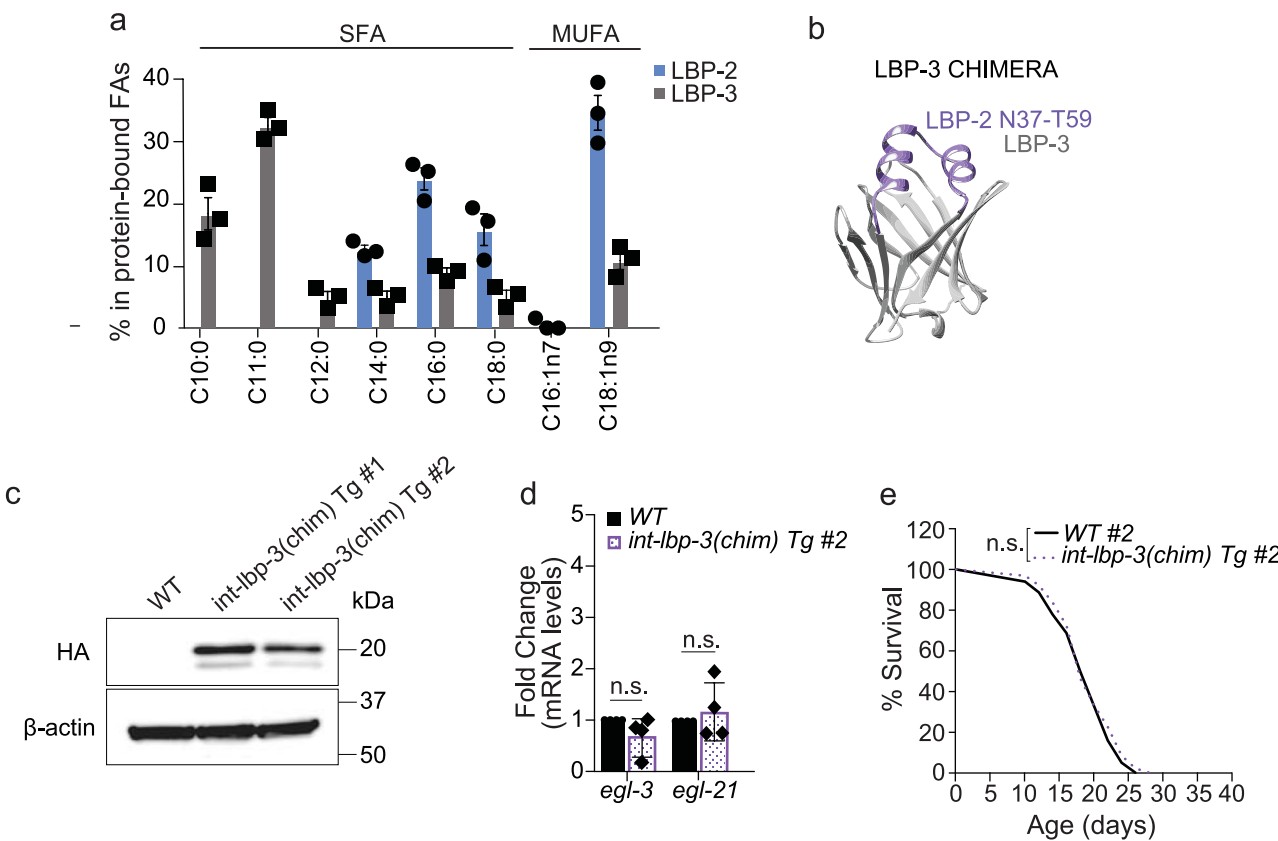

**Extended Data Fig. 7 | DGLA binding specificity of LBP-3 mediates its effects (related to Fig. 7). a)** After incubation with the *C. elegans* liposome, fatty acids bound to LBP-2 or LBP-3 proteins were analyzed with mass spectrometry. In addition to PUFAs shown in Fig. 5, SFAs and MUFAs also bind to LBP-2 and LBP-3 with different preferences. **b)** The LBP-3 chimeric protein structure predicted using AlphaFold2. The two helix regions from LBP-2 are highlighted in purple. **c)** Western-blot of WT and transgenic worms overexpressing LBP-3 chimeric proteins fused with 3XHA-tag. LBP-3::3xHA fusion proteins are detected in both chimeric lines. ß-actin is used as a control. **d, e)** Intestine-specific overexpression of the chimeric *lbp-3(chim)* does not affect the transcription of *egl-3* or *egl-21* (**d**) and fails to extend lifespan (**e**). (d) Error bars represent mean ± s.e.m., n=3 biologically independent samples, n.s. *p*>0.05 by two-way ANOVA with Holm-Sidak correction, ~300 worms per replicate. (e) n=3 biologically independent samples, n.s. *p*>0.05 by long-rank test, 72-100 worms per replicate. See Supplementary Table 4 for full lifespan data. Source numerical data and unprocessed blots are available in source data.

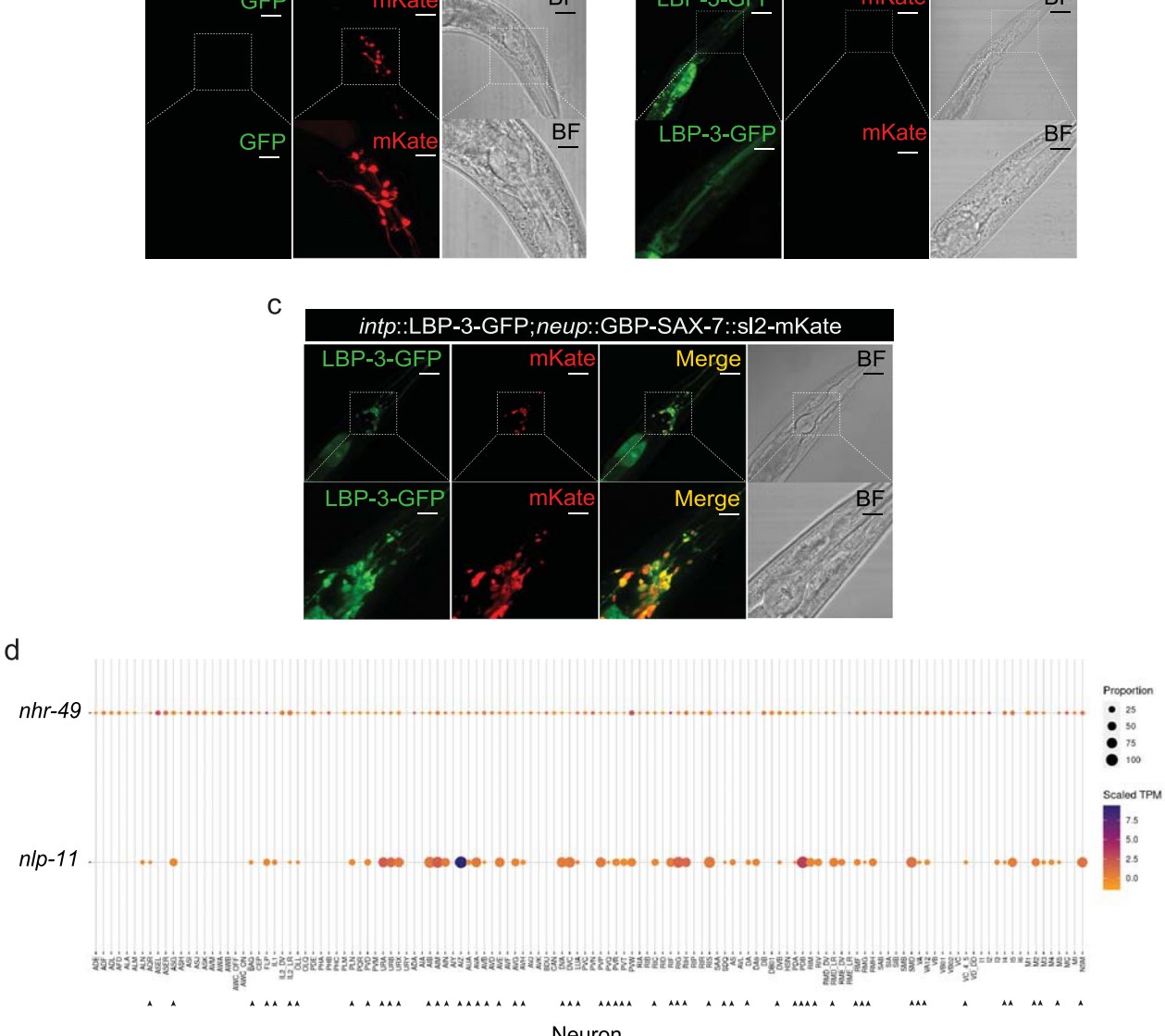

**Extended Data Fig. 8 | Neuronal transduction of peripheral lipid signals to regulate longevity (related to Fig. 8). a)** In the transgenic line expressing GBP polycistronic mKate in neurons alone, no GFP signals are detected in neurons. Scale bar 30μm and 10μm in the inset. **b)** In the transgenic line expressing secretable LBP-3::GFP selectively in the intestine alone, GFP signals are only strongly detected in the intestine and pharynx. Scale bar 30μm and 10μm in the inset. **c**) In the transgenic line expressing GBP tagged with the extracellular domain of SAX-7 (GBP::SAX-7) and polycistronic mKate in neurons and secretable LBP-3::GFP in the intestine, strong GFP signals are detected in neurons. Scale bar 30μm and 10μm in the inset. **d)** Heatmap showing averaged *nlp-11* and *nhr-49* gene expression per neuronal cell type (threshold equal to two). The color of circles represents relative gene expression, while the size of circles indicates the percentage of cells expressing the gene within each neuron cluster. The heatmap is generated using the online webtool CenGeneApp. Neuronal cell types shared between *nhr-49* and *nlp-11* are marked by black arrows.

# Reporting Summary

## Statistics

For all statistical analyses, confirm that the following items are present in the figure legend, table legend, main text, or Methods section.

| n/a | Confirmed | |
|---|---|---|
| ☐ | ☒ | The exact sample size (*n*) for each experimental group/condition, given as a discrete number and unit of measurement |
| ☐ | ☒ | A statement on whether measurements were taken from distinct samples or whether the same sample was measured repeatedly |
| ☐ | ☒ | The statistical test(s) used AND whether they are one- or two-sided *Only common tests should be described solely by name; describe more complex techniques in the Methods section.* |
| ☒ | ☐ | A description of all covariates tested |
| ☐ | ☒ | A description of any assumptions or corrections, such as tests of normality and adjustment for multiple comparisons |
| ☒ | ☐ | A full description of the statistical parameters including central tendency (e.g. means) or other basic estimates (e.g. regression coefficient) AND variation (e.g. standard deviation) or associated estimates of uncertainty (e.g. confidence intervals) |
| ☐ | ☒ | For null hypothesis testing, the test statistic (e.g. *F*, *t*, *r*) with confidence intervals, effect sizes, degrees of freedom and *P* value noted *Give P values as exact values whenever suitable.* |
| ☒ | ☐ | For Bayesian analysis, information on the choice of priors and Markov chain Monte Carlo settings |
| ☒ | ☐ | For hierarchical and complex designs, identification of the appropriate level for tests and full reporting of outcomes |
| ☒ | ☐ | Estimates of effect sizes (e.g. Cohen's *d*, Pearson's *r*), indicating how they were calculated |

*Our web collection on statistics for biologists contains articles on many of the points above.*

## Software and code

Policy information about availability of computer code

| Data collection | Fluorescent images were taken using confocal FV3000 (Olympus). Western-blot images were acquired using Image Quant LAS 500. Free fatty acids were identified using lipidsearch 4.2.27 software (Thermo Fisher Scientific). |
|---|---|
| Data analysis | The commercial softwares used for data collection are specified in the methods section. |

For manuscripts utilizing custom algorithms or software that are central to the research but not yet described in published literature, software must be made available to editors and reviewers. We strongly encourage code deposition in a community repository (e.g. GitHub). See the Nature Portfolio guidelines for submitting code & software for further information.

## Data

Policy information about availability of data

All manuscripts must include a data availability statement. This statement should provide the following information, where applicable:
- Accession codes, unique identifiers, or web links for publicly available datasets
- A description of any restrictions on data availability
- For clinical datasets or third party data, please ensure that the statement adheres to our policy

All data generated or analyzed during this study are included in this manuscript and the Extended Data Materials. The RNA-seq data has been deposited into the NCBI Sequence Read Archive (SRA) and the accession codes for each biological sample are SAMN25414087, SAMN25414088, SAMN25414089, SAMN25414090, SAMN25414091, SAMN25414002, SAMN25414093, SAMN25414094, SAMN25414095, SAMN25414096, SAMN25414097, SAMN25414098, and the BioProject accession code is PRJNA801907 (https://www.ncbi.nlm.nih.gov/bioproject/PRJNA801907).

# Field-specific reporting

Please select the one below that is the best fit for your research. If you are not sure, read the appropriate sections before making your selection.

☒ Life sciences  ☐ Behavioural & social sciences  ☐ Ecological, evolutionary & environmental sciences

For a reference copy of the document with all sections, see nature.com/documents/nr-reporting-summary-flat.pdf

# Life sciences study design

All studies must disclose on these points even when the disclosure is negative.

| | |
|---|---|
| Sample size | No sample-size calculation was performed. Sample size was used based on previous literature and recognized in the worm field. For each experiment, n values are provided in the figure legends and in the section of Statistics and Reproducibility. The amount of animals used for each experiment is reported in the Method session. |
| Data exclusions | No data were excluded from the analysis in the study |
| Replication | All experimental replications were repeated at least three independent times and the detailed information is specified in Figure Legends and Statistics And Reproducibility |
| Randomization | Worms with the same genotype were randomly chosen and grouped into experiments. |
| Blinding | Investigators were not blinded, as for each experiment and analysis worms from different genotypes were compared to wild-type/control |

# Reporting for specific materials, systems and methods

We require information from authors about some types of materials, experimental systems and methods used in many studies. Here, indicate whether each material, system or method listed is relevant to your study. If you are not sure if a list item applies to your research, read the appropriate section before selecting a response.

### Materials & experimental systems

| n/a | Involved in the study |
|---|---|
| ☐ | ☒ Antibodies |
| ☒ | ☐ Eukaryotic cell lines |
| ☒ | ☐ Palaeontology and archaeology |
| ☐ | ☒ Animals and other organisms |
| ☒ | ☐ Human research participants |
| ☒ | ☐ Clinical data |
| ☒ | ☐ Dual use research of concern |

### Methods

| n/a | Involved in the study |
|---|---|
| ☒ | ☐ ChIP-seq |
| ☒ | ☐ Flow cytometry |
| ☒ | ☐ MRI-based neuroimaging |

## Antibodies

| | |
|---|---|
| Antibodies used | Anti-HA, Rabbit, Cell Signaling, #C29F4, https://www.cellsignal.com/products/primary-antibodies/ha-tag-c29f4-rabbit-mab/3724  Anti-beta-actin, Mouse, Santa Cruz, sc-47778, https://www.scbt.com/p/beta-actin-antibody-c4 |
| Validation | Validation statement from the company: HA-Tag (C29F4) Rabbit mAb detects exogenously expressed proteins containing the HA epitope tag in all the species specified including C. elegans. In our study, wild type worms that do not express HA-tagged proteins were used as negative controls for the Anti-HA antibody (Cell Signaling, #C29F4). Validation of Anti-beta-actin, Mouse, Santa Cruz, sc-47778 was previously described (PMID:27534274, PMID:32966783, PMID: 30599151). |

## Animals and other organisms

Policy information about studies involving animals; ARRIVE guidelines recommended for reporting animal research

| | |
|---|---|
| Laboratory animals | Caenorhabditis elegans used in this study:  N2 Bristol Strain  raxIs3[ges-1p::lipl-4::sl2-GFP; myo-2p::mCherry]  egl-21(n476)  raxIs3[ges-1p::lipl-4::sl2-GFP; myo-2p::mCherry];egl-21(n476)  fat-3(wa22)  raxIs3[ges-1p::lipl-4::sl2-GFP; myo-2p::mCherry];fat-3(wa22)  fat-1(wa9) |

raxIs3[ges-1p::lipl-4::sl2-GFP; myo-2p::mCherry];fat-1(wa9)
lbp-3(rax60)
raxIs3[ges-1p::lipl-4::sl2-GFP; myo-2p::mCherry];lbp-3(rax60)
raxIs141[lbp-3p::lbp-3::sl2-GFP; myo-2p::mCherry]
raxIs119 [lbp-3p::lbp-3::gfp, myo-2p::mCherry]
Ex440[ges-1p::lbp-3::sl2-GFP;myo-2p::mCherry]
Ex441[ges-1p::lbp-3::sl2-GFP;myo-2p::mCherry]
Ex426[ges-1p::lbp-2SL2GFP;myo-2p::mCherry]
Ex545[ges-1p::nslbp3 Tg::sl2-GFP; myo-2p::mCherry]
Ex546[ges-1p::nslbp3 Tg::sl2-GFP; myo-2p::mCherry]
raxIs22[nlp-11Tg::sl2-GFP;myo-2p::mCherry]
nlp-11(rax51)
raxIs3[ges-1p::lipl-4::sl2-GFP; myo-2p::mCherry];nlp-11(rax51)
raxEx70[ges-1p::nlp-11::sl2-GFP; myo-2p::mCherry]
raxEx71[ges-1p::nlp-11::sl2-GFP; myo-2p::mCherry]
raxEx72[rab-3p::nlp-11; myo-2p::mCherry]
raxEx73[rab-3p::nlp-11; myo-2p::mCherry]
raxEx74[rab-3p::nlp-11; myo-2p::mCherry]
raxIs119 [lbp-3p::lbp-3::gfp, myo-2p::mCherry];nlp-11(rax51)
raxIs86[lbp-8p::lbp-8::3xflag::sl2-RFP; myo-2p::GFP]
raxIs86[lbp-8p::lbp-8::3xflag::sl2-RFP; myo-2p::GFP]; raxIs141[lbp-3p::lbp-3::sl2-GFP]
raxIs132[lbp-2p::lbp-2::sl2gf; myo-2p::mCherry]
raxIs119 [lbp-3p::lbp-3::gfp, myo-2p::mCherry];
raxIs114[sur-5p:lmp-1::RFP-3XHA;unc-76(+)]
raxEx509[ges-1p::lbp-3::RFP::sl2-GFP]
raxIs3[ges-1p::lipl-4::sl2-GFP; myo-2p::mCherry];
nre-1(hd20); lin-15b(hd126)
nre-1(hd20); lin-15b(hd126)
raxIs119 [lbp-3p::lbp-3::gfp, myo-2p::mCherry];egl-21(n476)
Is[ges-1p::RDE-1::unc54 3'UTR, myo-2p::RFP3];rde-1 (ne219)
nhr-49(nr2014)
nhr-80(tm1011)
raxIs119 [lbp-3p::lbp-3::gfp, myo-2p::mCherry];nhr-49(nr2041)
raxIs3[ges-1p::lipl-4::sl2-GFP; myo-2p::mCherry];nhr-49(nr2041)
raxIs3[ges-1p::lipl-4::sl2-GFP; myo-2p::mCherry];nhr-80(tm1011)
rab-3p::nhr-49::unc-54 3'UTR, myo-3p::mCherry
raxIs119 [lbp-3p::lbp-3::gfp, myo-2p::mCherry];nhr-49(nr2041); rab-3p::nhr-49::unc-54 3'UTR, myo-3p::mCherry
raxIs3[ges-1p::lipl-4::sl2-GFP; myo-2p::mCherry]; fat-3(wa22);lbp-3(rax60)
raxEx586 [ges-1p::lbp-3(chim)::sl2::gfp]
raxEx587 [ges-1p::lbp-3(chim)::sl2::gfp]
raxEx581 [ges-1p::lbp-3::gfp]
raxEx599 [ges-1p::lbp-3(chim)::3xHA::sl2::gfp]
raxEx600 [ges-1p::lbp-3(chim)::3xHA::sl2::gfp]
raxEx602 [rab-3p::GBP::SAX-7::sl2::mKate2]
raxEx603 [rab-3p::GBP::sl2::mKate2]
raxEx604 [nhr-49p::nhr-49::mKate2];raxIs22[nlp-11::sl2gfp]
raxEx603 [rab-3p::GBP::sl2::mKate2];raxEx581 [ges-1p::lbp-3::gfp]
raxEx509[ges-1p::lbp-3::RFP::sl2GFP];raxIs3[ges-1p::lipl-4::SL2GFP]
raxEx548[egl-21p::egl-21::sl2::GFP]
raxEx609 [ges-1p::nslbp-3::RFP::sl2GFP]
raxIs3[ges-1p::lipl-4::SL2GFP];raxIs103[sur-5p:lmp-1::RFP-3XHA;unc-76(+)]
raxIs103[sur-5p:lmp-1::RFP-3XHA;unc-76(+)]
raxEx402[ges-1p::lipl-4::sl2-GFP];rde-1 (ne219); Is[ges-1p::RDE-1::unc54 3'UTR, myo2p::RFP3]

| Wild animals | No wild animals were used in this study |
| Field-collected samples | No field-collected samples were used in this work |
| Ethics oversight | No ethical approval or guidance required |

Note that full information on the approval of the study protocol must also be provided in the manuscript.

