## [Peer Review File · Nature Cell Biology]

Peer Review Information

Journal: Nature Cell Biology

Manuscript Title: Lysosome Lipid Signaling from the Periphery to Neurons Regulates Longevity

Corresponding author name(s): Meng Wang

Reviewer Comments & Decisions:

Decision Letter, initial version:

Dear Professor Wang,

Thank you for submitting your manuscript "Lysosome Lipid Signaling from the Periphery to Neurons Regulates Longevity" to Nature Cell Biology. It has now been seen by 3 referees, who are experts in lysosome cell biology (referee 1); aging/metabolism/*C. elegans* (referee 2); and aging/*C. elegans* (referee 3). As you will see from their comments (attached below), they find this work of potential interest but have raised substantial concerns, which in our view would need to be addressed with considerable revisions before we can consider publication in Nature Cell Biology.

As always at the journal, we have discussed the reviewers' points in detail within the editorial team, including the chief editor. You will see that the reviewers identified some data that weren't clear or consistent and had similar questions and points of concern about the model. They shared technical and experimental concerns that are significant and valid. We feel that the reviews are overall constructive and that addressing them seriously with data and additional experimental work would strengthen the study. In particular, in our view, for reconsideration at NCB, it would be essential to:

- A) clarify the relationship between DAF-16 and LPL-4, Rev#1 point #4 and Rev#3 #8, as well as the relationship between *lipl-4* / *egl-21* / DAF-16 – Rev#3 points #7, 9
- B) clarify the type of nervous system cells at work; Rev#1 #1, Rev#2 #4
- C) investigate the mechanism of DGLA and LBP-3 action, Rev#1 #2, Rev#2 #2 and #3
- D) address whether supplementation of *lipl-4* Tg;*fat-3*(RNAi) animals with DGLA affects lifespan and expression of *egl-3* and *egl-21* in neurons or intestine – Rev#3 #19-20
- E) tackle the concerns with RNAseq results – Rev#3 #3, #15, #16

F) All other referee concerns pertaining to providing methodological details, clarifications and textual/figure changes, minor requests, data presentation issues, statistical analysis issues should also be addressed.

G) Finally please pay close attention to our guidelines on statistical and methodological reporting (listed below) as failure to do so may delay the reconsideration of the revised manuscript. In particular please provide:

- a Supplementary Figure including unprocessed images of all gels/blots in the form of a multi-page pdf file. Please ensure that blots/gels are labeled and the sections presented in the figures are clearly indicated.
- a Supplementary Table including all numerical source data in Excel format, with data for different figures provided as different sheets within a single Excel file. The file should include source data giving rise to graphical representations and statistical descriptions in the paper and for all instances where the figures present representative experiments of multiple independent repeats, the source data of all repeats should be provided.

We would be happy to consider a revised manuscript that would satisfactorily address these points, unless a similar paper is published elsewhere, or is accepted for publication in Nature Cell Biology in the meantime. We are committed to providing a fair and constructive peer-review process, so please feel free to contact me if you would like to discuss any of the referee comments further or anticipate any issues addressing the points above. Our typical revision period is 6 months; however, please let me know if you anticipate any pandemic-related challenges; we are happy to discuss the timeline for resubmission as needed.

Of note, although we agree with Rev #2 that additional analyses on the relative lipid abundances in the *lip1-4* overexpression and WT conditions (point #1) would provide valuable insights, we do not consider this point as essential to support the current conclusions as the points above. We encourage you to address it if technically possible and in particular if you may be able to answer this with the existing lipidomics analyses, but addressing it experimentally will not be indispensable for reconsideration of the manuscript at this journal.

- ensure that it conforms to our format instructions and publication policies (see below and www.nature.com/nature/authors/).
- provide a point-by-point rebuttal to the full referee reports verbatim, as provided at the end of this letter.
- provide the completed Editorial Policy Checklist (found here <https://www.nature.com/authors/policies/Policy.pdf>), and Reporting Summary (found here <https://www.nature.com/authors/policies/ReportingSummary.pdf>). This is essential for reconsideration of the manuscript and these documents will be available to editors and referees in the

2event of peer review. For more information see <http://www.nature.com/authors/policies/availability.html> or contact me.

Nature Cell Biology is committed to improving transparency in authorship. As part of our efforts in this direction, we are now requesting that all authors identified as 'corresponding author' on published papers create and link their Open Researcher and Contributor Identifier (ORCID) with their account on the Manuscript Tracking System (MTS), prior to acceptance. ORCID helps the scientific community achieve unambiguous attribution of all scholarly contributions. You can create and link your ORCID from the home page of the MTS by clicking on 'Modify my Springer Nature account'. For more information please visit www.springernature.com/orcid.

[REDACTED]

We hope that you will find our referees' comments and editorial guidance helpful. Please do not hesitate to contact me if there is anything you would like to discuss. Thank you again for considering NCB for your work.

Best wishes,

Melina

Melina Casadio, PhD
Senior Editor, Nature Cell Biology
ORCID ID: <https://orcid.org/0000-0003-2389-2243>

Reviewers' Comments:

Reviewer #1:
Remarks to the Author:

In this manuscript, the authors identified a fat-to-neuron lipid signaling pathway induced by lysosomal lipid metabolism to promote longevity in *C. elegans*. They demonstrated that LIPL-4-induced lysosomal lipolysis in the intestine up-regulates the polyunsaturated fatty acid DGLA, which binds to and triggers secretion of LBP-3, thus inducing the neuropeptide pathway in the neurons via the nuclear hormone receptor NHR-49 to promote longevity. The overall experiments are very well executed, and data are clear and convincing. I have some minor points that need to be addressed or

3clarified.

1. The author used neuronal expression but did not further specify the type of neurons that were used. Are specific types of neurons responsible for the lipid signaling and lifespan extension?
2. Does DGLA binding specifically trigger LBP-3 secretion? The authors may test the secretion of lipid-binding-defective LBP-3.
3. How is the LBP-3-DGLA signal received by neurons and relayed to NHR-49? The authors may include this point in the discussion.
4. I am a little confused about the *daf-16* RNAi data. The authors stated that *daf-16* RNAi did not suppress lifespan of *lipl-4* Tg worms (page 4: Supplementary Fig. 1C, Supplementary Table 4). But in Table S4, the *p*-value shows that the difference is statistically significant ($p < 0.001$).

Reviewer #2:

Remarks to the Author:

In this manuscript, Savini et al. show that lysosomal metabolism regulates an intestine-to-neuron lipid signaling pathway to extend lifespan in *Caenorhabditis elegans*. Specifically, the authors show that longevity is regulated by a cell-non-autonomous process that involves lysosomal lipolysis and secretion of a lipid binding protein (LBP-3) from the fat storage tissue (intestine). The lipid binding protein LBP-3 likely binds PUFAs, and this complex acts through the nuclear hormone receptor NHR-49 in neurons to upregulate the neuropeptide *nlp-11* in neurons, which confers lifespan extension.

This study is very interesting because it reveals a fat-to-nervous system axis and shows a cell-non-autonomous mechanism of PUFA action on neural peptides to control lifespan. This work has important implications for several fields, including lipid metabolism, longevity, and organ-to-organ communication, and will be of broad appeal to the readership of *Nature Cell Biology*. The manuscript is well written and the statistics used and conclusions drawn from the data presented are robust. Specifically, the authors do an excellent job of testing the function of and tissue-specificity of these genes using a combination of overexpression and loss of function single and double mutants. Addressing the following points would help strengthen the manuscript:

Major comments

1. It is interesting that *lipl-4* overexpression leads to a slight but significant increase in PUFAs and the accumulation of one MUFA. Could the authors provide more analyses on the relative lipid abundances in the *lipl-4* overexpression and WT conditions? During *lipl-4* overexpression, are the relative abundances of bound or unbound SFA, MUFA, and/or PUFAs changing? In addition, are TGs depleted due to constant lipolysis and/or is TG abundance steady but the side chains are depleted of PUFAs such as DGLA (20:3(n-6))?
2. The finding that LBP-3 can be exported from the intestine and that this secretion increases in the context of *lipl-4* overexpression is strong support for the author's tissue-tissue communication conclusion. Could the authors test how LBP-3 is acting on neurons, notably whether it is direct or indirect? Is the hypothesis that LBP-3 is exported to coelomocytes and then another molecule is secreted to neurons or is LBP-3 secreted and taken up by both coelomocytes and neurons? For

4example, to address this, could the authors investigate whether secreted LBP-3::RFP is present in neurons expressing nlp-11? Is there a known receptor for LBP-3?

3. It is interesting that DGLA leads to an increase in transcript level for neuropeptide processing. Could the authors test if this could be due to a direct effect of DGLA on neurons independent of its binding to LBP-3? Does supplementing DLGA increase the accumulation of LBP-3::RFP in coelomocytes?

4. Using the reporter of nlp-11::GFP expression or publicly available datasets for expression, could the authors test if nlp-11 is expressed in a clear neuron or glial subpopulation? Is NHR-49 expressed in the same subpopulation of neuron/glial cells as NLP-11?

Minor Comments

1. In Figure 1A, could the authors clarify what Gene Ratio or the two columns mean?

2. In the text or figure legends, it would be helpful to explain why the lifespan assays in Figure 1e had to be performed in the context of daf-16 RNAi.

3. In Figure 2I, could the coloring of the scale be changed to all blue for down-regulation, white for no change, and red for up-regulation? Also, could the authors clarify if the values shown are fold change or p-values?

4. In Figure 3E, could the authors discuss why neuropeptide machinery and nlp-11 transcripts are reduced in the context of a non-secreted LBP-3 protein?

5. Could the authors comment on why a single MUFA, 16:1n7 is increased in the context of lipl-4 overexpression?

Reviewer #3:

Remarks to the Author:

Here, Savini et al. use RNA-seq to explore the transcriptional differences between worms with LIPL-4 expressed only in the intestine (lipl-4 Tg) with wild-type worms. From these results, the authors chose to further study one of the enriched GO categories, neuropeptide signaling, which is in the upregulated set in the long-lived worms.

Neuron-specific RNAi of nlp-11 suppressed lipl-4 Tg longevity, and overexpression in neurons increased lifespan.

Next, the authors used lipidomic profiling and found that PUFAs are higher in the lipl-4 Tg worms, and they show that some of the PUFA enzymes (fat-1, fat-3) are required in the intestine for lipl-4 Tg's lifespan increase, and that expression of lbp-3 alone can increase lifespan. The level of LBP-3 protein fused with RFP is found in the coelomocytes, suggesting it might be secreted from intestine, which is further supported by the loss of function upon removal of the N-term secretory signal.

NHR-49 is required for the LBP-3 increase in lifespan, and its rescue in neurons suppresses this

5difference in lifespan, suggesting that neuronal NHR-49 acts downstream of LBP-3 to regulate lifespan. In the final section of the paper, the authors test the roles of PUFAs on LBP-3 secretion in *lipl-4* Tg animals, and find that LBP-3 binds to DGLA. Interestingly, supplementation of *lipl-4* Tg;*fat-3*(RNAi) animals with DGLA increased the expression of *egl-3* and *egl-21*. Whether there is an accompanying change in lifespan as a result of DGLA supplementation was not addressed by the authors. Overall, the authors show support for some parts of their model that LIPL-4 overexpression regulates lifespan through its regulation of PUFA and LBP trafficking to neurons and subsequent neuropeptide regulation, with significant weaknesses or inconsistencies throughout that can be addressed.

Comments:

1. The basis for the whole project isn't well justified. What is already known about LIPL-4? These authors previously published a whole paper on this protein and how it affects lifespan – are the current results consistent with that work and the results with LBP-8 and oleoylethanolamide? What was the impetus to understand the transcriptional regulation, given the conclusions of the previous Science paper? There is also no discussion at the end about how these two stories might fit together, either.
2. Please show the *lipl-4* Tg lifespan so that the readers can know why/whether it is interesting to determine the downstream mechanisms. The lifespan shown in the Supp looks as if the WT worms are sick, as they start dying quite early (day 8) and *lipl-4* Tg seems to specifically slow this death. Are the WT worms actually dying of matricide, which *lipl-4* Tg suppresses?
3. RNA-seq data needs a link to a public data set, and the authors should show a PCA or equivalent analysis to show reproducibility of samples, since only three of each were used for the comparison.
4. Did the authors examine the intestine, not just neurons, for expression of the genes they have studied? (Along those lines, "neuropeptide signaling pathway" isn't really unexpected, given previous work showing that insulins are regulated in the intestine in response to insulin and FOXO signaling.)
5. Hamilton, et al. (9) is an odd citation for the statement "Down-regulation of neuropeptide processing genes decreases the levels of mature ILPs, which can lead to reduced insulin/IGF-1 signaling that is known to prolong lifespan through the FOXO transcription factor9" as this is an RNAi screen paper, not a reference to work done specifically on ILPs and IIS/FOXO signaling, while there are entire papers studying ILP regulation of lifespan in *C. elegans* – e.g., papers on *daf-28*, *ins-7*, all of the ILPs (Pierce, et al.). Also, there are both agonist and antagonist ILPs, so this statement is also incorrect.
6. What is *egl-21*? An ILP? A neuropeptide? An enzyme that processes NLPs? It is not stated in the text. Now I see it in Fig. 1c, but it should have been mentioned in the text, and the justification for studying it better explained.
7. Perhaps I have this backwards, but if a gene is upregulated in the GO terms of a long-lived mutant, why would one expect it to live longer if it is knocked down ("Consistently, we found that the loss-of-function mutant of *egl-21* has extended lifespan, and the lifespan extension was suppressed by RNA interference (RNAi)..") – isn't this the wrong direction? If it is not, please explain to the readers how this direction is flipped. Along those lines, was *egl-21* actually increased in the RNA-seq data? It looks like it is not. Why then would one expect the loss of a carboxypeptidase that is not regulated by *lipl-4* Tg to have an effect in either direction?
8. The *lipl-4* Tg on *daf-16* RNAi result suggests that *lipl-4* is downstream of DAF-16, but this is not mentioned at all.

69. Again, it feels like a step has been skipped: why is egl-21 kd immediately being tested on daf-16 RNAi, instead of just comparing lipl-4 Tg vs lipl-4 Tg;egl-21? What is that result without eliminating daf-16? What is the goal of this experiment? The authors should better explain the logic of the work here.
10. The scale of the log-2 fold-change of I/NLPs is odd, as the two most highly expressed insulins look like they are oppositely regulated, not ~5-fold and 8-fold increased in the same direction. Perhaps a white/light blue/dark blue scale would be more intuitive to readers.
11. Statistical comparisons in some of the lifespans are to the wrong thing. One would like to know how the knockdown of a gene (RNAi) affects the mutant in question (lipl-4 Tg), not how the RNAi affects the wt vs the mutant. For example, in Figure 1f, the comparison between lipl-4 Tg on EV vs nlp-11 should be shown; same for Fig 1i.
12. Perhaps that information is in Supp Table 4, but that table was not included in the supplemental files.
13. Is there a Supp Table 3? Does that have the results of the RNAi screen mentioned in the text?
14. Is nlp-51 important? Compared to the extremely low (no error bars, so cannot tell if there is even a fold-change) fold change of nlp-11, nlp-51 seems highly differentially expressed, as are a few others.
15. Throughout, the RT-PCR results are not consistent with the RNA-seq data – why is this? Too few RNA-seq replicates?
16. Same issue in comparisons with RNAi of fat-1, fat-3, etc. RNAi – what the reader wants to know is whether the knockdown in the lipl-4 Tg EV vs RNAi is significant, not a repeat of the lipl-4 Tg vs WT information. Please fix.
17. How was the role of NHR-49 first found? Did the authors do a screen? (“In searching for factors mediating the transcriptional up-regulation of neuropeptide genes, we discovered that the nuclear hormone receptor NHR-49 acts in neurons downstream of LIPL-4- LBP-3 signaling.”) Where are these results?
18. The statistical comparison in Figure 3h lacks the relevant comparison between lbp-3 Tg and lbp-3 Tg;nhr-49. Also, please use a different color or more obvious difference between the lines for the lf and neu strains.
19. Does supplementation of lipl-4 Tg;fat-3(RNAi) animals with DGLA increase the expression of egl-3 and egl-21 in the neurons, or in the intestine?
20. Does this rescue also affect lifespan? Please show, since this seems to be the crux of the argument.
21. Is overexpression of LIPL-4 important? Does it happen in any examples of longevity mutants?

Writing/citations:

1. First sentence: Aging isn't a “progressively declining process” – if anything, aging accelerates with time. The authors probably mean that “Aging is the process of progressive decline” or that “Aging is the process of progressive decline”.
2. The first three references are self-citations of reviews that do not refer to the specific findings mentioned in the sentences (by other authors)– please replace these with primary literature citations that refer to the point of the sentence. If there is a limitation on references, please replace with more specific references information.
3. Some introductory sentences about PUFAs and their previously-described roles in lifespan regulation, particularly work done by other C. elegans labs, would be appropriate (e.g., Watts lab

7work, Brunet lab work on PUFAs and longevity), as would discussion of previous NHR-49 findings (e.g., Antebi lab) to put the findings in context.

AUTHOR CONTRIBUTIONS – must be included after the Acknowledgements, detailing the contributions

8of each author to the paper (e.g. experimental work, project planning, data analysis etc.). Each author should be listed by his/her initials.

Methods should be written concisely, but should contain all elements necessary to allow interpretation and replication of the results. As a guideline, Methods sections typically do not exceed 3,000 words. The Methods should be divided into subsections listing reagents and techniques. When citing previous methods, accurate references should be provided and any alterations should be noted. Information must be provided about: antibody dilutions, company names, catalogue numbers and clone numbers for monoclonal antibodies; sequences of RNAi and cDNA probes/primers or company names and catalogue numbers if reagents are commercial; cell line names, sources and information on cell line identity and authentication. Animal studies and experiments involving human subjects must be reported in detail, identifying the committees approving the protocols. For studies involving human subjects/samples, a statement must be included confirming that informed consent was obtained. Statistical analyses and information on the reproducibility of experimental results should be provided in a section titled "Statistics and Reproducibility".

All Nature Cell Biology manuscripts submitted on or after March 21 2016 must include a Data availability statement at the end of the Methods section. For Springer Nature policies on data availability see <http://www.nature.com/authors/policies/availability.html>; for more information on this particular policy see <http://www.nature.com/authors/policies/data/data-availability-statements-data-citations.pdf>. The Data availability statement should include:

- Accession codes for primary datasets (generated during the study under consideration and

designated as "primary accessions") and secondary datasets (published datasets reanalysed during the study under consideration, designated as "referenced accessions"). For primary accessions data should be made public to coincide with publication of the manuscript. A list of data types for which submission to community-endorsed public repositories is mandated (including sequence, structure, microarray, deep sequencing data) can be found here <http://www.nature.com/authors/policies/availability.html#data>.

- Unique identifiers (accession codes, DOIs or other unique persistent identifier) and hyperlinks for datasets deposited in an approved repository, but for which data deposition is not mandated (see here for details <http://www.nature.com/sdata/data-policies/repositories>).
- At a minimum, please include a statement confirming that all relevant data are available from the authors, and/or are included with the manuscript (e.g. as source data or supplementary information), listing which data are included (e.g. by figure panels and data types) and mentioning any restrictions on availability.
- If a dataset has a Digital Object Identifier (DOI) as its unique identifier, we strongly encourage including this in the Reference list and citing the dataset in the Methods.

We recommend that you upload the step-by-step protocols used in this manuscript to the Protocol Exchange. More details can be found at www.nature.com/protocolexchange/about.

All imaging data should be accompanied by scale bars, which should be defined in the legend. Cropped images of gels/blots are acceptable, but need to be accompanied by size markers, and to retain visible background signal within the linear range (i.e. should not be saturated). The boundaries of panels with low background have to be demarked with black lines. Splicing of panels should only be considered if unavoidable, and must be clearly marked on the figure, and noted in the legend with a statement on whether the samples were obtained and processed simultaneously. Quantitative comparisons between samples on different gels/blots are discouraged; if this is unavoidable, it should only be performed for samples derived from the same experiment with gels/blots were processed in parallel, which needs to be stated in the legend.

Figures should be provided at approximately the size that they are to be printed at (single column is 86 mm, double column is 170 mm) and should not exceed an A4 page (8.5 x 11"). Reduction to the scale that will be used on the page is not necessary, but multi-panel figures should be sized so that

10the whole figure can be reduced by the same amount at the smallest size at which essential details in each panel are visible. In the interest of our colour-blind readers we ask that you avoid using red and green for contrast in figures. Replacing red with magenta and green with turquoise are two possible colour-safe alternatives. Lines with widths of less than 1 point should be avoided. Sans serif typefaces, such as Helvetica (preferred) or Arial should be used. All text that forms part of a figure should be rewritable and removable.

- For line art, graphs, charts and schematics we prefer Adobe Illustrator (.AI), Encapsulated PostScript (.EPS) or Portable Document Format (.PDF). Files should be saved or exported as such directly from the application in which they were made, to allow us to restyle them according to our journal house style.
- We accept PowerPoint (.PPT) files if they are fully editable. However, please refrain from adding PowerPoint graphical effects to objects, as this results in them outputting poor quality raster art. Text used for PowerPoint figures should be Helvetica (preferred) or Arial.
- We do not recommend using Adobe Photoshop for designing figures, but we can accept Photoshop generated (.PSD or .TIFF) files only if each element included in the figure (text, labels, pictures, graphs, arrows and scale bars) are on separate layers. All text should be editable in 'type layers' and line-art such as graphs and other simple schematics should be preserved and embedded within 'vector smart objects' - not flattened raster/bitmap graphics.
- Some programs can generate Postscript by 'printing to file' (found in the Print dialogue). If using an application not listed above, save the file in PostScript format or email our Art Editor, Allen Beattie for advice (a.beattie@nature.com).

TABLES – main tables should be provided as individual Word files, together with a brief title and

11legend. For supplementary tables see below.

The total number of Supplementary Figures (not including the “unprocessed scans” Supplementary Figure) should not exceed the number of main display items (figures and/or tables (see our Guide to Authors and March 2012 editorial <http://www.nature.com/ncb/authors/submit/index.html#suppinfo>; <http://www.nature.com/ncb/journal/v14/n3/index.html#ed>). No restrictions apply to Supplementary Tables or Videos, but we advise authors to be selective in including supplemental data.

GUIDELINES FOR EXPERIMENTAL AND STATISTICAL REPORTING

REPORTING REQUIREMENTS – To improve the quality of methods and statistics reporting in our papers we have recently revised the reporting checklist we introduced in 2013. We are now asking all life sciences authors to complete two items: an Editorial Policy Checklist (found here <https://www.nature.com/authors/policies/Policy.pdf>) that verifies compliance with all required editorial policies and a reporting summary (found

12here <https://www.nature.com/authors/policies/ReportingSummary.pdf>) that collects information on experimental design and reagents. These documents are available to referees to aid the evaluation of the manuscript. Please note that these forms are dynamic 'smart pdfs' and must therefore be downloaded and completed in Adobe Reader. We will then flatten them for ease of use by the reviewers. If you would like to reference the guidance text as you complete the template, please access these flattened versions at <http://www.nature.com/authors/policies/availability.html>.

Author Rebuttal to Initial comments

Reviewers' Comments:

Reviewer #1:

Remarks to the Author:

13In this manuscript, the authors identified a fat-to-neuron lipid signaling pathway induced by lysosomal lipid metabolism to promote longevity in C. elegans. They demonstrated that LIPL4-induced lysosomal lipolysis in the intestine up-regulates the polyunsaturated fatty acid DGLA, which binds to and triggers secretion of LBP-3, thus inducing the neuropeptide pathway in the neurons via the nuclear hormone receptor NHR-49 to promote longevity. The overall experiments are very well executed, and data are clear and convincing. I have some minor points that need to be addressed or clarified.

Response: We really appreciate the reviewer's positive comment on our work.

1. *The author used neuronal expression but did not further specify the type of neurons that were used. Are specific types of neurons responsible for the lipid signaling and lifespan extension?*

Response: We agree with the reviewer that it would be interesting to identify the specific type(s) of neurons involved in the regulation. As suggested by the reviewer 2, we have used the *nlp-11* and *nhr-49* expression pattern to investigate this. Using the CenGenApp developed by the Hobert's group (PMID:34237253), we found that *nlp-11* expresses in 62 neuronal cell types. For *nhr-49*, it is detected in 120 neuronal cell types based on the CenGenApp. Together, there are 57 neuronal cell types overlapped between *nlp-11* and *nhr-49*. We have included a graphical representation in the Supplementary Fig. 3f and a Supplementary Table 6 to show these neuronal cell types. We have also generated a new transgenic strain expressing the NHR-49::mKate2 fusion protein driven by its endogenous promoter (new Fig. 3n), and crossed it with the *nlp11::GFP Tg* strain. We discovered many overlapping neurons between *nlp-11* and *nhr-49* (new Fig. 3n). It would be interesting to further narrow down one or few neuronal cell types from this list in future studies, but we think this categorization won't change the conceptual framework of the manuscript and is beyond the scope of the current work.

2. *Does DGLA binding specifically trigger LBP-3 secretion? The authors may test the secretion of lipid-binding-defective LBP-3.*

Response: We thank the reviewer for raising an interesting point to test whether DGLA regulates LBP-3 secretion. To address this point, we conducted DGLA supplementation experiments. We found that the DGLA supplementation restores the induction of LBP-3::RFP secretion in the *lip1-4 Tg* worms with the *fat-3* RNAi inactivation (new Fig. 4e-f), supporting the role of DGLA in triggering LBP-3 secretion.

We would love to try what the reviewer has suggested. Unfortunately, the structure of LBP-3 is currently lacking. We therefore do not have the information regarding the amino acids that are crucial for lipid binding and cannot design a lipid-binding-defective form of LBP-3. At the same time, based on our previous experience with LBP-8, even with the structural information, the alternation of the amino acids responsible for lipid binding could enlarge the lipid binding pocket and enable the protein to bind more types of lipids instead of disrupting lipid binding. Removal of the lipid-binding domain results in protein misfolding and degradation. Due to these technical limitations, we are unable to generate lipid-binding-defective LBP-3 and test its secretion.

3. *How is the LBP-3-DGLA signal received by neurons and relayed to NHR-49? The authors may include this point in the discussion.*

Response: Thank the reviewer for his/her suggestion to include additional discussion on how the LBP-3-DGLA signal is received by neurons and relayed to NHR-49. This point is now addressed in the revised discussion (lines 338-343).

In addition, we have conducted new experiments to confirm the interaction between LBP-3 and neurons. Direct visualization of LBP-3 uptake in neurons is not as easy as in coelomocytes. We thus took the advantage of a GFP nanobody (GBP) to capture secreted LBP-3::GFP and enhance the sensitivity of detection.

We first generated a transgenic line that expresses GBP fused with the extracellular domain of SAX-7 (GBP::SAX-7) in neurons and also expresses polycistronic mKate under the control of the same promoter. The GBP::SAX-7 fusion has been utilized by Dr. Kang Shen's group to visualize the protein release from PVD neuron dendrites [PMID:31735664]. Dr. Shen has generously shared the construct with us. We then crossed this line with the transgenic strain expressing intestinespecific LBP-3::GFP. As shown in new Supplementary Fig. 3e, we observed LBP-3::GFP signals in mKate positive neurons. As controls, we did not detect GFP signals in mKate positive neurons in the transgenic line without crossing with the intestine-specific LBP-3::GFP strain (new Supplementary Fig. 3c) nor in the intestine-specific LBP-3::GFP strain without neuron-expressing GBP (new Supplementary Fig. 3d). These results support that secreted LBP-3 from the intestine presents in a proximity close to the neuronal surface, where they are captured by GBP::SAX-7.

Endocytosis-mediated uptake of FABPs has been suggested in some kidney studies [PMID: 30401801, 15696188]. To test whether LBP-3 can be internalized into neuronal cells, we

generated another transgenic line to express GBP alone in neurons and crossed it with the intestine-specific LBP-3::GFP transgenic strain (Fig. 3g). We detected LBP-3::GFP signals in neuronal cell bodies (new Fig. 3h). This result supports that LBP-3 interacts with neurons and is internalized into neuronal cells, and we cannot rule out the existence of specific LBP-3 receptors on the neuronal membrane. In future studies, we will further investigate the detailed mechanism by which LBP-3 is internalized and keep searching for possible LBP-3 receptors.

Previous studies in both mammals and *C. elegans* have reported that intracellular FABPs can facilitate the nuclear delivery of lipid molecules that activate PPAR-alpha/NHR-49. Based on the new finding that LBP-3 is internalized into neuronal cells, we thus speculate that LBP-3-DGLA complex might directly transport into the nucleus to activate NHR-49. However, we could not rule out the possibility that within neuronal cells, secondary lipid signals are derived from internalized DGLA and activate NHR-49 in the nucleus. These discussions have been included in the revised manuscript (lines 338-343).

4. I am a little confused about the daf-16 RNAi data. The authors stated that daf-16 RNAi did not suppress lifespan of lipl-4 Tg worms (page 4: Supplementary Fig. 1C, Supplementary Table 4). But in Table S4, the p-value shows that the difference is statistically significant (p<0.001).

Response: We would like to clarify that our conclusion is that “the **lifespan extension** in the *lipl-4 Tg* worms is not suppressed by *daf-16 RNAi* inactivation (Supplementary Fig. 1d, Supplementary Table 5)”, but not “*daf-16 RNAi* did not suppress lifespan of *lipl-4 Tg* worms”. The *p-values* in the Supplementary Table 5 show that *daf-16 RNAi* significantly shortens the lifespans of both WT and *lipl-4 Tg* worms ($p < 0.001$, *daf-16 RNAi* vs. vector), but under either the vector control or *daf-16 RNAi* condition, the lifespan extension caused by *lipl-4 Tg* is significant ($p < 0.001$, *lipl-4 Tg* vs. WT). The percentages of lifespan extension (48.7%, 55.0% and 49.3%) in the vector control condition are comparable to those (54.7%, 84.7% and 66.3%) in the *daf-16 RNAi* condition.

Because of the reviewers’ confusion, we now added the percentages of lifespan extension in the supplementary Table 4 and 5 to further increase the clarify of data presentation.

Reviewer #2:

Remarks to the Author:

In this manuscript, Savini et al. show that lysosomal metabolism regulates an intestine-to-neuron lipid signaling pathway to extend lifespan in Caenorhabditis elegans. Specifically, the authors show that longevity is regulated by a cell-non-autonomous process that involves lysosomal

lipolysis and secretion of a lipid binding protein (LBP-3) from the fat storage tissue (intestine). The lipid binding protein LBP-3 likely binds PUFAs, and this complex acts through the nuclear hormone receptor NHR-49 in neurons to upregulate the neuropeptide nlp-11 in neurons, which confers lifespan extension.

This study is very interesting because it reveals a fat-to-nervous system axis and shows a cellnon-autonomous mechanism of PUFA action on neural peptides to control lifespan. This work has important implications for several fields, including lipid metabolism, longevity, and organo-organ communication, and will be of broad appeal to the readership of Nature Cell Biology. The manuscript is well written and the statistics used and conclusions drawn from the data presented are robust. Specifically, the authors do an excellent job of testing the function of and tissue-specificity of these genes using a combination of overexpression and loss of function single and double mutants.

Addressing the following points would help strengthen the manuscript:

Response: We thank the reviewer for finding our story interesting for different biological fields and appreciate her/his constructive suggestions to further improve our manuscript.

Major comments

1. *It is interesting that *lipl-4* overexpression leads to a slight but significant increase in PUFAs and the accumulation of one MUFA. Could the authors provide more analyses on the relative lipid abundances in the *lipl-4* overexpression and WT conditions? During *lipl-4* overexpression, are the relative abundances of bound or unbound SFA, MUFA, and/or PUFAs changing? In addition, are TGs depleted due to constant lipolysis and/or is? TG abundance steady but the side chains are depleted of PUFAs such as DGLA (20:3(n-6))*

Response: We agree with the reviewer that it is interesting to understand how PUFAs are induced in the free fatty acid pool. To directly examine whether and how this induction is related to lipid profile changes in the lysosome, we have purified lysosomes from both *lipl-4 Tg* and WT worms and conducted lipidomic profiling. We found that only the level of triacylglycerol (TAG) is significantly decreased (more than 3-fold) in the lysosome purified from the *lipl-4 Tg* worms compared to WT (new Supplementary Fig. 2a). Furthermore, among decreased TAG species, 63% of them contain PUFA side chains (new Fig. 2b). These results suggest that the increased level of PUFAs is likely due to an increased lipolysis of TAG in the lysosome.

2. *The finding that LBP-3 can be exported from the intestine and that this secretion increases in the context of *lipl-4* overexpression is strong support for the author's tissue-tissue communication conclusion. Could the authors test how LBP-3 is acting on neurons, notably whether it is direct or indirect? Is the hypothesis that LBP-3 is exported to coelomocytes and then another molecule is secreted to neurons or is LBP-3 secreted and taken up by both coelomocytes and neurons? For example, to address this, could the authors investigate whether secreted LBP-3::RFP is present in neurons expressing *nlp-11*? Is there a known receptor for LBP-3?*

Response: We appreciate the interest of the reviewer on the interaction between secreted LBP-3 and neurons. We agree with the reviewer that our hypothesis is that LBP-3 is secreted and taken up by both coelomocytes and neurons. To test this hypothesis, we examined whether LBP-3 is taken up by neurons. Direct visualization of LBP-3 uptake in neurons is not as easy as in coelomocytes. We thus took the advantage of a GFP nanobody (GBP) to capture secreted LBP3::GFP and enhance the sensitivity of detection.

We first generated a transgenic line that expresses GBP fused with the extracellular domain of SAX-7 (GBP::SAX-7) in neurons and also expresses polycistronic mKate under the control of the same promoter. The GBP::SAX-7 fusion has been utilized by Dr. Kang Shen's group to visualize the protein release from PVD neuron dendrites [PMID:31735664]. Dr. Shen has generously shared the construct with us. We then crossed this line with the transgenic strain expressing intestinespecific LBP-3::GFP. As shown in new Supplementary Fig. 3e, we observed LBP-3::GFP signals in mKate positive neurons. As controls, we did not detect GFP signals in mKate positive neurons in the transgenic line without crossing with the intestine-specific LBP-3::GFP strain (new Supplementary Fig. 3c) nor in the intestine-specific LBP-3::GFP strain without neuron-expressing GBP (new Supplementary Fig. 3d). These results support that secreted LBP-3 from the intestine presents in a close proximity with neuronal surface, where they are captured by GBP::SAX-7.

Currently, we are not aware of LBP-3 receptors. However, endocytosis-mediated uptake of FABPs has been suggested in some kidney studies [PMID: 30401801, 15696188]. To test whether LBP-3 can be internalized into neuronal cells, we generated another transgenic line to express GBP alone in neurons and crossed it with the intestine-specific LBP-3::GFP transgenic strain (Fig. 3g). We detected LBP-3::GFP signals in neuronal cell bodies (new Fig. 3h) This result strongly supports that LBP-3 interacts with neurons and is internalized into neuronal cells, and we cannot rule out the existence of specific LBP-3 receptors on the neuronal membrane. In future studies, we will further investigate the detailed mechanism by which LBP-3 is internalized

and keep searching for possible LBP-3 receptors. However, we believe these studies are beyond the scope of the current manuscript.

3. *It is interesting that DGLA leads to an increase in transcript level for neuropeptide processing. Could the authors test if this could be due to a direct effect of DGLA on neurons independent of its binding to LBP-3? Does supplementing DGLA increase the accumulation of LBP-3::RFP in coelomocytes?*

Response: We thank the reviewer for these suggestions.

To test whether the effect of DGLA on neurons is dependent on LBP-3, we have supplemented DGLA to the *lip1-4* *Tg* strain with both the *fat-3(lf)* and the *lbp-3(lf)* mutants. We found that in the absence of LBP-3, the DGLA supplementation fails to restore the neuropeptide gene induction (new Fig. 5a). Thus, DGLA requires LBP-3 to exert its effects on neurons. Based on our current knowledge of FABPs, this requirement highly supports that the effect of DGLA on neurons is due to its binding to LBP-3.

Ideally, we would like to generate a lipid-binding defective LBP-3 by amino acid mutation and examine whether the mutant blocks the effect of DGLA on neurons. However, this is technically impossible right now. 1) Because the structure of LBP-3 is currently lacking, we do not have the information regarding the amino acids that are crucial for lipid binding and cannot design a lipid-binding-defective form of LBP-3. 2) Based on our previous experience with LBP-8, even with the structural information, the alternation of those amino acids responsible for lipid binding could enlarge the lipid binding pocket and enable the protein to bind more types of lipids instead of disrupting lipid binding. 3) Removal of the lipid-binding domain results in protein misfolding and degradation.

To address the reviewer's question whether LBP-3 lipid binding is responsible for its effects, we have taken an alternative approach. LBP-2 and 3 are close FABP homologs. Based on our results, we know that LBP-3, but not LBP-2, is responsible for the regulation of neuropeptide genes (new Fig. 2g, 2k, 2n, 5b). We have analyzed the lipid binding preferences of LBP-2 and LBP-3 through profiling the *C. elegans* liposome that binds to LBP-2 and LBP-3 using mass spectrometry. We found that LBP-3 shows a distinct lipid binding specificity with a preference toward DGLA, and this preference was not detected with LBP-2 (new Fig. 5c, Supplementary Fig. 5a). Then using AlphaFold2, we predicted and compared their structures, and identified two cap-like alpha helices that are potentially crucial for lipid binding (new Fig. 5e). We thus designed a chimeric protein by replacing the two cap-like LBP-3 alpha helices from N38 to K60

19with those present in LBP-2 (new Fig. 5f, Supplementary Fig. 5b). We generated transgenic lines expressing the chimeric protein only in the intestine. Using western blot, we confirmed that the chimeric protein expresses normally (new Supplementary Fig. 5c). Next, we performed qRT-PCR and found that in this chimeric transgenic line, there is no significant change in neuropeptide transcript levels (new Fig. 5g and Supplementary Fig. 5d). This chimeric line also shows no lifespan extension (new Fig. 5h, Supplementary Fig. 5e). Together, these results suggest that the lipid binding specificity of LBP-3 is required for its regulation of neurons and longevity.

We also conducted DGLA supplementation experiments and examined the level of LBP-3::RFP in coelomocytes. We found that the DGLA supplementation is sufficient to restore the increased level of LBP-3::RFP in coelomocytes of the *lipl-4* *Tg* worms with the *fat-3* RNAi inactivation (new Fig. 4e-f).

*4. Using the reporter of *nlp-11::GFP* expression or publicly available datasets for expression, could the authors test if *nlp-11* is expressed in a clear neuron or glial subpopulation? Is *NHR-49* expressed in the same subpopulation of neuron/glial cells as *NLP-11*?*

Response: Thank the reviewer for her/his great suggestions on checking publicly available datasets. We have used the *nlp-11* and *nhr-49* expression pattern to investigate this. Using the CenGenApp developed by the Hobert's group [PMID:34237253], we found that *nlp-11* expresses in 62 neuronal cell types. Based on the *nlp-11::GFP* reporter generated in our group and the *nlp-11* expression pattern reported in publicly available datasets, no glial expression is detected. For *nhr-49*, it is detected in 120 neuronal cell types based on the CenGenApp. Together, there are 57 neuronal cell types overlapped between *nlp-11* and *nhr-49*. We have included a graphical representation in Supplementary Fig. 3f and a new Supplementary Table 6 to list these neuronal cell types. We also generated a new transgenic strain expressing the NHR49::mKate2 fusion protein driven by its endogenous promoter (new Fig. 3n), and crossed it with the *nlp-11::GFP* *Tg* strain. We discovered many overlapping neurons between *nlp-11* and *nhr-49* (new Fig. 3n). It would be interesting to further narrow down one or few neuronal cell types from this list in future studies, but we think this categorization won't change the conceptual framework of the manuscript and is beyond the scope of the current work.

Minor Comments

1. In Figure 1A, could the authors clarify what Gene Ratio or the two columns mean?

Response: We thank the reviewer for the suggestion on increasing the clarity of Figure 1A. The

Gene Ratio represents the number of differentially expressed genes associated with Gene Ontology biological processes over the total number of differentially expressed genes. The first column was from down-regulated genes, which causes confusion. We have now revised this figure panel with only one column to present the analysis based on the genes that are significantly upregulated in *lipl-4 Tg* (new Fig. 1b).

2. *In the text or figure legends, it would be helpful to explain why the lifespan assays in Figure 1e had to be performed in the context of daf-16 RNAi.*

Response: We appreciate the reviewer's advice. We have now included a short explanation in the figure legends when using *daf-16* RNAi.

3. *In Figure 2I, could the coloring of the scale be changed to all blue for down-regulation, white for no change, and red for up-regulation? Also, could the authors clarify if the values shown are fold change or p-values?*

Response: The old Fig. 2i includes neuropeptide genes that are significantly up-regulated in the *lipl-4 Tg* worms ($p < 0.05$). The color scale ranges from lowest induction (blue) to the maximum (red). This color choice is confusing based on the reviewer's comment. Thus, we have changed the color scale to Red Hot (new Fig. 2j). The values represented were expressed in Log2 Fold Change and all the changes are significant with $p < 0.05$. To avoid the confusion, we have changed the Log2 Fold Change into Fold Change and also removed $p < 0.05$ from the brackets.

4. *In Figure 3E, could the authors discuss why neuropeptide machinery and *nlp-11* transcripts are reduced in the context of a non-secreted LBP-3 protein?*

Response: Thank the reviewer for raising this interesting point. The current hypothesis that we have is that the non-secreted LBP-3 protein can still bind DGLA but sequester it inside the cell. With the overexpression of the non-secreted LBP-3, the number of DGLA available for the endogenous secretable LBP-3 to bind is decreased. As a result, the transcriptional levels of the neuropeptide genes are reduced. We have included new discussion regarding this point in the revised manuscript (lines 320-326).

5. *Could the authors comment on why a single MUFA, 16:1n7 is increased in the context of *lipl4* overexpression?*

Response: We agree with the reviewer that the induction of palmitoleic acid in the *lipl-4 Tg* worms is interesting. Previous studies reported that MUFA supplementation such as oleic acid, palmitoleic acid or vaccenic acid is sufficient to extend worm lifespan [PMID:28379943]. We did not detect the transcriptional induction of delta-9 desaturases *fat-5*, *6* and *7*, which are responsible for MUFA biosynthesis, in the *lipl-4 Tg* worms, and the levels of other MUFAs are not increased. Thus, the induction of palmitoleic acid is unlikely due to the overall up-regulation of MUFA biosynthesis via delta-9 desaturases. Furthermore, through profiling the liposome bound to LBP-3, we did not detect an enrichment of palmitoleic acid (Supplementary Fig. 5a). Therefore, although we could not rule out the possibility that palmitoleic acid contributes to *lipl4 Tg* longevity, it might not be involved in the LBP-3-mediated endocrine signaling mechanism.

We have included new discussion with a new reference regarding this point in the revised manuscript (lines 347-356).

Reviewer #3:

Remarks to the Author:

Here, Savini et al. use RNA-seq to explore the transcriptional differences between worms with LIPL-4 expressed only in the intestine (lipl-4 Tg) with wild-type worms. From these results, the authors chose to further study one of the enriched GO categories, neuropeptide signaling, which is in the upregulated set in the long-lived worms.

Neuron-specific RNAi of nlp-11 suppressed lipl-4 Tg longevity, and overexpression in neurons increased lifespan.

Next, the authors used lipidomic profiling and found that PUFAs are higher in the lipl-4 Tg worms, and they show that some of the PUFA enzymes (fat-1, fat-3) are required in the intestine for lipl-4 Tg's lifespan increase, and that expression of lbp-3 alone can increase lifespan. The level of LBP-3 protein fused with RFP is found in the coelomocytes, suggesting it might be secreted from intestine, which is further supported by the loss of function upon removal of the Nterm secretory signal.

NHR-49 is required for the LBP-3 increase in lifespan, and its rescue in neurons suppresses this difference in lifespan, suggesting that neuronal NHR-49 acts downstream of LBP-3 to regulate lifespan.

In the final section of the paper, the authors test the roles of PUFAs on LBP-3 secretion in lipl-4 Tg animals, and find that LBP-3 binds to DGLA. Interestingly, supplementation of lipl-4 Tg;fat3(RNAi) animals with DGLA increased the expression of egl-3 and egl-21. Whether there is an accompanying change in lifespan as a result of DGLA supplementation was not addressed by the authors.

Overall, the authors show support for some parts of their model that LIPL-4 overexpression regulates lifespan through its regulation of PUFA and LBP trafficking to neurons and subsequent neuropeptide regulation, with significant weaknesses or inconsistencies throughout that can be addressed.

Comments:

1. The basis for the whole project isn't well justified. What is already known about LIPL-4? These authors previously published a whole paper on this protein and how it affects lifespan – are the current results consistent with that work and the results with LBP-8 and oleoylethanolamide? What was the impetus to understand the transcriptional regulation, given the conclusions of the previous Science paper? There is also no discussion at the end about how these two stories might fit together, either.

Response: As shown in the *Science* paper, LIPL-4 is a lipase localized at the lysosome and its constitutive expression in the intestine, the fat storage tissue of *C. elegans* prolongs lifespan and triggers the nuclear translocation of LBP-8 and oleoylethanolamide to activate transcriptional response. Based on these findings, we are motivated to systemically profile transcriptional changes in the *lipl-4 Tg* worms. From the RNA-seq analysis, we detected the transcriptional upregulation of neuropeptide signaling genes. In responding to the reviewer's comment, we have revised the introductory paragraph (lines 59-65) to make this rationale clearer.

We also appreciate the reviewer's suggestion to discuss the interaction between the LBP-8 and the LBP-3 mediated mechanisms. In the revised manuscript, we have included new results showing that 1) *lbp-8 Tg* and *lbp-3 Tg* have additive effects in prolonging lifespan (new Fig. 3j); 2) the transcriptional changes of the neuropeptide genes in the *lbp-8 Tg* worms are less than 35% (new Fig. 3i). Based on these new results, we hypothesize that *lipl-4* overexpression induces both LBP-8-mediated cell-autonomous signaling and LBP-3-mediated cell-non-autonomous signaling to promote longevity. In addition to these new experimental results, new discussion has also been included in the revised manuscript (lines 363-365) to address the interaction between these two downstream mechanisms.

2. Please show the lipl-4 Tg lifespan so that the readers can know why/whether it is interesting to determine the downstream mechanisms. The lifespan shown in the Supp looks as if the WT worms are sick, as they start dying quite early (day 8) and lipl-4 Tg seems to specifically slow this death. Are the WT worms actually dying of matricide, which lipl-4 Tg suppresses?

23Response: We appreciate the reviewer's comments. Throughout the manuscript, the *lipl-4 Tg* lifespans have been always shown together with other conditions (Figure 1h, 1i, 1j, 1l, 2e, 2f, 2h), to make the comparison easy to follow. We are not sure which *lipl-4 Tg* lifespan is not shown. We guess that the reviewer suggested us to show the *lipl-4 Tg* lifespan at the beginning, and thus have included it as new Figure 1a.

When conducting longitudinal lifespan assays, it is common to have small fluctuation. Thus, it is very critical to run experiments for comparison at the same time, and repeat experiments in independent replicates. This is how we have conducted all the experiments. For lifespan comparison, the standard is to use mean lifespan from the Kaplan-Meier survival analysis. The day when animals start to die in a population is not an accurate measurement, because it could be largely affected by random death events.

Thanks to the reviewer's comment, we noticed that Supplementary Figure 1A has not be mentioned in the main text. This particular experiment was conducted using the *lipl-4* transgenic worms and their non-transgenic siblings without genome integration. Since this non-integrated line has not been used in the manuscript, we have removed this supplementary figure panel to avoid confusion.

3. RNA-seq data needs a link to a public data set, and the authors should show a PCA or equivalent analysis to show reproducibility of samples, since only three of each were used for the comparison.

Response: We appreciate the reviewers' suggestions. The RNA-seq data sets are now deposited into the NCBI Sequence Read Archive (SRA), SAMN25414087, SAMN25414088, SAMN25414089, SAMN25414090, SAMN25414091, SAMN25414002, SAMN25414093, SAMN25414094, SAMN25414095, SAMN25414096, SAMN25414097, SAMN25414098 and the public access will become available upon publication.

We have also conducted PCA analyses of independent replicates for *lipl-4 Tg* vs. WT and *lbp-3 Tg* vs. WT to show the reproducibility of samples, and the results are included in the revised manuscript (new Supplementary Fig. 1a, 2h).

4. Did the authors examine the intestine, not just neurons, for expression of the genes they have studied? (Along those lines, "neuropeptide signaling pathway" isn't really unexpected, given previous work showing that insulins are regulated in the intestine in response to insulin and FOXO signaling.)

Response: We appreciate the reviewer’s suggestion to examine the intestinal expression of the neuropeptide genes that we studied. In our studies, we showed that *egl-21* mediates the lifespan extension in the *lip1-4 Tg* worms and in the *lbp-3 Tg* worms (new Fig. 1h, Fig. 2l, and Supplementary Fig. 1b). To examine the expression of *egl-21*, we have generated its transgenic GFP reporter and confirmed that *egl-21* is specifically expressed in neurons but not the intestine (new Fig. 1g), which is consistent with the published result [PMID: 12657671]. We have also conducted smFISH to directly visualize *egl-21* mRNA transcripts and confirmed its exclusive expression in neurons (new Fig. 1f). For *nlp-11*, we have already showed that it is expressed in both neurons and the intestine. To characterize its functional tissue for regulating longevity, we have conducted both loss-of-function and gain-of-function studies. As shown in the manuscript, *nlp-11* inactivation in the intestine failed to suppress the lifespan extension in the *lip1-4 Tg* worms (new Fig. 1l), and the intestine-specific overexpression of *nlp-11* also failed to prolong lifespan (new Fig. 1n). On the other hand, *nlp-11* neuron-specific knockdown suppresses *lip1-4 Tg* longevity (new Fig. 1i) and neuronal overexpression of *nlp-11* is sufficient to prolong lifespan (Fig. 1m). Therefore, it is neuronal *nlp-11* that is involved in the longevity regulation.

As suggested by the reviewer, we have revised the main text and changed the sentence to “In addition to “immune response” and “defense response” gene ontology (GO) categories that are commonly associated with longevity regulation, we also discovered the enrichment of “neuropeptide signaling pathway” (lines 70-72).

5. Hamilton, et al. (9) is an odd citation for the statement “Down-regulation of neuropeptide processing genes decreases the levels of mature ILPs, which can lead to reduced insulin/IGF-1 signaling that is known to prolong lifespan through the FOXO transcription factor⁹” as this is an RNAi screen paper, not a reference to work done specifically on ILPs and IIS/FOXO signaling, while there are entire papers studying ILP regulation of lifespan in *C. elegans* – e.g., papers on *daf-28*, *ins-7*, all of the ILPs (Pierce, et al.). Also, there are both agonist and antagonist ILPs, so this statement is also incorrect.

Response: We appreciate the reviewer’s comment. We have revised the text as “Previous genomic RNAi screens in *C. elegans* found that the inactivation of *egl-3*, which encodes the proprotein convertase acting upstream of EGL-21 in neuropeptide processing, extends lifespan, which is suppressed by the inactivation of the *daf-16*/FOXO transcription factor. Consistently, we found that *egl-21(lf)* also has extended lifespan and this lifespan-extension requires *daf-16* (Supplementary Fig. 1c, Supplementary Table 5). Given that ILPs regulate lifespan and DAF-16/FOXO is the key mediator of longevity in the insulin/IGF-1 signaling pathway, the requirement of DAF-16 suggests that the longevity effect conferred by *egl-21(lf)* is possibly due

25to a reduction in agonist ILP maturation and insulin/IGF-1 signaling.” (lines 89-98) We have also included additional references on ILP regulation of longevity as suggested by the reviewer.

6. What is egl-21? An ILP? A neuropeptide? An enzyme that processes NLPs? It is not stated in the text. Now I see it in Fig. 1c, but it should have been mentioned in the text, and the justification for studying it better explained.

Response: We are sorry that the information regarding EGL-21 enzymatic activity in our main text was not clearly stated. We have revised the text to increase the clarity.

“Next, we examined the role of the neuropeptide signaling pathway in longevity regulation using the loss-of-function of *egl-21*, *egl-21(lf)*. The EGL-21 enzyme is required for neuropeptide processing by removing basic residues from the C-terminal of cleaved peptides,” (lines 84-86).

7. Perhaps I have this backwards, but if a gene is upregulated in the GO terms of a long-lived mutant, why would one expect it to live longer if it is knocked down (“Consistently, we found that the loss-of-function mutant of egl-21 has extended lifespan, and the lifespan extension was suppressed by RNA interference (RNAi)..”) – isn’t this the wrong direction? If it is not, please explain to the readers how this direction is flipped. Along those lines, was egl-21 actually increased in the RNA-seq data? It looks like it is not. Why then would one expect the loss of a carboxypeptidase that is not regulated by lipl-4 Tg to have an effect in either direction?

Response: We appreciate the reviewer’s comment. The lifespan extension caused by the *egl-21* mutant is not a flip of the direction but because EGL-21 processes both lifespan-extending and lifespan-shortening neuropeptides. Among three types of neuropeptides processed by EGL-21, one group consists of 40 insulin-like peptides (ILPs) that signal through the DAF-2/insulin/IGF-1 receptor. Some of these ILPs are agonists of DAF-2, and their loss in the *egl-21* mutant will cause reduced activation of DAF-2 that is known to extend lifespan in a DAF-16/FOXO dependent manner. Thus, the lifespan extension in the *egl-21(lf)* mutant is likely due to reduced insulin/IGF-1 signaling, which is supported by the result that the *daf-16* inactivation suppresses the lifespan extension in the *egl-21(lf)* mutant.

As shown in Fig. 1e and 2j, the transcript level of *egl-21* is induced by 2.1-fold and 1.9-fold in *lipl-4 Tg* and *lbp-3 Tg*, respectively when compared to WT (also shown in Supplementary Table 1 and Supplementary Table 2). The induction is further confirmed using qRT-PCR (new Fig. 1c, 2k, 2n). Using a log scale in the y-axis when presenting the RNA-seq data might have caused the

26confusion on the induction of *egl-21* gene expression in *lipl-4 Tg* and *lbp-3 Tg*. We have now revised the figure panels to increase the clarity of data presentation.

9. Again, it feels like a step has been skipped: why is egl-21 kd immediately being tested on daf16 RNAi, instead of just comparing lipl-4 Tg vs lipl-4 Tg;egl-21? What is that result without eliminating daf-16? What is the goal of this experiment? The authors should better explain the logic of the work here.

Response: As suggested by the reviewer, we have examined the lifespans of WT, *egl-21(lf)*, *lipl4 Tg*, and *lipl-4 Tg;egl-21(lf)*. As shown in new Supplementary Fig. 1b, *lipl-4 Tg* cannot further enhance the lifespan extension of the *egl-21(lf)* mutant, which together with the result with *daf16* RNAi supports the requirement of *egl-21* for the pro-longevity effect of *lipl-4 Tg*. Thank the reviewer for the suggestion.

There are three reasons to inactivate *daf-16*:

1. Confirming the lifespan-extending effect caused by ILP loss in the *egl-21(lf)* mutant. The previous study showed that the *egl-3* RNAi extends lifespan and this lifespan-extending effect is dependent on *daf-16* [PMID: 15998808]. The authors thus predicted that the lifespan extension caused by the *egl-3* inactivation is due to reducing ILP processing. When we detected the lifespan extension in the *egl-21(lf)* mutant, we thought the mechanism would be the same as the *egl-3* RNAi and thus examined the requirement of *daf-16* using its RNA inactivation. We confirmed that the lifespan extension caused by *egl-21(lf)* is dependent on *daf-16* (new Supplementary Fig. 1c).
2. Examining the involvement of ILPs in mediating the pro-longevity effect of *lipl-4 Tg*. ILPs are known ligand for insulin/IGF-1 signaling, and the longevity regulatory effect of insulin/IGF-1 signaling requires *daf-16*. To examine the interaction between *lipl-4 Tg* and insulin/IGF-1 signaling, we used the RNAi inactivation of *daf-16*. We discovered that *lipl-4 Tg* can extend lifespan in the *daf-16* knockdown background as effectively as in WT background (new Supplementary Fig. 1d), suggesting that ILP-related insulin/IGF-1 signaling does not contribute to *lipl-4 Tg* longevity.
3. Generating a cleaner background to examine the interaction between *lipl-4 Tg* and *egl-21*. EGL-21 regulates the processing of not only ILPs but also FMRFamide-related peptides (FLPs) and neuropeptide-like proteins (NLPs). The *daf-16* RNAi helps eliminate the contribution of ILPs via DAF-2 to lifespan regulation and examine the effects of NLPs and FLPs.

We have revised the text to better explain the logic behind the experimental design (lines 89103) and added notes to the figure legends as suggested by the reviewer #2.

*8. The *lipl-4* Tg on *daf-16* RNAi result suggests that *lipl-4* is downstream of DAF-16, but this is not mentioned at all.*

Response: Our results do not support that “*lipl-4* is downstream of DAF-16”. First, we would like to clarify that our conclusion is that “the **lifespan extension** in the *lipl-4* Tg worms is not suppressed by *daf-16* RNAi inactivation (Supplementary Fig. 1d, Supplementary Table 5)”. The *p-values* in the Supplementary Table 5 show that *daf-16* RNAi significantly shortens the lifespans of both WT and *lipl-4* Tg worms ($p < 0.001$, *daf16* RNAi vs. vector), but under either the vector control or *daf-16* RNAi condition, the lifespan extension caused by *lipl-4* Tg is significant ($p < 0.001$, *lipl-4* Tg vs. WT). The percentages of lifespan extension (48.7%, 55.0% and 49.3%) in the vector control condition are comparable to those (54.7%, 84.7% and 66.3%) in the *daf-16* RNAi condition.

Because of the reviewers’ confusion, we now added the percentages of lifespan extension in the supplementary Table 4 and 5 to further increase the clarity of data presentation.

10. The scale of the log-2 fold-change of I/NLPs is odd, as the two most highly expressed insulins look like they are oppositely regulated, not ~5-fold and 8-fold increased in the same direction. Perhaps a white/light blue/dark blue scale would be more intuitive to readers.

Response: We appreciate the reviewer’s suggestion. We have now adjusted the color scale as Red Hot to increase the clarity of the graphical data presentation.

*11. Statistical comparisons in some of the lifespans are to the wrong thing. One would like to know how the knockdown of a gene (RNAi) affects the mutant in question (*lipl-4* Tg), not how the RNAi affects the wt vs the mutant. For example, in Figure 1f, the comparison between *lipl-4* Tg on EV vs *nlp-11* should be shown; same for Fig 1i.*

Response: We appreciate the reviewer’s suggestion to present statistical comparisons between RNAi and EV controls. Both comparisons were included in the supplementary tables. In figure panels, we previously only included one comparison to avoid overcrowding. Based on the reviewer’s comment, we have now included both comparisons.

12. *Perhaps that information is in Supp Table 4, but that table was not included in the supplemental files.*

Response: We apologize that this information was not found by the reviewer. In addition to the comparison showed graphically in the figures, we also provided Supplementary Table 4 and 5 with distinct comparisons between genotypes and conditions utilized in the current manuscript.

13. *Is there a Supp Table 3? Does that have the results of the RNAi screen mentioned in the text?*

Response: We thank the reviewer for his/her comment. We have now included the RNAi screen results into the Supplementary Table 3.

14. *Is nlp-51 important? Compared to the extremely low (no error bars, so cannot tell if there is even a fold-change) fold change of nlp-11, nlp-51 seems highly differentially expressed, as are a few others.*

Response: We thank the reviewer for his/her comment. All the neuropeptide genes shown in Figure 1e are significantly up-regulated in the *lipl-4 Tg* worms with $p < 0.05$. Now we have included error bars in this panel. Although *nlp-51* is highly up-regulated, its RNAi activation does not suppress the lifespan extension in the *lipl-4 Tg* worms in the screen (Supplementary Table 3).

15. *Throughout, the RT-PCR results are not consistent with the RNA-seq data – why is this? Too few RNA-seq replicates?*

Response: The qRT-PCR results are consistent with the RNA-seq data. As shown in Fig. 1c and Fig. 1e, there is no significant difference in fold changes between RNA-seq and qRT-PCR results. Using a log scale in the y-axis when presenting the RNA-seq data might have caused the confusion. We have revised the figure to increase the clarity of data presentation.

16. *Same issue in comparisons with RNAi of fat-1, fat-3, etc. RNAi – what the reader wants to know is whether the knockdown in the lipl-4 Tg EV vs RNAi is significant, not a repeat of the lipl-4 Tg vs WT information. Please fix.*

Response: We appreciate the reviewer's suggestion. As previously stated in #11, we have now updated all RNAi lifespan graphs accordingly.

17. *How was the role of NHR-49 first found? Did the authors do a screen? (“In searching for factors mediating the transcriptional up-regulation of neuropeptide genes, we discovered that the nuclear hormone receptor NHR-49 acts in neurons downstream of LIPL-4- LBP-3 signaling.”) Where are these results?*

Response: We appreciate the reviewer’s interest on the discovery of NHR-49’s involvement.

NHR-49 was first found when we were trying to understand the interaction between LBP-8 and LBP-3 in regulating neuropeptide genes. To address this interaction, we have conducted three different experiments in parallel. One of them is to test whether the induction of neuropeptide genes by *lipl-4 Tg* or by *lbp-3 Tg* is dependent on NHR-49 and/or NHR-80. Our previous work showed that NHR-49 and NHR-80 act downstream of *lipl-4 Tg* and *lbp-8 Tg* to regulate transcription and longevity. We found that the loss-of-function mutant of NHR-49 fully suppresses the induction of the neuropeptide genes in *lipl-4 Tg* and *lbp-3 Tg*, but the loss-of-function mutant of NHR-80 has no such effect. We have now included these results in Fig. 3k-l.

The other two experiments that we conducted to examine the interaction between LBP-8 and LBP3 are:

1. We tested whether *lbp-8 Tg* and *lbp-3 Tg* have an additive effect in extending lifespan. The result showed an additive effect (new Fig. 3j), suggesting they act independently from each other to promote longevity.
2. We tested whether the neuropeptide genes are induced in the *lbp-8 Tg* worms. The results show that their changes are less than 35% (new Fig. 3i), suggesting a negligible role of *lbp8* in regulating the neuropeptide signaling pathway.

We did not originally include the results regarding the interaction between *lbp-8* and *lbp-3*, because we thought it is irrelevant to the regulatory mechanism of *lbp-3*. Based on the reviewer’s comments, we now think these results will make the model more complete and clearer and therefore include them in the revision. We want to thank the reviewer for his/her comments.

18. *The statistical comparison in Figure 3h lacks the relevant comparison between lbp-3 Tg and lbp-3 Tg;nhr-49. Also, please use a different color or more obvious difference between the lines for the lf and neu strains.*

Response: We thank the reviewer for his/her suggestions. We have added the comparison between *lbp-3 Tg* and *lbp-3 Tg;nhr-49(lf)* and changed the graph line colors to increase the clarity of data presentation.

*19. Does supplementation of *lipl-4 Tg;fat-3(RNAi)* animals with DGLA increase the expression of *egl-3* and *egl-21* in the neurons, or in the intestine?*

Response: We thank the reviewer for his/her comment. As discussed in #4, we have confirmed the specific expression of *egl-21* in neurons using both smFISH (new Fig. 1f) and transgenic reporters (new Fig. 1g). Furthermore, using smFISH, we confirmed that *egl-21* is specifically induced in neurons upon the DGLA supplementation (new Supplementary Fig. 4b-c). On the other hand, *egl-3* is expressed in both neurons and the intestine. To test whether its expression is induced in the intestine upon the DGLA supplementation, we have dissected out the intestine and conducted qRT-PCR analysis. We found that the DGLA supplementation does not increase the expression of *egl-3* in the intestine (new supplementary Fig. 4a).

20. Does this rescue also affect lifespan? Please show, since this seems to be the crux of the argument.

Response: We thank the reviewer for his/her suggestion. Based on the suggestion, we have supplemented DGLA to the *lipl-4 Tg* strain with *fat-3* RNAi and confirmed that the DGLA supplementation rescues the lifespan extension (new Fig. 4g and Supplementary Table 5).

21. Is overexpression of LIPL-4 important? Does it happen in any examples of longevity mutants?

Response: The *lipl-4* transcript level is up-regulated in the long-lived insulin/IGF-1 *daf-2* mutant and in the long-lived germline-deficient *glp-1* mutant [PMID: 18988854]. It is also induced upon fasting [PMID: 23392608] that is shown to prolong lifespan [PMID: 17081160, PMID:9789046, PMID:926867, PMID:12954481]. Therefore, increased *lipl-4* levels are important for longevity regulation. We have included new introduction and additional references regarding this (lines 5962).

Writing/citations:

1. *First sentence: Aging isn't a "progressively declining process" – if anything, aging accelerates with time. The authors probably mean that "Aging is the process of progressive decline" or that "Aging is the process of progressive decline".*

31Response: We thank the reviewer for the comment. We have now corrected the statement in the introduction of the main text.

2. *The first three references are self-citations of reviews that do not refer to the specific findings mentioned in the sentences (by other authors)– please replace these with primary literature citations that refer to the point of the sentence. If there is a limitation on references, please replace with more specific references information.*

Response: We thank the reviewer for the suggestion. We previously concerned the limitation on references and thus cited review articles. Based on the reviewer’s suggestion, we have cited three research articles on organelle and tissue crosstalk in longevity regulation.

1. Hughes, A. L. & Gottschling, D. E. An early age increase in vacuolar pH limits mitochondrial function and lifespan in yeast. *Nature* **492**, 261–265 (2012).
2. Durieux, J., Wolff, S. & Dillin, A. The cell-non-autonomous nature of electron transport chain-mediated longevity. *Cell* **144**, 79–91 (2011).
3. Follick, A. *et al.* Lysosomal signaling molecules regulate longevity in *Caenorhabditis elegans*. *Science* **347**, 83–86 (2015).

We have also cited three research articles on lipid signaling in organelle and tissue interactions.

1. Cao, H. *et al.* Identification of a Lipokine, a Lipid Hormone Linking Adipose Tissue to Systemic Metabolism. *Cell* **134**, 933–944 (2008).
2. Davis, O. B. *et al.* NPC1-mTORC1 Signaling Couples Cholesterol Sensing to Organelle Homeostasis and Is a Targetable Pathway in Niemann-Pick Type C. *Dev Cell* **56**, 260276.e7 (2021).
3. Follick, A. *et al.* Lysosomal signaling molecules regulate longevity in *Caenorhabditis elegans*. *Science* **347**, 83–86 (2015).

3. *Some introductory sentences about PUFAs and their previously-described roles in lifespan regulation, particularly work done by other C. elegans labs, would be appropriate (e.g., Watts lab work, Brunet lab work on PUFAs and longevity), as would discussion of previous NHR-49 findings (e.g., Antebi lab) to put the findings in context.*

Response: We thank the reviewer for the suggestion. We have now included introduction and discussion on the role of PUFAs in longevity regulation and discussed the previous findings on

32NHR-49 and other nuclear hormone receptors in longevity regulation. These references have been included:

1. Han, S. *et al.* Mono-unsaturated fatty acids link H3K4me3 modifiers to *C. elegans* lifespan. *Nature* **544**, 185–190 (2017).
2. Qi, W. *et al.* The ω -3 fatty acid α -linolenic acid extends *Caenorhabditis elegans* lifespan via NHR-49/PPAR α and oxidation to oxylipins. *Aging Cell* **16**, 1125–1135 (2017).
3. Chamoli, M. *et al.* Polyunsaturated fatty acids and p38-MAPK link metabolic reprogramming to cytoprotective gene expression during dietary restriction. *Nat Commun* **11**, 4865 (2020).
4. Ratnappan, R. *et al.* Germline Signals Deploy NHR-49 to Modulate Fatty-Acid β Oxidation and Desaturation in Somatic Tissues of *C. elegans*. *Plos Genet* **10**, e1004829 (2014).
5. Goudeau, J. *et al.* Fatty acid desaturation links germ cell loss to longevity through NHR80/HNF4 in *C. elegans*. *Plos Biol* **9**, e1000599 (2011).
6. Heestand, B. N. *et al.* Dietary Restriction Induced Longevity Is Mediated by Nuclear Receptor NHR-62 in *Caenorhabditis elegans*. *Plos Genet* **9**, e1003651 (2013).
7. Gems, D. *et al.* Two pleiotropic classes of *daf-2* mutation affect larval arrest, adult behavior, reproduction and longevity in *Caenorhabditis elegans*. *Genetics* **150**, 129–55 (1998).
8. Gerisch, B., Weitzel, C., Kober-Eisermann, C., Rottiers, V. & Antebi, A. A hormonal signaling pathway influencing *C. elegans* metabolism, reproductive development, and life span. *Dev Cell* **1**, 841–51 (2001).

Decision Letter, first revision:

18th February 2022

Dear Dr. Wang,

Thank you very much for submitting your revised manuscript "Lysosome Lipid Signaling from the Periphery to Neurons Regulates Longevity" (NCB-S45781A). It has now been seen by the original referees and their comments are below. The reviewers find that the paper has improved in revision, and therefore we'll be happy in principle to publish it in *Nature Cell Biology*, pending minor revisions to satisfy the referees' final requests and to comply with our editorial and formatting guidelines.

The current version of your manuscript is in a PDF format. Could you please email us a copy of the file in an editable format (Microsoft Word or LaTeX)? We can not proceed with PDFs at this stage.

33Once we receive the Word/editable article file, we will start performing detailed checks on your paper and will send you a checklist detailing our editorial and formatting requirements about a week after that. Please do not upload the final materials and make any revisions until you receive this additional information from us.

Thank you again for your interest in Nature Cell Biology. Please do not hesitate to contact me if you have any questions.

Sincerely,

Melina

Melina Casadio, PhD
Senior Editor, Nature Cell Biology
ORCID ID: <https://orcid.org/0000-0003-2389-2243>

Reviewer #1 (Remarks to the Author):

The authors have addressed all my concerns in the revised manuscript. I support publication of the study in NCB.

Reviewer #2 (Remarks to the Author):

This study is very interesting because it identifies a signal (involving the lysosome) from the periphery to neurons for the regulation of longevity. This work is exciting and has important implications for organ communication in several species. The authors have addressed all the points that were raised and the revised manuscript is significantly improved as a result.

A very minor point that remains regards the Principal Component Analysis in Extended Data Figure 1A: could the authors check if the PC axes might have been inverted or, alternatively, could they provide a more detailed explanation for it? It is surprising that the PC1 axis underlies so much of the variance yet the conditions separate mostly on the PC2 axis.

Reviewer #3 (Remarks to the Author):

Thank you for carefully responding to all of my critiques.

9th March 2022

34Dear Dr. Wang,

Thank you for your patience as we've prepared the guidelines for final submission of your Nature Cell Biology manuscript, "Lysosome Lipid Signaling from the Periphery to Neurons Regulates Longevity" (NCB-S45781A). Please carefully follow the step-by-step instructions provided in the attached file, and add a response in each row of the table to indicate the changes that you have made. Please also check and comment on any additional marked-up edits we have proposed within the text. Ensuring that each point is addressed will help to ensure that your revised manuscript can be swiftly handed over to our production team.

We would like to start working on your revised paper, with all of the requested files and forms, as soon as possible (preferably within one week). Please get in contact with us if you anticipate delays.

In recognition of the time and expertise our reviewers provide to Nature Cell Biology's editorial process, we would like to formally acknowledge their contribution to the external peer review of your manuscript entitled "Lysosome Lipid Signaling from the Periphery to Neurons Regulates Longevity". For those reviewers who give their assent, we will be publishing their names alongside the published article.

Nature Cell Biology offers a Transparent Peer Review option for new original research manuscripts submitted after December 1st, 2019. As part of this initiative, we encourage our authors to support increased transparency into the peer review process by agreeing to have the reviewer comments, author rebuttal letters, and editorial decision letters published as a Supplementary item. When you submit your final files please clearly state in your cover letter whether or not you would like to participate in this initiative. Please note that failure to state your preference will result in delays in accepting your manuscript for publication.

Cover suggestions

As you prepare your final files we encourage you to consider whether you have any images or illustrations that may be appropriate for use on the cover of Nature Cell Biology.

We accept TIFF, JPEG, PNG or PSD file formats (a layered PSD file would be ideal), and the image

35should be at least 300ppi resolution (preferably 600-1200 ppi), in CMYK colour mode.

Nature Cell Biology has now transitioned to a unified Rights Collection system which will allow our Author Services team to quickly and easily collect the rights and permissions required to publish your work. Approximately 10 days after your paper is formally accepted, you will receive an email in providing you with a link to complete the grant of rights. If your paper is eligible for Open Access, our Author Services team will also be in touch regarding any additional information that may be required to arrange payment for your article.

Please note that *Nature Cell Biology* is a Transformative Journal (TJ). Authors may publish their research with us through the traditional subscription access route or make their paper immediately open access through payment of an article-processing charge (APC). Authors will not be required to make a final decision about access to their article until it has been accepted. Find out more about Transformative Journals

Please use the following link for uploading these materials:
[REDACTED]

Best regards,

Nyx Hills
Staff
Nature Cell Biology

On behalf of

Melina Casadio, PhD
Senior Editor, Nature Cell Biology
ORCID ID: <https://orcid.org/0000-0003-2389-2243>

Reviewer #1:

Remarks to the Author:

The authors have addressed all my concerns in the revised manuscript. I support publication of the study in NCB.

Reviewer #2:

Remarks to the Author:

This study is very interesting because it identifies a signal (involving the lysosome) from the periphery to neurons for the regulation of longevity. This work is exciting and has important implications for organ communication in several species. The authors have addressed all the points that were raised and the revised manuscript is significantly improved as a result.

A very minor point that remains regards the Principal Component Analysis in Extended Data Figure 1A: could the authors check if the PC axes might have been inverted or, alternatively, could they provide a more detailed explanation for it? It is surprising that the PC1 axis underlies so much of the variance yet the conditions separate mostly on the PC2 axis.

Reviewer #3:

Remarks to the Author:

Thank you for carefully responding to all of my critiques.

Author Rebuttal, first revision:

Reviewers' Comments:

37Reviewer #1 (Remarks to the Author):

The authors have addressed all my concerns in the revised manuscript. I support publication of the study in NCB.

Response: We thank the reviewer for his/her suggestions and for supporting the publication of our manuscript at NCB.

Reviewer #2 (Remarks to the Author):

This study is very interesting because it identifies a signal (involving the lysosome) from the periphery to neurons for the regulation of longevity. This work is exciting and has important implications for organ communication in several species. The authors have addressed all the points that were raised and the revised manuscript is significantly improved as a result.

A very minor point that remains regards the Principal Component Analysis in Extended Data Figure 1A: could the authors check if the PC axes might have been inverted or, alternatively, could they provide a more detailed explanation for it? It is surprising that the PC1 axis underlies so much of the variance yet the conditions separate mostly on the PC2 axis.

Response: We thank the reviewer for his/her positive comments on the revised manuscript, and also his/her suggestions on the PCA analysis. Now, we de novo performed PCA analysis based on the expression of genes which are differentially expressed in *lip1-4* Tg compared to wild type (WT) worms (Fold change > 1.5 and $p < 0.05$), and Extended Data Figure 1a is updated.

Reviewer #3 (Remarks to the Author):

Thank you for carefully responding to all of my critiques.

Response: We appreciate the reviewer's suggestions that have helped us improve the manuscript.

Final Decision Letter:

Dear Dr Wang,

I am pleased to inform you that your manuscript, "Lysosome Lipid Signaling from the Periphery to Neurons Regulates Longevity", has now been accepted for publication in Nature Cell Biology.

Please note that *Nature Cell Biology* is a Transformative Journal (TJ). Authors may publish their research with us through the traditional subscription access route or make their paper immediately open access through payment of an article-processing charge (APC). Authors will not be required to make a final decision about access to their article until it has been accepted. Find out more about Transformative Journals

39Authors may need to take specific actions to achieve compliance with funder and institutional open access mandates. If your research is supported by a funder that requires immediate open access (e.g. according to Plan S principles) then you should select the gold OA route, and we will direct you to the compliant route where possible. For authors selecting the subscription publication route, the journal's standard licensing terms will need to be accepted, including self-archiving policies. Those licensing terms will supersede any other terms that the author or any third party may assert apply to any version of the manuscript.

If you have not already done so, we strongly recommend that you upload the step-by-step protocols used in this manuscript to the Protocol Exchange (www.nature.com/protocolexchange), an open online resource established by Nature Protocols that allows researchers to share their detailed experimental know-how. All uploaded protocols are made freely available, assigned DOIs for ease of citation and are fully searchable through nature.com. Protocols and Nature Portfolio journal papers in which they are used can be linked to one another, and this link is clearly and prominently visible in the online versions of both papers. Authors who performed the specific experiments can act as primary authors for the Protocol as they will be best placed to share the methodology details, but the Corresponding Author of the present research paper should be included as one of the authors. By uploading your Protocols to Protocol Exchange, you are enabling researchers to more readily reproduce or adapt the methodology you use, as well as increasing the visibility of your protocols and papers. You can also establish a dedicated page to collect your lab Protocols. Further information can be found at www.nature.com/protocolexchange/about

With kind regards,

Melina Casadio, PhD
Senior Editor, Nature Cell Biology
ORCID ID: <https://orcid.org/0000-0003-2389-2243>

40Click here if you would like to recommend Nature Cell Biology to your librarian
<http://www.nature.com/subscriptions/recommend.html#forms>

** Visit the Springer Nature Editorial and Publishing website at www.springernature.com/editorial-and-publishing-jobs for more information about our career opportunities. If you have any questions please click here.**